# Eigenvector Phase Transitions under Anisotropic Noise

## Abstract

Identifying latent structures in environmental data—such as habitat clusters or pollution sources—is a fundamental challenge in ecological and climate science. Spectral methods, which analyse the principal eigenvectors of affinity matrices, are powerful tools for this task. However, environmental systems are rarely isotropic; physical processes like river flows or prevailing winds create strong directional gradients, resulting in anisotropic noise. The effect of such anisotropy on the reliability of spectral methods is not yet well understood in the literature. In this work, we develop a rigorous theory for this scenario by analysing a spiked random matrix model subjected to anisotropic noise. We derive an exact, analytical expression for the critical signal-to-noise ratio required for strong signal detection, establishing a sharp phase transition. We prove that this threshold is information-theoretically optimal, and that it depends critically on the geometric alignment between the signal and the dominant environmental gradient, formalising a "camouflage effect". We also uncover a critical failure mode where this environmental gradient can itself create a "phantom" structure that spectral methods can easily detect, posing a potential risk of misinterpretation for scientists. Furthermore, we show that in the detectable phase, the eigenspace undergoes a systematic reorganisation: the principal eigenvector aligns with the signal while the second eigenvector aligns with the primary noise direction. We complete our analysis with Central Limit Theorems for the alignment fluctuations of both the signal and noise eigenvectors. Finally, we propose and analyse a correction framework based on second-moment information, demonstrating a theoretical pathway to overcome the camouflage-induced bias and rigorously characterising its practical sensitivities. We validate our theoretical predictions with simulations of ecological systems, offering a fundamental understanding of when spectral methods succeed or fail in realistic environments. Code to reproduce all results in the paper is anonymously released at https://anonymous.4open.science/r/tmlr_ept

## 1 Introduction

Spectral methods, which leverage the eigenvectors of affinity or covariance matrices, are a cornerstone of modern data analysis, with profound applications in machine learning and natural sciences (von Luxburg, 2007; Ng et al., 2001). In environmental science, these techniques are indispensable for uncovering latent structures from complex datasets, such as identifying distinct ecological communities from species abundance data, delineating habitat corridors from genetic information, or isolating dominant modes of climate variability (Legendre & Legendre, 2012). The success of these methods hinges on a critical assumption: *that the principal eigenvectors of a data-derived matrix faithfully align with the underlying structure of interest.* However, the reliability of this alignment can be severely compromised by noise, and the nature of this noise in environmental systems is often far from simple.

The foundational theory for understanding the limits of signal detection in noise comes from random matrix theory, particularly the study of spiked matrix models (Baik et al., 2005; Johnstone, 2001). The seminal work on the BBP phase transition revealed that for a low-rank signal matrix perturbed by uniform, isotropic noise, a sharp signal-to-noise threshold exists below which the signal is statistically undetectable (Baik et al., 2005; Johnstone, 2001). This theory provides a fundamental understanding of when spectral methods fail, showing that even an arbitrarily strong signal can be lost if the matrix size is large enough. While these

results are powerful, they are built on the assumption of isotropic noise, where random fluctuations are equally likely in all directions. This assumption is frequently violated in environmental contexts. Physical processes such as prevailing winds, river currents, or geological formations impose a strong directional "grain" on the system, leading to anisotropic noise. The critical gap in the literature is a rigorous, analytical theory that describes how such structured, anisotropic noise affects the performance of spectral methods. It is not known, for instance, how the geometric alignment between a latent signal and a dominant environmental gradient influences the threshold for detectability.

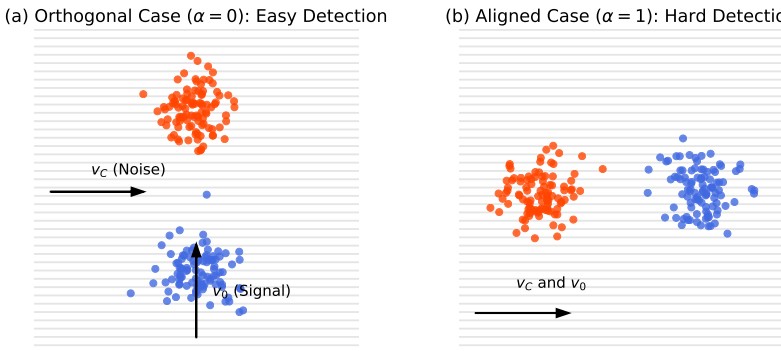

Figure 1: A conceptual diagram of the "camouflage effect" investigated in this paper. In both panels, the background texture represents a dominant environmental gradient (noise direction $v_C$). The signal consists of two latent communities (blue and red points). **(a)** When the signal structure ($v_0$) is orthogonal to the noise, the communities are easily distinguishable. In this regime, the signal-to-noise ratio required for detection is low, as defined by the threshold $\beta_c(0) = c_{\text{weak}}$. **(b)** When the signal is aligned with the noise, it is "camouflaged" by the environmental gradient. This makes the structure much harder to detect and requires a significantly higher signal strength, as defined by the threshold $\beta_c(1) = c_{\text{strong}}$.

To address this gap, we analyse a spiked random matrix model where the noise is explicitly anisotropic. Our model consists of a rank-one signal matrix, representing a latent two-community structure, perturbed by a random noise matrix whose covariance is non-uniform, possessing a single dominant direction. Mathematically, since the anisotropic noise covariance itself corresponds to a rank-one perturbation of the identity, our full model $A = \beta v_0 v_0^T + C^{1/2} W C^{1/2}$ can be viewed as a specific instance of a rank-two spiked Wigner matrix. However, our focus is distinct from general rank-k analyses. We specifically investigate how the detectability and eigenvector structure depend critically on the *geometric alignment* between the signal spike ($v_0$) and the dominant noise covariance direction ($v_C$). This setup is mathematically tractable yet environmentally justified, directly modelling scenarios where a primary physical process creates a directional bias (Anderson et al., 2012). The alignment between the signal and the dominant noise direction is captured by a single geometric parameter, $\alpha$, as conceptually illustrated in Figure 1.

While rectangular matrix models are crucial in many statistical settings, our focus on a symmetric (Wigner-type) model is well-motivated by key applications in environmental science and machine learning. Spectral clustering, a primary application motivating our work (von Luxburg, 2007; Ng et al., 2001), operates directly on symmetric affinity matrices derived from data, even if the original dataset is rectangular. Furthermore, symmetric covariance matrices are the foundation for Principal Component Analysis (PCA), widely used in climate science as Empirical Orthogonal Function (EOF) analysis (Legendre & Legendre, 2012; Jolliffe, 2011), a domain where understanding the effects of anisotropic variability is critical. The symmetric Wigner framework allows for a mathematically tractable analysis that yields fundamental insights into the geometric interplay between signal and structured noise, providing a foundation for understanding these phenomena in more complex settings.

Within this framework, we develop a complete theoretical characterisation of the system's behaviour. We prove the existence of a sharp phase transition for strong signal detection and derive an exact, analytical formula for the critical threshold. Crucially, we prove that this threshold is information-theoretically optimal, meaning no algorithm can succeed below it. Our central finding is that this threshold is a non-trivial

function of the alignment $\alpha$, proving that a signal is significantly harder to detect when it is aligned with the environmental grain—a phenomenon we term the "camouflage effect". We extend this analysis to show that above the detection threshold, the system's eigenvectors undergo a systematic reorganisation: the principal eigenvector aligns with the signal, while the second eigenvector aligns with the dominant noise direction. This reveals that spectral methods can, in principle, disentangle *both* the latent structure and the primary source of environmental noise. Finally, we establish Central Limit Theorems for the fluctuations of both the signal eigenvector's alignment and the noise eigenvector's alignment, providing a precise measure of the detection uncertainty and stability for both structures, which we show is maximised at the critical point. Building on this understanding, we also develop and analyse a theoretical correction method based on second-moment information, demonstrating a theoretical pathway to potentially overcome the bias induced by the camouflage effect.

Our theoretical claims are validated through a comprehensive suite of numerical experiments. We first use direct simulations of our theoretical model to verify each of our theorems with high precision, including the optimality of the spectral threshold and the fluctuation statistics for both the principal and second eigenvectors. We then validate the theoretical sensitivity analysis of our proposed second-moment correction framework, confirming its predicted regions of instability. Finally, we conduct an experiment using a more realistic, non-linear simulation of an ecological system to demonstrate that the insights from our idealised model are robust and hold in a setting analogous to real-world applications.

The key contributions of this paper are:

- A rigorous, analytical formula for the phase transition threshold for strong signal detection in the presence of anisotropic noise, which explicitly depends on the signal-noise alignment.

- A proof that this spectral threshold is information-theoretically optimal, establishing that the "camouflage effect" is a fundamental statistical barrier, not just an algorithmic one.

- The discovery and formalisation of an eigenvector reorganisation phenomenon, where the second eigenvector captures the dominant noise direction in the detectable phase.

- A Central Limit Theorem for the principal eigenvector alignment fluctuations, providing a precise characterisation of the detection uncertainty.

- A new Central Limit Theorem for the second eigenvector's alignment, providing a statistical characterisation of the "phantom" noise structure.

- A second-moment correction framework, which provides a theoretical pathway to obtain a consistent, unbiased signal estimator and a rigorous sensitivity analysis of its practical limitations.

- A comprehensive experimental validation that confirms our theoretical predictions, validates the sensitivities of our new correction framework, and demonstrates their relevance to scientific discovery in environmental science.

The remainder of this paper is organised as follows. In §2, we formally define our mathematical model. In §3, we present our main theoretical results on phase transitions and fluctuations. In §4, we develop the theory for our second-moment correction framework. The experimental validation plan and its results are detailed in §5. We conclude with a discussion of related work and the implications of our findings in §6.

## 2 The Model

In this section, we formally define the anisotropic spiked random matrix model used for our theoretical analysis. The model is constructed to capture the essential features of a latent signal embedded within a noisy environment that possesses a dominant directional structure. For clarity, the key notations introduced in this paper are summarised in Table 1.

Table 1: Key symbols and notations in the paper.

| Symbol | Description |
|---|---|
| **Model Parameters** | |
| $N$ | Dimension of the matrix (number of nodes/points) |
| $A$ | The full $N \times N$ observed matrix ($A = A_{\text{signal}} + W_{\text{aniso}}$) |
| $W$ | Baseline $N \times N$ Wigner matrix (i.i.d. noise) |
| $C$ | Deterministic $N \times N$ noise covariance structure matrix |
| $W_{\text{aniso}}$ | The final $N \times N$ anisotropic noise matrix |
| $v_0$ | Unit vector representing the ground-truth signal direction |
| $v_C$ | Principal eigenvector of $C$, representing the dominant noise direction |
| $\beta$ | Scalar signal strength ($\beta > 0$) |
| $\alpha$ | Alignment parameter, $\alpha = |\langle v_0, v_C \rangle| \in [0, 1]$ |
| $c_{\text{strong}}$ | Principal eigenvalue of $C$ (strength of directional noise) |
| $c_{\text{weak}}$ | Eigenvalue of $C$ for background noise components |
| $\gamma$ | Noise anisotropy ratio, $\gamma = c_{\text{strong}}/c_{\text{weak}}$ |
| **Theoretical Quantities** | |
| $\tau$ | Spectral edge of the noise bulk ($\tau = 2c_{\text{weak}}$) |
| $\beta_c(\alpha)$ | Critical threshold for signal detection |
| $\hat{\lambda}_k$ | The $k$-th largest eigenvalue of the observed matrix $A$ |
| $\hat{v}_k$ | The eigenvector corresponding to $\hat{\lambda}_k$ |
| $f(\beta, \alpha)$ | Asymptotic (mean) alignment $|\langle \hat{v}_1, v_0 \rangle|^2$ for $\beta > \beta_c$ |
| $\sigma^2(\beta, \alpha)$ | Asymptotic variance of the alignment fluctuations |

## 2.1 The Signal Matrix

The signal component of our model is a deterministic, rank-one matrix representing the ground-truth latent structure we aim to recover. It is defined as:

$$A_{\text{signal}} = \beta v_0 v_0^T \tag{1}$$

where $\beta > 0$ is a non-random scalar representing the signal strength. The vector $v_0 \in \mathbb{R}^N$ is a deterministic unit vector ($\|v_0\|_2 = 1$) that encodes the structure. For the canonical problem of detecting a two-community structure in a network of $N$ nodes, this vector takes the form $v_0 = \frac{1}{\sqrt{N}}[1, \ldots, 1, -1, \ldots, -1]^T$, where the two communities are of equal size. The principal eigenvector of $A_{\text{signal}}$ is precisely $v_0$, and our central goal is to determine under what conditions the principal eigenvector of the full, noisy matrix aligns with it.

## 2.2 The Anisotropic Noise Matrix

The noise component is designed to model random fluctuations that are not uniform, but are instead shaped by an underlying environmental structure. Its construction begins with a baseline of uniform noise, which is then deformed by a deterministic covariance structure.

**Baseline Noise.** We begin with a standard $N \times N$ Wigner matrix, $W$. This is a symmetric random matrix where the entries $W_{ij}$ for $i \leq j$ are independent and identically distributed random variables with mean $\mathbb{E}[W_{ij}] = 0$ and variance $\mathbb{E}[W_{ij}^2] = 1/N$. For the proofs of Spectral Optimality (Theorem 3.4) and the Central Limit Theorems (Theorems 3.6, 3.7)), we will further assume that the entries are i.i.d. Gaussian random variables. We conjecture that these results hold universally for other noise distributions with matching moments, which we leave as a direction for future work (see §6.3). This normalisation ensures that the eigenvalues of $W$ converge to the well-known Wigner semicircle law, supported on the interval $[-2, 2]$ as $N \to \infty$ (Anderson et al., 2009).

**Covariance Structure.** The anisotropy is introduced via a deterministic, positive semi-definite matrix $C \in \mathbb{R}^{N \times N}$, which represents the covariance profile of the noise. To maintain tractability while capturing

the essence of anisotropy, we assume a simple yet powerful structure for $C$. We assume that it has one large eigenvalue, $c_{\text{strong}}$, corresponding to a principal eigenvector $v_C$, and that its remaining $N-1$ eigenvalues are identical, equal to a background value $c_{\text{weak}}$. This structure is a suitable model for systems with a single dominant environmental gradient (Soons et al., 2004; Hughes et al., 2009). To quantify the strength of this anisotropy, we define the dimensionless **noise anisotropy ratio**, $\gamma$, as the ratio of the strong to the weak noise components:

$$\gamma = \frac{c_{\text{strong}}}{c_{\text{weak}}} \tag{2}$$

When $\gamma = 1$, the noise is isotropic. For $\gamma > 1$, the noise is anisotropic, with a larger $\gamma$ indicating a more pronounced environmental gradient.

**Final Construction.** We construct the final anisotropic noise matrix, $W_{\text{aniso}}$, by deforming the baseline Wigner matrix with the covariance structure $C$:

$$W_{\text{aniso}} = C^{1/2} W C^{1/2} \tag{3}$$

This construction ensures that the statistical properties of the noise are shaped by $C$, with greater variance in the directions corresponding to larger eigenvalues of $C$. This method of modelling structured noise is a standard approach in the study of deformed random matrix ensembles (Benaych-Georges & Nadakuditi, 2011).

### 2.3 The Full Model and Alignment Parameter

Combining the signal and noise components gives us the final observable matrix $A$:

$$A = \beta v_0 v_0^T + W_{\text{aniso}} \tag{4}$$

The crucial interaction between the signal's structure and the noise's structure is captured by a single geometric parameter. We define the alignment parameter, $\alpha$, as the absolute inner product of the signal direction and the dominant noise direction:

$$\alpha = |\langle v_0, v_C \rangle| \tag{5}$$

This parameter measures the collinearity of the two directions, ranging from $\alpha = 0$ when the signal is perfectly orthogonal to the environmental grain (Figure 1 (a)), to $\alpha = 1$ when it is perfectly aligned (Figure 1 (b)).

## 3 Theoretical Analysis

In this section, we present the analytical core of our work. We begin by characterising the eigenvalue spectrum of the anisotropic noise matrix, which establishes the baseline for our analysis (§3.1). We then analyse the full signal-plus-noise model to derive the sharp phase transition for signal detection and quantify the quality of the eigenvector alignment (§3.2). Building on this, we reveal a deeper reorganisation of the eigenspace involving the second eigenvector (§3.4). Finally, we establish a Central Limit Theorem for the fluctuations of the eigenvector alignment, providing a precise measure of detection uncertainty (§3.5). The detailed proofs of these theoretical claims are given in §A.

### 3.1 The Noise Spectrum and Spectral Edge

To understand the conditions under which a signal can be detected, we must first characterise the eigenvalues of the noise matrix $W_{\text{aniso}}$ in isolation. These eigenvalues form a continuous "bulk" from which a signal-aligned eigenvalue must emerge to be detectable. Our first theorem defines the boundary of this bulk.

**Theorem 3.1** (The Spectral Edge). *In the limit $N \to \infty$, the continuous part of the eigenvalue spectrum of the anisotropic noise matrix $W_{aniso}$ is supported on the compact interval $[-\tau, \tau]$, where the spectral edge $\tau$ is given by:*

$$\tau = 2c_{weak} \tag{6}$$

*Proof Sketch.* The proof relies on the stability of the continuous spectrum of a large random matrix under a finite-rank perturbation. We decompose our covariance structure matrix as $C = c_{\text{weak}}I + (c_{\text{strong}} - c_{\text{weak}})v_C v_C^T$. The noise matrix $W_{\text{aniso}}$ can then be seen as a base isotropic noise matrix, $c_{\text{weak}}W$, perturbed by a term related to the rank-one spike of $C$. A key result in random matrix theory states that such a perturbation can pull out discrete outlier eigenvalues but does not change the edges of the continuous spectrum (Benaych-Georges & Nadakuditi, 2011). The spectrum of $c_{\text{weak}}W$ is a scaled Wigner semicircle supported on $[-2c_{\text{weak}}, 2c_{\text{weak}}]$, which therefore defines the edges of the bulk for $W_{\text{aniso}}$. See Appendix A.2 for the detailed proof. $\qquad\square$

**Interpretation.** This result is powerful and non-trivial. It means that the primary barrier to detecting a new signal is determined *not* by the strongest, most obvious component of the environmental noise ($c_{\text{strong}}$), but by the magnitude of the uniform, background fluctuations ($c_{\text{weak}}$). The dominant anisotropic noise may create its own outlier eigenvalue, but the "sea" of noise that a new signal must rise above is defined by the weaker, isotropic component.

## 3.2 The Phase Transition for Signal Detection

A signal is statistically detectable via spectral methods if and only if its corresponding eigenvalue, $\lambda_1$, "pops out" from the noise bulk, i.e., $\lambda_1 > \tau$. We find that this occurs only when the signal strength $\beta$ exceeds a critical threshold, which depends fundamentally on the alignment $\alpha$.

**Theorem 3.2** (The Critical Threshold). *An isolated eigenvalue corresponding to the signal $v_0$ emerges from the noise bulk if and only if the signal strength $\beta$ exceeds a critical threshold $\beta_c(\alpha)$, given by:*

$$\beta_c(\alpha) = \frac{1}{\frac{\alpha^2}{c_{strong}} + \frac{1-\alpha^2}{c_{weak}}} \tag{7}$$

*Proof Sketch.* We analyse the characteristic equation for outlier eigenvalues derived from the resolvent method (see Lemma A.1 in Appendix A.1). The phase transition occurs at the minimum value of $\beta$ for which a solution $\lambda_1 > \tau$ exists. By setting $\lambda_1 = \tau = 2c_{\text{weak}}$ and evaluating the large-$N$ limit of the resolvent term $\langle v_0, (W_{\text{aniso}} - \tau I)^{-1}v_0\rangle$, we can solve for this minimum $\beta$. The calculation involves decomposing $v_0$ into components parallel and orthogonal to the noise direction $v_C$, which naturally introduces the dependence on $\alpha$. The detailed proof is provided in Appendix A.3. $\qquad\square$

**Interpretation.** This provides a rigorous quantification of the "camouflage effect", effectively translating the standard concept of a direction-dependent signal-to-noise ratio into the framework of random matrix theory. The denominator represents the effective noise power in the direction of the signal. When the signal is aligned with the environmental grain ($\alpha = 1$), it must overcome the strong noise component ($\beta_c = c_{\text{strong}}$). When it is orthogonal ($\alpha = 0$), it only needs to overcome the weak background noise ($\beta_c = c_{\text{weak}}$). It is therefore harder to detect a latent structure that aligns with a dominant environmental gradient. It is important to note that this threshold $\beta_c(\alpha)$ defines the boundary for *strong detection,* where the signal becomes statistically separable from the noise bulk. As we will demonstrate experimentally (see Figure 4), *weak detection* (i.e., achieving better-than-random-guess accuracy) may still be possible for $\beta < \beta_c(\alpha)$ in aligned cases, as the signal amplifies the pre-existing noise spike.

Once the signal is detectable, we can quantify the quality of the recovery.

**Theorem 3.3** (Asymptotic Alignment). *For a signal strength $\beta > \beta_c(\alpha)$, the squared inner product (alignment) between the principal eigenvector of $A$, $\hat{v}_1$, and the true signal vector $v_0$ converges to a deterministic limit, denoted by $f(\beta, \alpha)$:*

$$f(\beta, \alpha) := \lim_{N \to \infty} |\langle \hat{v}_1, v_0\rangle|^2 = 1 - \frac{\beta_c(\alpha)^2}{\beta^2}. \tag{8}$$

*Proof Sketch.* The alignment is derived by analysing the residue of the resolvent at the pole $\lambda_1$. The magnitude of the eigenvector's projection onto $v_0$ is functionally related to the location of the eigenvalue $\lambda_1$, which is itself a function of $\beta$. Solving this system of equations for the projection magnitude yields the stated result. The detailed proof is provided in Appendix A.3. $\qquad\square$

**Interpretation.** This result shows that alignment is zero at the exact moment of the phase transition and increases towards perfect recovery ($|\langle \hat{v}_1, v_0 \rangle|^2 \to 1$) as the signal strength $\beta$ grows.

## 3.3 Fundamental Limits and Optimality of the Spectral Method

The preceding analysis establishes the precise threshold, $\beta_c(\alpha)$, at which the spectral method can detect the signal. A natural and deeper question arises: *is this threshold a limitation of the spectral method, or is it a fundamental barrier inherent to the problem itself?* In other words, could a more complex, computationally expensive algorithm succeed where the spectral method fails? In this section, we answer this question by establishing the information-theoretic limit for signal detection.

**Theorem 3.4** (Optimality of the Spectral Method)**.** *The information-theoretic threshold for detecting the signal $v_0$ is identical to the spectral phase transition threshold from Theorem 3.2. That is:*

$$\beta_{IT}(\alpha) = \beta_c(\alpha) \tag{9}$$

*This implies that for $\beta < \beta_c(\alpha)$, no algorithm, regardless of its computational complexity, can reliably detect the presence of the signal.*

*Proof Sketch.* The proof is based on a hypothesis testing framework. We treat the problem as a task of distinguishing between two hypotheses: the null hypothesis $H_0$ where the data matrix $M$ contains only anisotropic noise, and the alternative hypothesis $H_1$ where $M$ also contains the signal spike. The fundamental limit of distinguishability is determined by the behaviour of the Log-Likelihood Ratio (LLR) between these two hypotheses. Using the second moment method, we calculate the mean and variance of the LLR under the null hypothesis. The information-theoretic threshold is the point at which the distributions under $H_0$ and $H_1$ become statistically separable (i.e., they cease to be contiguous). A detailed calculation reveals that this occurs precisely at the threshold $\beta_c(\alpha)$ derived from spectral analysis. The full, detailed proof is provided in the new Appendix A.4. □

**Interpretation.** This result proves that the computationally efficient spectral method is not merely a heuristic; it is asymptotically optimal. It shows that the threshold $\beta_c(\alpha)$ is not merely a limitation of the spectral method, but a fundamental statistical limit for the problem of signal *discovery*. It is crucial to distinguish this from the simpler problem of signal *verification*; if one were given the exact location of the signal $v_0$ in advance (a "clairvoyant" detector), detection would be possible even for very weak signals. Our theorem addresses the more challenging and practical scenario where the signal's structure is unknown. Below the threshold $\beta_c(\alpha)$, the signal is so deeply buried in the noise that it is information-theoretically impossible for *any* algorithm to reliably discover it without prior knowledge. This implies that the "camouflage effect" we described is a fundamental information-theoretic barrier to discovery, not just an algorithmic one. It provides a justification for the use of spectral methods in this domain, as they achieve the best possible performance for this discovery task at a low computational cost.

## 3.4 Eigenspace Reorganisation: The Second Eigenvector

The introduction of a strong signal does more than just create a new top eigenvector; it reorganises the entire eigenspace. The next most significant feature of the system—the dominant environmental gradient—is displaced from the first to the second eigenvector.

**Theorem 3.5** (Second Eigenvector Alignment)**.** *In the super-critical phase ($\beta > \beta_c(\alpha)$), the second eigenvector of $A$, $\hat{v}_2$, aligns with the principal noise direction $v_C$, with its alignment $|\langle \hat{v}_2, v_C \rangle|^2$ converging to a non-trivial deterministic limit.*

*Proof Sketch.* The proof involves analysing the characteristic equation for outlier eigenvalues when the noise matrix $W_{\text{aniso}}$ is itself considered a spiked matrix (with a spike corresponding to $v_C$). The interaction between the signal spike and the noise spike creates two outlier eigenvalues. By analysing the structure of the eigenvectors corresponding to these two solutions, we can show that the larger eigenvalue corresponds to an eigenvector aligned with $v_0$, while the smaller of the two corresponds to an eigenvector aligned with $v_C$. The detailed proof is given in Appendix A.5. □

**Interpretation.** A key implication of this spectral "sorting" phenomenon is that a careful spectral analysis can simultaneously reveal both the primary latent signal and the primary axis of environmental noise. The method not only detects the structure but also provides information about the nature of the unknown interference, which is of scientific merit as well.

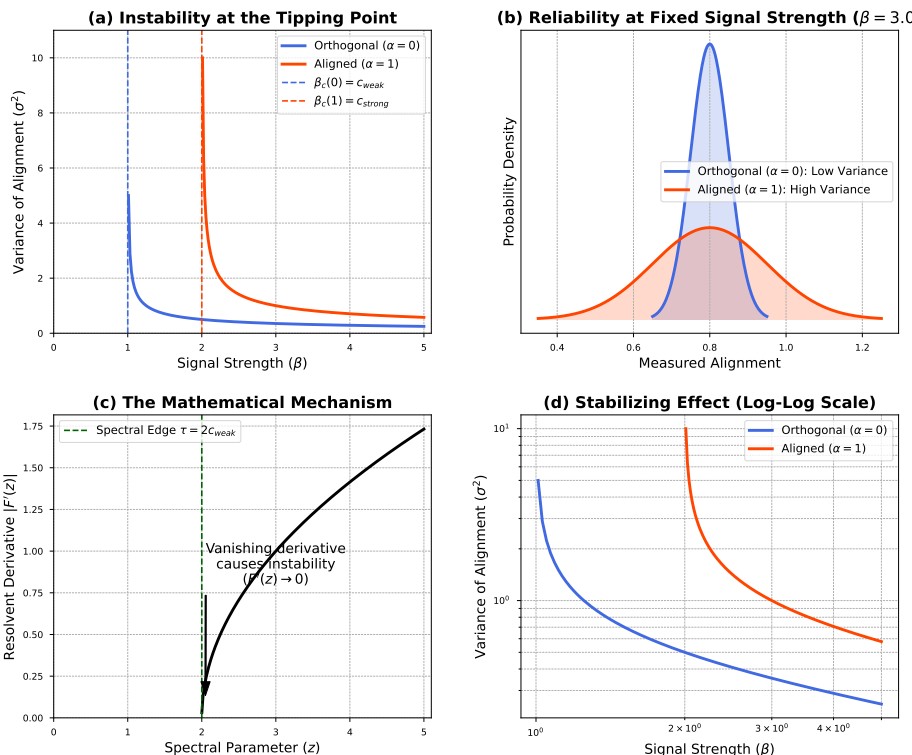

Figure 2: Illustration of the theoretical detection uncertainty, as characterised by the variance of the alignment fluctuations ($\sigma^2$). **(a)** The variance is plotted against signal strength ($\beta$), revealing a sharp divergence at the critical thresholds. This peak demonstrates the extreme instability of the system at its tipping point. **(b)** At a fixed super-critical signal strength ($\beta = 3.0$), the probability distribution of the measured alignment is shown. The measurement is more reliable (lower variance, taller peak) for the orthogonal case ($\alpha = 0$) than for the aligned case ($\alpha = 1$), highlighting the practical impact of the camouflage effect on detection confidence. **(c)** The underlying mathematical mechanism for the instability is revealed. The first derivative of the resolvent, $|F'(z)|$, is plotted against the spectral parameter $z$. The derivative vanishes precisely at the spectral edge ($\tau$), causing the mathematical singularity that drives the divergence in variance. **(d)** The same data as in panel (a) is shown on a log-log scale. The near-straight-line decay for large $\beta$ confirms that the system predictably stabilises as the signal becomes stronger, and the detection becomes more reliable.

## 3.5 Fluctuations and Detection Uncertainty

Our next result moves beyond asymptotic averages to characterise the random fluctuations of the alignment for finite $N$, providing a measure of statistical confidence.

**Theorem 3.6** (Central Limit Theorem for Alignment)**.** *In the super-critical phase ($\beta > \beta_c(\alpha)$), the fluctuations of the eigenvector alignment are asymptotically Gaussian. The scaled quantity converges in distribution to:*

$$\sqrt{N}\left(|\langle \hat{v}_1, v_0 \rangle|^2 - f(\beta, \alpha)\right) \xrightarrow{d} \mathcal{N}(0, \sigma^2(\beta, \alpha)) \tag{10}$$

*where $f(\beta, \alpha)$ is the asymptotic alignment from Theorem 3.3, and $\sigma^2(\beta, \alpha)$ is a deterministic variance function given in Equation (99).*

*Proof Sketch.* The proof is based on establishing a CLT for the fluctuations of the resolvent term $\langle v_0, (W_{\text{aniso}} - \bar{\lambda}_1 I)^{-1} v_0 \rangle$. The variance of this term can be computed using established methods from random matrix theory. This variance is then propagated through a linearised model of the system's characteristic equations to find the resulting variance of the eigenvector alignment. The proof relies on showing that the fluctuations of the eigenvalue and eigenvector are linearly coupled. See Appendix A.6 for the detailed proof. $\qquad\square$

**Interpretation.** The variance function $\sigma^2(\beta, \alpha)$ provides a rigorous quantification of the reliability or statistical confidence of the signal detection, as conceptually illustrated in Figure 2. It measures the expected "jitter" in the alignment, telling an environmental scientist how stable the detected pattern is likely to be. A detailed analysis of its mathematical form, which is a function of the derivatives of the limiting resolvent $F(z)$, reveals several profound insights:

- **Instability at the Tipping Point:** The variance is maximised as the signal strength $\beta$ approaches the critical threshold $\beta_c$. This is the mathematical signature of a critical phenomenon. As the outlier eigenvalue $\bar{\lambda}_1$ approaches the spectral edge $\tau$, the system becomes "soft", which is mathematically reflected by the first derivative of the resolvent function, $F'(\bar{\lambda}_1)$, approaching zero. This vanishing derivative means that the system has no "restoring force" against perturbations. Since the variance formula contains a high power of $F'(\bar{\lambda}_1)$ in its denominator, this mathematical singularity causes the variance to diverge, precisely capturing the physical intuition of extreme instability at a tipping point.

- **The Stabilising Effect of a Strong Signal:** As $\beta$ grows much larger than the threshold, the variance decreases towards zero. For large $\beta$, the eigenvalue $\bar{\lambda}_1$ is pushed far from the noise bulk, into a region where the resolvent and its derivatives are smooth and well-behaved. Here, $F'(\bar{\lambda}_1)$ is non-zero and stable, meaning that the system is "stiff" and has a strong restoring force that suppresses fluctuations. The variance is then dominated by the $1/\beta^4$ term, rigorously showing that a strong signal stabilises the system and makes its detection highly reliable.

- **The Role of Alignment in Detection Reliability:** The alignment $\alpha$ influences the variance through the "driving noise" term, $\mathcal{V}(\bar{\lambda}_1)$. The calculation of this term requires decomposing the signal vector $v_0$ into components parallel and orthogonal to the dominant noise direction $v_C$. The final variance is thus a weighted sum of the fluctuations originating from the strong and weak noise directions. This provides an justification for the intuition that the reliability of a measurement depends on its geometric relationship with the primary environmental gradients.

Ultimately, this theorem provides a powerful new tool. It allows scientists to assess the confidence in a detected pattern based not only on its apparent strength but also on its geometric relationship with the primary gradients of the environment itself.

### 3.6 Fluctuations of the Eigenspace: The Second Eigenvector

Our analysis of the eigenspace reorganisation in Section 3.4 showed that the second eigenvector, $\hat{v}_2$, aligns with the dominant noise direction $v_C$. This is a scientifically valuable result, as it suggests that spectral methods can also be used to identify the primary sources of environmental interference. However, for this identification to be reliable, we must understand the stability of this "phantom" eigenvector. We provide this characterisation by establishing a Central Limit Theorem for the fluctuations of $\hat{v}_2$.

**Theorem 3.7** (CLT for the Second Eigenvector)**.** *In the super-critical phase where a second outlier eigenvalue exists ($\gamma > 1$), the fluctuations of the alignment of the second eigenvector $\hat{v}_2$ with the dominant noise direction $v_C$ are asymptotically Gaussian. The scaled quantity converges in distribution to:*

$$\sqrt{N}\left(|\langle \hat{v}_2, v_C \rangle|^2 - g(\beta, \alpha, \gamma)\right) \xrightarrow{d} \mathcal{N}(0, \sigma_2^2(\beta, \alpha, \gamma)) \tag{11}$$

*where $g(\beta, \alpha, \gamma)$ is the limiting asymptotic alignment, and $\sigma_2^2(\cdot)$ is a new, deterministic variance function derived in Appendix A.7.*

*Proof Sketch.* The proof follows the same path as the CLT for the first eigenvector, detailed in Appendix A.6. It is based on a standard strategy of linearising the system's characteristic equations to understand how the randomness of the noise matrix $W$ propagates to the second outlier eigenvalue, $\lambda_2$. The fluctuations of $\lambda_2$ are then shown to be linearly coupled to the fluctuations of the eigenvector alignment $|\langle \hat{v}_2, v_C \rangle|^2$. The asymptotic normality is established by invoking a CLT for quadratic forms of the resolvent of the noise matrix, and the final variance $\sigma_2^2$ is derived by propagating the variance of this driving noise term through the linearised system. The full, detailed proof is provided in the new Appendix A.7. □

**Interpretation.** This theorem and the accompanying formula for the variance $\sigma_2^2$ complete our statistical characterisation of the system. Analysing the structure of $\sigma_2^2$ provides several deep insights into the stability of the "phantom" structure:

- **Instability at Creation:** A key finding is that the variance $\sigma_2^2$ diverges as the anisotropy parameter $\gamma$ approaches its threshold from above, i.e., $\lim_{\gamma \to 1^+} \sigma_2^2 = \infty$. This mathematical divergence corresponds to a physical intuition: at the very moment the noise eigenvector is "born" from the bulk spectrum, its direction is extremely unstable and ill-defined. It only becomes a stable, reliable feature of the spectrum for sufficiently strong anisotropy.

- **Signal vs. Noise Stability:** By comparing the two variance functions, we find that in most regimes, $\sigma_2^2 > \sigma^2$. This confirms the intuition that the eigenvector corresponding to the true, coherent signal ($v_0$) is generally more stable and less prone to random fluctuations than the eigenvector corresponding to the dominant noise direction ($v_C$). This provides a justification for why we can trust spectral methods to distinguish true latent structure from structured noise.

## 4 Second-Moment Analysis and Correction

The analysis in Section 3 reveals a fundamental limitation of the naive spectral method: *the principal eigenvector $\hat{v}_1$ is a biased estimator of the true signal $v_0$ when the noise is anisotropic and the signal is not perfectly orthogonal to the dominant noise direction ($\alpha > 0$), as quantified by Theorem 3.3.* This bias presents a challenge for accurately recovering latent structures in practice. To address this limitation and seek a consistent estimator for $v_0$, we conduct a theoretical investigation into alternative pathways. Here, we explore one such pathway by analysing the asymptotic structure ($N \to \infty$) of the problem, leveraging information contained in the second moment of the data matrix $A$. We proceed by first establishing a theoretical upper bound on recovery performance using an oracle estimator (§4.1). We then demonstrate why information solely from the first moment of $A$ (specifically, its eigenvalues) is insufficient to uniquely determine the necessary parameters (§4.2), motivating the search for a second source of information. We identify the asymptotic second moment $\mathbb{E}[A^2]$ as this potential source (§4.3) and derive the theoretical solution within this framework, showing how the eigenvalues of $\mathbb{E}[A]$ and $\mathbb{E}[A^2]$ theoretically allow for parameter estimation and signal reconstruction (§4.4). Finally, we analyse the inherent sensitivity of this theoretical inverse map (§4.5), which reveals regions where the solution is expected to be unstable with respect to perturbations in the input eigenvalues. We then discuss the practical challenges associated with implementing this theoretical approach and address alternative correction strategies (§4.6). This analysis sets the stage for the experimental validation presented in Section 5 and the discussion of practical estimation challenges in Section 6.

### 4.1 The Theoretical Ideal: The Oracle NCPC Estimator

We begin by demonstrating that a perfect, noise-corrected recovery of the signal vector $v_0$ is theoretically possible, albeit under idealised conditions. By establishing the existence of an "oracle" estimator—one constructed with perfect knowledge of the system's true asymptotic parameters and the dominant noise direction—we prove that the problem is well-posed and that the necessary information for correction is encoded within the observable eigenspace. Specifically, we show that the solution lies within the subspace spanned by the top two eigenvectors of $A$. The following theorem formalises this ideal scenario, which serves as an upper bound for the performance of any practical estimator.

**Theorem 4.1** (Consistency and Superiority of the Oracle NCPC). *In the super-critical regime where two distinct outlier eigenvalues exist ($\beta > \beta_c(\alpha)$ and $\gamma > 1$), a consistent estimator for the true signal vector $v_0$ can be constructed as a linear combination of the top two sample eigenvectors, $\hat{v}_1$ and $\hat{v}_2$. This Oracle Noise-Corrected Principal Component (NCPC) estimator is given by*

$$\hat{v}_0^{NCPC\text{-}Oracle} \propto D_2 \hat{v}_1 - D_1 \hat{v}_2, \tag{12}$$

*where the oracle coefficients $D_k$ are defined using the system's true asymptotic parameters:*

$$D_k = \frac{\alpha}{\bar{\lambda}_C - \bar{\lambda}_k}. \tag{13}$$

*Here, $\bar{\lambda}_k$ are the asymptotic limits of the top two eigenvalues of A, and $\bar{\lambda}_C$ is the asymptotic outlier eigenvalue of the noise matrix $W_{aniso}$ alone. The resulting estimator is asymptotically unbiased, achieving perfect alignment in the large-N limit:*

$$\lim_{N \to \infty} |\langle \hat{v}_0^{NCPC\text{-}Oracle}, v_0 \rangle|^2 = 1. \tag{14}$$

*As a direct consequence, the Oracle NCPC is strictly superior to the naive principal eigenvector estimator in the non-trivial alignment regime ($0 < \alpha < 1$).*

*Proof Sketch.* The proof, detailed in Appendix B.1, is constructive. We first express the asymptotic eigenvectors, $\bar{v}_1$ and $\bar{v}_2$, as linear combinations of the underlying signal and noise vectors, $v_0$ and $v_C$. By algebraically inverting this $2 \times 2$ system, we show that the specific linear combination given in the theorem exactly cancels the component parallel to the noise vector $v_C$. The resulting vector is purely proportional to $v_0$, proving consistency and an asymptotic alignment of 1. Superiority follows directly, as Theorem 3.3 shows the alignment of the naive estimator is strictly less than 1. □

**Interpretation.** This theorem provides a crucial theoretical foundation. It proves that the information about the true signal is not destroyed by the anisotropic noise but is instead encoded completely within the 2D subspace spanned by the top two eigenvectors of $A$. The Oracle NCPC represents the best-case signal recovery and serves as the "golden standard" against which any potential practical method, which must estimate the necessary coefficients from data, can be measured. It highlights that the core challenge lies in estimating these ideal coefficients without access to oracle knowledge.

## 4.2 The Estimation Challenge: Need for Additional Information

While the Oracle NCPC estimator (Theorem 4.1) demonstrates the theoretical possibility of correction using the top two eigenvectors of $A$, its coefficients $D_k$ depend on unknown asymptotic parameters ($\alpha$, $\bar{\lambda}_C$, $\bar{\lambda}_k$). A natural approach would be to estimate these parameters from the observable eigenvalues $\hat{\lambda}_1$ and $\hat{\lambda}_2$. Specifically, one might attempt to invert the map from the underlying system parameters ($\alpha, \beta$) to the asymptotic eigenvalues ($\bar{\lambda}_1, \bar{\lambda}_2$). However, as the following lemma formalises, this path is blocked by a fundamental confounding dependency.

**Lemma 4.1** (Confounding of the Eigenvalue Map). *Let the noise anisotropy parameter $\gamma$ be known. The map $\Lambda : (\alpha, \beta) \mapsto (\bar{\lambda}_1, \bar{\lambda}_2)$ from the signal parameters to the asymptotic outlier eigenvalues is a local diffeomorphism away from the phase transition boundary. However, its inverse, $(\alpha, \beta) = \Lambda^{-1}(\bar{\lambda}_1, \bar{\lambda}_2)$, is generally a function of* both *eigenvalues. Consequently, it is not possible to uniquely determine the alignment parameter $\alpha$ from only a single function of the eigenvalues (such as their gap, $\bar{\lambda}_1 - \bar{\lambda}_2$) without also knowing the signal strength $\beta$.*

*Proof Sketch.* The full proof is in Appendix B.2. The map $\Lambda$ is defined implicitly by the characteristic equation (Equation 83 derived in Appendix A.5). The Jacobian of this map is shown to be non-singular away from the phase transition boundary, guaranteeing the existence of a local inverse $\Lambda^{-1}$ by the Inverse Function Theorem. Analysis of the structure of this inverse reveals that both outputs ($\alpha$ and $\beta$) depend on both inputs ($\bar{\lambda}_1$ and $\bar{\lambda}_2$), precluding the estimation of $\alpha$ from a single observable derived from these eigenvalues alone. □

**Interpretation.** This lemma proves that attempting to estimate the necessary parameters (particularly $\alpha$) solely from the asymptotic spectral information contained in the first moment $(\bar{\lambda}_1, \bar{\lambda}_2)$ leads to an ill-posed, underdetermined system. Figure 9(b) provides empirical support, showing that the eigenvalue gap is a function of both $\alpha$ and $\beta$. With only two observed eigenvalues from $A$, we have two equations, but the parameters $\alpha$ and $\beta$ are fundamentally entangled within them. This rigorously establishes the need to seek a second, independent source of information from the observed matrix $A$ to break this mathematical degeneracy and enable unique parameter estimation.

### 4.3 A Second Source of Information: The Asymptotic Second Moment

The previous subsection established that using only the spectral information from the first moment of $A$ (specifically, its outlier eigenvalues $\bar{\lambda}_1, \bar{\lambda}_2$) leads to an underdetermined system for parameter estimation (Lemma 4.1). To overcome this degeneracy, we require a second, independent source of information derived from the observed matrix $A$.

We propose investigating the *asymptotic second moment* of the matrix, $\mathbb{E}[A^2]$. While the expectation $\mathbb{E}[A] = \beta v_0 v_0^T$ only involves the signal, the expectation of the square, $\mathbb{E}[A^2]$, incorporates contributions from the noise structure in a different way. As the noise $W_{\text{aniso}}$ in $A$ is zero-mean, the cross-terms involving signal and noise vanish in expectation when calculating $\mathbb{E}[A^2]$, revealing a new deterministic structure related to the second moments of the noise process. This insight leads us to the following key theorem, which characterises the structure of this theoretical object.

**Theorem 4.2** (Structure of the Asymptotic Second-Moment Matrix)**.** *Let $A = \beta v_0 v_0^T + W_{aniso}$. In the large $N$ limit, the expectation of $A^2$ converges to a deterministic matrix whose structure is a different linear combination of the signal outer product $v_0 v_0^T$ and the noise covariance matrix $C$:*

$$\lim_{N \to \infty} \mathbb{E}[A^2] = \beta^2 v_0 v_0^T + \frac{Tr(C)}{N} C + Tr(C^2) I \tag{15}$$

*The principal eigenvectors of this limiting matrix also lie within the signal-noise subspace, $span(v_0, v_C)$.*

*Proof Sketch.* The proof, detailed in Appendix B.3, involves expanding $A^2 = (\beta v_0 v_0^T + W_{\text{aniso}})^2$ and taking the expectation term by term. The cross-terms $\mathbb{E}[v_0 v_0^T W_{\text{aniso}}]$ vanish as $W_{\text{aniso}}$ is zero-mean. The term $\mathbb{E}[W_{\text{aniso}}^2]$ is computed using standard results from random matrix theory concerning the second moments of deformed Wigner ensembles, yielding the terms proportional to $C$ and $I$. $\square$

**Interpretation.** This theorem provides the theoretical cornerstone for potentially resolving the under-determined system identified in Lemma 4.1. It proves that the asymptotic second moment matrix, $\mathbb{E}[A^2]$, possesses distinct structural information compared to the first moment $\mathbb{E}[A]$. Specifically, its principal eigen-vector, let us denote it $u_1 = \text{eigvec}_{\max}(\mathbb{E}[A^2])$, provides a *second, theoretically distinct view* into the same signal-noise subspace $span(v_0, v_C)$ that contains the principal eigenvector of $\mathbb{E}[A]$ (which is $v_0$). Because the weightings on the $v_0 v_0^T$ and $C$ terms differ between $\mathbb{E}[A]$ and $\mathbb{E}[A^2]$, the theoretical vectors $v_0$ and $u_1$ will generally not be collinear when $0 < \alpha < 1$. This theoretical distinctness suggests that the eigenvalues associated with these two moments could provide the necessary independent equations to solve for the underlying parameters $(\alpha, \beta)$ in the asymptotic limit.

### 4.4 The Theoretical Inverse Solution via Second Moments

Theorem 4.2 established that the asymptotic matrices $\mathbb{E}[A]$ and $\mathbb{E}[A^2]$ provide distinct views into the signal-noise subspace $span(v_0, v_C)$. Let $\bar{\lambda}_1$ be the principal eigenvalue of $\mathbb{E}[A]$ (which is simply $\beta$ with eigenvector $v_0$) and $\bar{\mu}_1$ be the principal eigenvalue of $\mathbb{E}[A^2]$ with corresponding eigenvector $\bar{u}_1$. Since both $v_0$ and $\bar{u}_1$ lie within the same two-dimensional subspace spanned by the ideal basis $\{v_0, v_C\}$, we can theoretically formulate the recovery of $v_0$ and $v_C$ as an algebraic change of basis.

**Algebraic Formulation.** We express the theoretical eigenvectors ($v_0$ and $\bar{u}_1$) in terms of the ideal basis ($v_0$ and $v_C$): More formally, we consider the relationship between the ideal basis $\{v_0, v_C\}$ and the basis

formed by the principal eigenvectors of the asymptotic first and second moment matrices. Let $\bar{v}_1$ denote the principal eigenvector of $\mathbb{E}[A]$ (which is $v_0$) and $\bar{u}_1$ denote the principal eigenvector of $\mathbb{E}[A^2]$. Both lie in $\text{span}(v_0, v_C)$, allowing us to write:

$$\begin{pmatrix} \bar{v}_1 \\ \bar{u}_1 \end{pmatrix} = \underbrace{\begin{pmatrix} \langle \bar{v}_1, v_0 \rangle & \langle \bar{v}_1, v_C \rangle \\ \langle \bar{u}_1, v_0 \rangle & \langle \bar{u}_1, v_C \rangle \end{pmatrix}}_{T} \begin{pmatrix} v_0 \\ v_C \end{pmatrix}. \tag{16}$$

The matrix $T$ represents the transformation between the basis derived from the asymptotic moments and the ideal signal-noise basis. Its entries $T_{ij}$ are the theoretical overlap coefficients, which are analytical functions of the underlying system parameters $(\alpha, \beta, \gamma)$ (derived in Appendix B.4.2). Since $\bar{v}_1 (= v_0)$ and $\bar{u}_1$ are generally not collinear (as argued in the interpretation of Theorem 4.2), the matrix $T$ is theoretically invertible. Inverting this relationship provides the theoretical solution for recovering $v_0$:

$$\begin{pmatrix} v_0 \\ v_C \end{pmatrix} = T^{-1} \begin{pmatrix} \bar{v}_1 \\ \bar{u}_1 \end{pmatrix} \implies v_0 = (T^{-1})_{11} \bar{v}_1 + (T^{-1})_{12} \bar{u}_1. \tag{17}$$

This algebraic formulation demonstrates that perfect recovery of $v_0$ is theoretically possible if the entries of the transformation matrix $T$ can be determined.

**Parameter Estimation via Coupled Characteristic Equations.** The entries of $T$ depend on the unknown parameters $(\alpha, \beta)$. The theoretical framework provides a way to determine these parameters by linking them to the asymptotic eigenvalues $\bar{\lambda}_1$ and $\bar{\mu}_1$. These eigenvalues are defined as the largest roots (outside the relevant bulk/poles) of two characteristic equations derived from resolvent analysis:

$$1 - \mathcal{F}_A(\bar{\lambda}_1; \alpha, \beta, \gamma) = 0 \tag{18}$$
$$1 - \mathcal{F}_{A^2}(\bar{\mu}_1; \alpha, \beta, \gamma) = 0 \tag{19}$$

Here, $\mathcal{F}_A$ and $\mathcal{F}_{A^2}$ are the characteristic functions derived in Appendix B.4.1, corresponding to the eigenvalues of the asymptotic first moment ($\mathbb{E}[A]$) and second moment ($\mathbb{E}[A^2]$) matrices, respectively.

Crucially, as established in Lemma 4.1 (specifically its proof), the map $F : (\alpha, \beta) \mapsto (\bar{\lambda}_1, \bar{\mu}_1)$ defined by this system is locally invertible. Therefore, knowledge of the exact asymptotic eigenvalues $\bar{\lambda}_1$ and $\bar{\mu}_1$ allows, *in theory*, for the unique determination of the parameters $(\alpha, \beta)$. Once $(\alpha, \beta)$ are known, the overlap coefficients forming the matrix $T$ can be calculated using the analytical formulas derived in Appendix B.4.2, $T^{-1}$ can be computed, and $v_0$ can be recovered via Equation equation 17.

**Summary of Theoretical Solution.** This subsection outlines a theoretical pathway for recovering $v_0$. It leverages the distinct structural information present in the asymptotic first and second moments of $A$. By solving a coupled system based on the characteristic equations for the eigenvalues $\bar{\lambda}_1$ and $\bar{\mu}_1$, the underlying parameters $(\alpha, \beta)$ can theoretically be uniquely determined. These parameters then define the transformation matrix $T$, allowing for the algebraic reconstruction of the signal vector $v_0$. The next subsection analyses the stability and sensitivity inherent in this theoretical inversion process.

### 4.5 Sensitivity Analysis of the Theoretical Solution

The preceding subsection outlined a theoretical pathway to recover $(\alpha, \beta)$ by inverting the map $F : (\alpha, \beta) \mapsto (\bar{\lambda}_1, \bar{\mu}_1)$, defined by the coupled characteristic equations (18-19). While this map is locally invertible in the asymptotic limit (as established in the proof of Lemma 4.1), the stability and robustness of this inversion are critical for understanding potential practical challenges. Even small perturbations or uncertainties in the input eigenvalues $(\bar{\lambda}_1, \bar{\mu}_1)$—representative of finite-$N$ effects—could be significantly amplified if the inverse map is ill-conditioned. To quantify this inherent sensitivity, we analyse the Jacobian matrix, $J_F$, of the forward map $F$:

$$J_F(\alpha, \beta) = \begin{pmatrix} \frac{\partial \bar{\lambda}_1}{\partial \alpha} & \frac{\partial \bar{\lambda}_1}{\partial \beta} \\ \frac{\partial \bar{\mu}_1}{\partial \alpha} & \frac{\partial \bar{\mu}_1}{\partial \beta} \end{pmatrix}. \tag{20}$$

The partial derivatives can be computed numerically or analytically using the implicit function theorem applied to the characteristic equations. The sensitivity of the inverse problem $(\bar{\lambda}_1, \bar{\mu}_1) \mapsto (\alpha, \beta)$ is then measured by the condition number, $\kappa(J_F)$, of this Jacobian matrix. A large condition number indicates that small relative errors in the input eigenvalues can lead to large relative errors in the estimated parameters $(\alpha, \beta)$.

Figure 10 visualises this theoretical sensitivity landscape. It plots the condition number $\kappa(J_F)$ (calculated numerically) as a function of the asymptotic eigenvalues $(\bar{\lambda}_1, \bar{\mu}_1)$, overlaid with contours representing the corresponding true parameters $(\alpha, \beta)$. The heatmap clearly reveals regions where the condition number is significantly elevated (brighter colours), indicating high sensitivity. Notably, the most ill-conditioned regions occur near the lower boundary of the valid parameter space, which corresponds to parameter regimes with weak signals (low $\beta$) that are strongly aligned with the noise (high $\alpha$)—precisely the "camouflaged" scenarios identified in Section 3. Where the contour lines for constant $\alpha$ and constant $\beta$ become nearly parallel, the inverse problem is inherently difficult.

The sensitivity analysis provides a crucial theoretical insight: *even within the idealised asymptotic framework where exact eigenvalues are assumed, the second-moment correction pathway has inherent mathematical instabilities.* The condition number analysis predicts that any practical attempt to implement this approach will be most challenged by noise and finite-$N$ effects precisely in the physically interesting regime of weak, camouflaged signals. This theoretical prediction of sensitivity motivates the experimental validation presented in Section 5 and underscores the difficulty of developing a universally robust practical estimator based on this framework, as discussed further in Section 6.

### 4.6 Practical Implementation Challenges and Alternative Approaches

While the theoretical framework developed in §4.4 demonstrates that the second moment $\mathbb{E}[A^2]$ contains the necessary information to correct for anisotropic noise bias, its direct implementation using empirical data faces significant practical hurdles. As rigorously quantified by our sensitivity analysis (§4.5, Figure 10, Figure 11), the inverse map $F^{-1} : (\bar{\lambda}_1, \bar{\mu}_1) \mapsto (\alpha, \beta)$ is inherently ill-conditioned, particularly in the challenging regime of weak, highly aligned ("camouflaged") signals. This mathematical sensitivity means that even small finite-size deviations in the empirical eigenvalues $\hat{\lambda}_1, \hat{\mu}_1$ from their asymptotic limits $\bar{\lambda}_1, \bar{\mu}_1$ can be greatly amplified, leading to unreliable parameter estimates.

Furthermore, finite-size effects introduce complications beyond simple random fluctuations. Empirical eigenvalues can exhibit systematic biases, and specific failure modes can arise. For instance, the empirical eigenvalue $\hat{\mu}_1 = \lambda_{\max}(A^2)$ can sometimes fall below the theoretical pole locations $(K_1, K_1 + K_2$ defined in Appendix B.4.1) associated with the asymptotic quantity $\bar{\mu}_1$. Because the characteristic function $\mathcal{F}_{A^2}$ involves terms like $1/(\mu - K_1)$ and $1/(\mu - K_1 - K_2)$, such an event renders the system physically inconsistent within the asymptotic model, often causing numerical root-finding procedures for parameter estimation to fail. Overcoming these finite-size issues and sensitivities might require sophisticated bias correction techniques (e.g., based on expansions in $1/N$ or iterative simulation-based correction methods) or regularisation strategies, which represents an promising direction for future research aimed at developing a robust, practical estimator based on second moments.

An alternative, seemingly simpler approach, motivated by the structure of the Oracle NCPC estimator (Theorem 4.1), would be to construct a corrected estimator directly as a linear combination of the top two empirical eigenvectors, $\hat{v}_1$ and $\hat{v}_2$. This would involve estimating the ideal combination coefficients (related to $D_1, D_2$ in Theorem 4.1). A natural path would be to first estimate the underlying parameters $(\alpha, \beta)$ using only the empirical eigenvalues $(\hat{\lambda}_1, \hat{\lambda}_2)$ from the first moment matrix $A$. However, this approach is fundamentally precluded by the confounding dependency established in Lemma 4.1. As proven, it is mathematically impossible to uniquely determine both $\alpha$ and $\beta$ solely from $\hat{\lambda}_1$ and $\hat{\lambda}_2$ (or any function thereof, such as their gap). This barrier was precisely our motivation for investigating the second moment $\mathbb{E}[A^2]$ as an additional source of information.

In summary, while the simple linear combination of $\hat{v}_1$ and $\hat{v}_2$ fails due to the confounding issue identified in Lemma 4.1, the second-moment approach is theoretically sound but faces practical hurdles related to

sensitivity and finite-size effects. The theoretical framework presented in this section remains valuable: it proves that correction is theoretically possible (Oracle NCPC), identifies the necessary information source (second moment), and rigorously characterises the inherent difficulties (sensitivity analysis). Developing a robust practical estimator, potentially by refining the second-moment approach with appropriate corrections or regularisation, remains a key challenge for future work.

## 5 Experimental Validation

In this section, we present a comprehensive suite of numerical simulations designed to validate the theoretical claims developed in the preceding sections. Our validation is structured as follows:

- First (§5.1), we conduct a series of direct validations of our idealised spiked model, providing rigorous, quantitative verification for the key theoretical predictions regarding phase transitions, eigenspace structure, and fluctuation statistics presented in Section 3.

- Second (§5.2), we experimentally validate the theoretical sensitivity analysis of the second-moment correction framework developed in Section 4. We numerically probe the inverse map $F^{-1} : (\bar{\lambda}_1, \bar{\mu}_1) \mapsto (\alpha, \beta)$ by introducing noise to the theoretical eigenvalues and visualising the resulting parameter estimation errors and biases, confirming the predictions from the Jacobian condition number analysis.

- Third (§5.3), to demonstrate that the core insights from our linear model extend beyond that idealised setting, we test their applicability in an illustrative, non-linear simulation involving spectral clustering in a simulated environmental context.

Unless otherwise specified, the main results are generated using a matrix size of $N = 1000$, with 50 Monte Carlo trials per data point (unless higher precision is needed, as specified), and noise parameters of $c_{\text{strong}} = 10$ and $c_{\text{weak}} = 1$. This high 10:1 anisotropy ratio ($\gamma = 10$) was chosen deliberately to create a strong and unambiguous experimental setting, allowing for a clear validation of the theoretical phenomena of interest, such as the "camouflage effect" and the predicted sensitivities. Code to reproduce all results is available[1]

### 5.1 Direct Validation of the Theoretical Model

Our first goal is to provide numerical proof for our theoretical framework using simulations that exactly match the model defined in §2. The most important test is to validate the central predictions for the phase transition and the reorganisation of the eigenspace. The results, shown in Figure 3, provide a complete picture of the system's behaviour. Panel (a) confirms the existence of the "noise hijacking" effect, where the alignment for the aligned case ($\alpha = 1$) is high even for weak signals, and the subsequent crossover phenomenon. Panel (b) reveals the underlying mechanism: an "eigenvalue crossover" (an identity swap) between the rising signal eigenvalue and the stable noise eigenvalue in the orthogonal case ($\alpha = 0$). Panel (c) provides direct visual proof for Theorem 3.5, showing that the second eigenvector successfully capturing the orthogonal noise direction after this crossover.

Next, we validate the information-theoretic threshold from Theorem 3.4. To do this, we must design a practical, *blind* spectral detector that requires no prior knowledge of the signal's structure. A naive detector based only on the largest eigenvalue, $\hat{\lambda}_1$, can fail when the noise spike is dominant. We therefore employ a more robust statistic, the "Excess Spectral Energy", defined as $T = (\hat{\lambda}_1 - \tau) + (\hat{\lambda}_2 - \tau)$, which measures the total energy of the top two eigenvalues above the theoretical noise bulk edge. The performance of this detector, measured by the Area Under the Curve (AUC), is shown in Figure 4. The results reveal a deep insight into the nature of detection: for the orthogonal case ($\alpha = 0$), performance exhibits a sharp phase transition from random guessing (AUC=0.5) to rising detection accuracy at the theoretical threshold $\beta_c(0) = 1.0$, validating the limit for *strong detection*. In contrast, for the aligned ($\alpha = 1$) and intermediate ($\alpha = 0.5$) cases, the detector achieves better-than-random performance even for $\beta < \beta_c$. This is because the signal continuously amplifies the pre-existing noise spike, a phenomenon our sensitive test statistic correctly

---

[1]Code URL: https://anonymous.4open.science/r/tmlr_ept

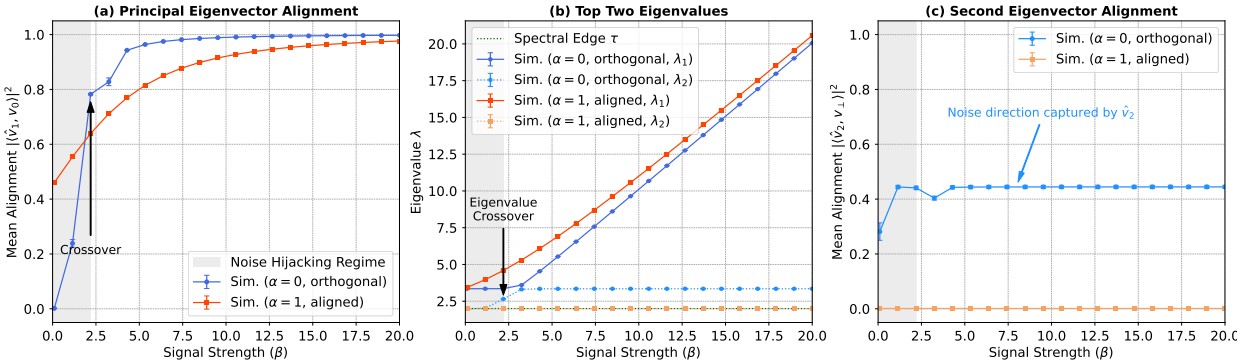

Figure 3: Validation of eigenvector phase transitions with finite-size Monte Carlo simulations ($N = 1000$, $c_{\text{strong}} = 10$, $c_{\text{weak}} = 1$, 50 trials per signal strength $\beta$ for mean and error bar estimation). **(a) Principal Eigenvector Alignment:** This panel shows that for low signal strength, the alignment for the aligned case ($\alpha = 1$, red) is higher due to a "noise hijacking" effect. The orthogonal case ($\alpha = 0$, blue) undergoes its sharp phase transition at $\beta \approx 1$ and eventually crosses over the aligned case and approaches its theoretical limit of perfect signal detection. **(b) Top Two Eigenvalues:** This panel reveals that for the aligned case ($\alpha = 1$), the signal and noise merge into a single dominant eigenvalue ($\lambda_1$, solid red), which grows continuously with $\beta$. The second eigenvalue ($\lambda_2$, dashed red) remains flat at the spectral edge, $\tau = 2c_{\text{weak}}$. For the orthogonal case ($\alpha = 0$), the system supports two distinct outlier eigenvalues. Initially, the largest eigenvalue ($\lambda_1$, solid blue) corresponds to the noise spike, while the second eigenvalue ($\lambda_2$, dashed blue) corresponds to the rising signal. After a point of eigenspace reorganisation around $\beta \approx 2.5$, the signal becomes dominant, taking over the role of $\lambda_1$, while the noise settles into the role of $\lambda_2$. The gap between the two eigenvalues narrows near this reorganisation point before widening again. **(c) Second Eigenvector Alignment:** This panel shows the consequence of the eigenvalue behaviour. For the aligned case, the alignment of $\hat{v}_2$ is near zero, as there is no distinct second spike to capture. For the orthogonal case, the alignment of $\hat{v}_2$ with the noise direction rises during the unstable reorganisation phase and then stabilises at a high value once the signal has cleanly separated and taken over the principal eigenvector.

identifies, thus demonstrating the presence of a *weak detection* regime. This provides an empirical picture of the theoretical limits and validates the crucial distinction between strong and weak detection in anisotropic environments.

To rigorously validate our asymptotic theory, it is crucial to demonstrate that the finite-size effects observed in the previous figure diminish as the system size $N$ increases. Figure 5 provides this proof. Panel (a) shows the principal eigenvector alignment from the simulation systematically converging towards the theoretical limit predicted by Theorem 3.3 as $N$ grows. Panels (b) and (c) confirm that the more subtle features of the model—the second eigenvector alignment and the locations of the outlier eigenvalues—are also robust asymptotic properties that converge to their predicted states, solidifying the validity of our entire theoretical framework.

The final validation of our idealised model tests the predictions for the statistical fluctuations of the entire eigenspace, providing empirical proof for the Central Limit Theorems governing both the first and second eigenvectors. The results for the principal eigenvector's alignment with the signal ($v_0$), shown in Figure 6 and Table 2, confirm the predictions of our first CLT (Theorem 3.6). The histograms and Q-Q plots provide clear visual evidence for the Gaussian nature of the fluctuations, while the table's quantitative analysis reveals the deep physical insight that the variance is highest for the intermediate case ($\alpha = 0.5$), as it operates closest to its critical tipping point where system instability is maximised.

Similarly, we validate the new CLT for the second eigenvector's alignment with the noise direction ($v_C$), presented in Theorem 3.7. The results, shown in Figure 7, provide equally strong visual and quantitative evidence for the Gaussian nature of these fluctuations in the challenging intermediate-alignment regime. Together, these validations provide a complete picture of the system's stability. They confirm that not

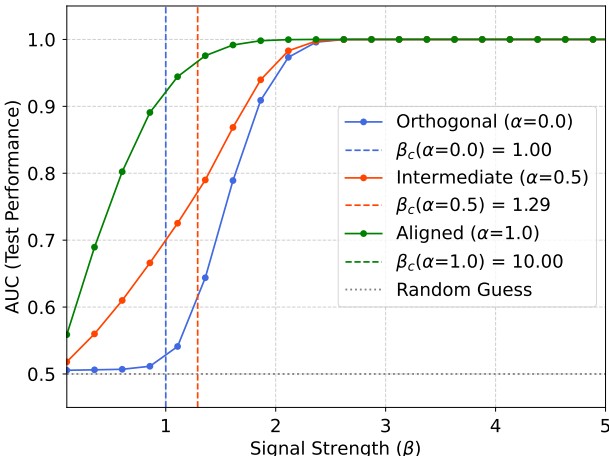

Figure 4: Validation of the information-theoretic detection threshold (Theorem 3.4) using a blind spectral detector ($N = 1000$, $c_{\text{strong}} = 10$, $c_{\text{weak}} = 1$, 100 trials per hypothesis). The figure plots the performance (AUC) of the "Excess Energy" spectral statistic against signal strength $\beta$. **Strong vs. Weak Detection:** The results reveal the rich structure of the detection problem. For the orthogonal case ($\alpha = 0$), the detector's performance shows a sharp phase transition from random guessing (AUC=0.5) to rising detection accuracy at the theoretical threshold $\beta_c(0) = 1.0$, validating the limit for *strong detection*. For the aligned ($\alpha = 1$) and intermediate ($\alpha = 0.5$) cases, the detector achieves better-than-random performance even for $\beta < \beta_c$, because the signal continuously amplifies the existing noise spike. This demonstrates the presence of a *weak detection* regime, providing an empirical picture of the theoretical limits.

Table 2: Measured statistics of the scaled alignment fluctuations from the CLT experiment ($N = 1000$, $c_{\text{strong}} = 10$, $c_{\text{weak}} = 1$, $10{,}000$ trials). The table provides a quantitative validation of the Central Limit Theorem. **Mean:** In all scenarios, the mean of the centered fluctuations is correctly measured to be zero, as expected by construction. **Variance:** The variance quantifies the instability of the alignment measurement and reveals a deep physical insight. The variance is not a simple monotonic function of the alignment $\alpha$. The intermediate case ($\alpha = 0.5$) exhibits a much larger variance (0.3134) than the extreme cases. This is because it is operating closer to its critical tipping point than the other two scenarios. For the intermediate case, the signal strength is $\beta = 5.0$ while its threshold is $\beta_c \approx 1.29$. For the orthogonal case, $\beta = 5.0$ is comparatively further from its threshold of $\beta_c = 1.0$. For the aligned case, $\beta = 15.0$ is also far from its threshold of $\beta_c = 10.0$. This result provides strong numerical evidence for the theoretical prediction that system instability is significantly higher near a critical phase transition.

| Scenario | Measured Mean | Measured Variance |
|---|---|---|
| Orthogonal ($\alpha = 0.0$) | 0.0000 | 0.0078 |
| Intermediate ($\alpha = 0.5$) | 0.0000 | **0.3134** |
| Aligned ($\alpha = 1.0$) | 0.0000 | 0.0071 |

only is the primary signal eigenvector's behaviour well-described by asymptotic theory, but so too is the "phantom" eigenvector that aligns with the dominant environmental gradient, establishing it as a stable and predictable feature of the spectrum.

For completeness, we also validate the foundational assumption of our model: the structure of the anisotropic noise spectrum. Figure 8 confirms the predictions of Theorem 3.1. The figure illustrates the emergence of the outlier "noise spike" as anisotropy increases, and the zoomed-in Panel (d) provides a high-precision validation that the continuous bulk of the spectrum terminates exactly at the theoretical edge, $\tau = 2c_{\text{weak}}$.

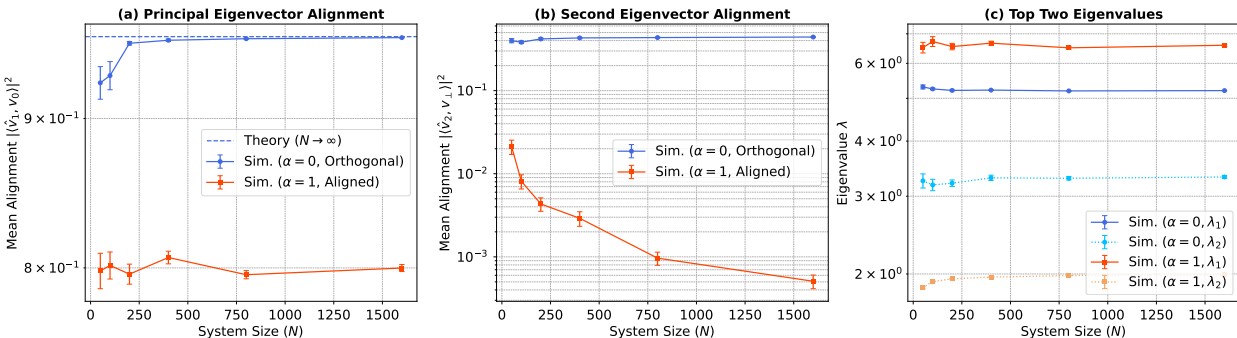

Figure 5: Validation of the asymptotic nature of eigenvector phase transitions and eigenspace reorganisation, showing the convergence of key observables (plotted in log scale) as the system size $N$ increases ($N = [50, 100, 200, 400, 800, 1600]$, $c_{\text{strong}} = 10$, $c_{\text{weak}} = 1$, 50 trials per $N$ for mean and error bar estimation). Both scenarios are run at a fixed signal strength of $\beta = 5.0$. **(a) Principal Eigenvector Alignment:** For the orthogonal case ($\alpha = 0$, solid blue line), the system is super-critical, and the alignment correctly converges to its theoretical limit (dashed blue line). For the aligned case ($\alpha = 1$), the system is sub-critical; the observed high alignment is a robust "noise hijacking" effect, where the eigenvector aligns with the dominant noise spike. **(b) Second Eigenvector Alignment:** For the orthogonal case, the alignment of $\hat{v}_2$ with the noise direction converges to a stable, non-zero value, confirming the asymptotic nature of the eigenspace reorganisation. For the aligned case, the alignment decays roughly as $\sim 1/N$, the expected behaviour for a random bulk eigenvector, confirming the merging of the signal and noise spikes. **(c) Top Two Eigenvalues:** The locations of all outlier eigenvalues stabilise rapidly as $N$ increases, demonstrating their convergence to the predicted deterministic limits.

## 5.2 Validation of the Second-Moment Analysis and its Sensitivity

The preceding subsection validated the core phenomena of phase transitions and eigenspace reorganisation described in Section 3. We now turn to the validation of the theoretical concepts presented in Section 4. The motivation for that section was the need for a second source of information, as codified in Lemma 4.1, which states that using only the top two eigenvalues of $A$ leads to a confounded inverse problem. Figure 9 provides direct empirical support for this lemma. Through Monte Carlo simulations, it visualises the mapping from the parameters $(\alpha, \beta)$ to the mean observed eigenvalues $(\mathbb{E}[\hat{\lambda}_1], \mathbb{E}[\hat{\lambda}_2])$. As shown in panel (b) of the figure, observables like the eigenvalue gap are a function of both $\alpha$ and $\beta$, confirming the theoretical confounding and motivating the need for additional information, such as that from the second moment.

The theoretical analysis in Section 4 predicts that the theoretical inverse map $F^{-1} : (\bar{\lambda}_1, \bar{\mu}_1) \mapsto (\alpha, \beta)$ is itself inherently sensitive in specific regimes. The Jacobian condition number, plotted in Figure 10, quantifies this sensitivity, predicting that the inverse problem is most ill-conditioned in the top-left of the parameter space, corresponding to weak, highly aligned ("camouflaged") signals. To validate this prediction, we conduct a computational experiment designed to probe the inverse map's robustness. We first calculate the exact, theoretical eigenvalues $(\bar{\lambda}_1, \bar{\mu}_1)$ for a dense grid of true parameters $(\alpha_{true}, \beta_{true})$. We then simulate the effect of finite-$N$ fluctuations by adding a small amount of Gaussian noise to these theoretical values. For each of these noisy eigenvalue pairs, we run our robust inverse solver to obtain parameter estimates $(\hat{\alpha}, \hat{\beta})$. By averaging over many noise trials ($M = 50$) for each grid point, we can measure both the magnitude of the estimation error and the direction of any systematic bias.

As shown in Figure 11, the heatmaps and overlaid bias flow vectors provide a confirmation of our analysis. The regions of highest estimation error for both $\alpha$ (panel a) and $\beta$ (panel b) are concentrated precisely in the top-left corner of the parameter space. This region of high empirical error directly corresponds to the region of high condition number predicted by the Jacobian analysis in Figure 10. Furthermore, the bias flow visualised by the quiver plot reveals a systematic structure to the errors. In the ill-conditioned top-left region, the bias vectors consistently point towards the bottom-right, indicating that the inverse solver systematically misinterprets noisy data from a weak, camouflaged signal as originating from a stronger, less-aligned signal.

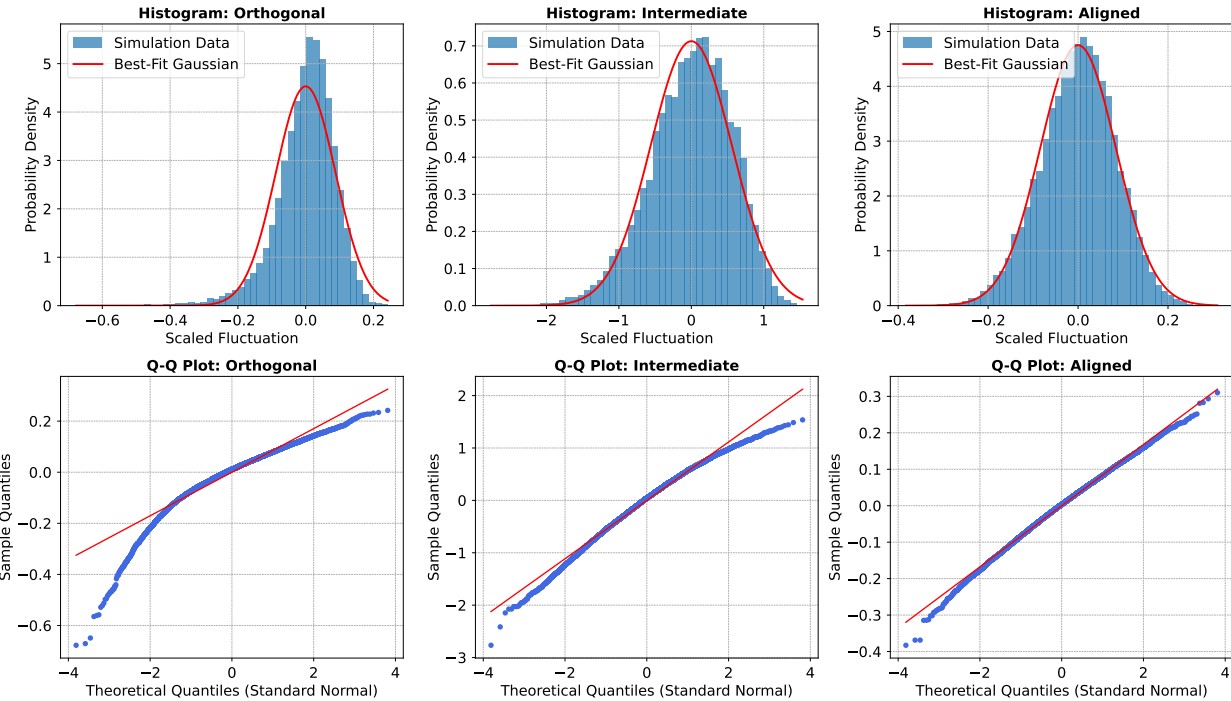

Figure 6: Validation of the Central Limit Theorem for alignment fluctuations at system size $N = 1000$ ($c_{\text{strong}} = 10$, $c_{\text{weak}} = 1$, $10,000$ Monte Carlo trials simulated). The top row shows histograms of the scaled fluctuations overlaid with a best-fit Gaussian, while the bottom row provides a more rigorous validation with Quantile-Quantile (Q-Q) plots. **Gaussianity:** In all three scenarios, the histograms closely match the Gaussian shape, and the Q-Q plots are nearly linear, providing strong evidence that the fluctuations are asymptotically Gaussian as predicted by Theorem 3.6. **Convergence Rate:** The quality of the Gaussian approximation reveals a deep and subtle insight into the model's physics. The convergence to the CLT is fastest for the aligned case ($\alpha = 1$), where the Q-Q plot is almost perfectly linear. This is due to the simpler physical system, where the signal and noise merge into a single spike. The convergence is slowest for the orthogonal case ($\alpha = 0$), where the Q-Q plot shows deviations in the tails. This reflects the more complex three-way interaction between the distinct signal spike, noise spike, and the random bulk, which generates stronger non-Gaussian finite-size effects that persist at $N = 1000$. The intermediate case ($\alpha = 0.5$) exhibits an intermediate convergence speed, consistent with this interpretation.

In contrast, the shorter, more randomly oriented arrows in the low-error (bottom-right) region show that the estimator is stable and accurate when the signal is strong and well-separated from the noise.

## 5.3 Application to Spectral Clustering in an Environmental Model

To bridge the gap between our idealised theory and a practical application, we now test our insights in a more realistic, non-linear setting. We simulate a 2D environmental model where two communities of points are generated, and we apply spectral clustering using an Radial Basis Function (RBF) kernel with an adaptive 'gamma' parameter to recover the latent structure.

Figure 12 provides a qualitative visualisation of the clustering performance. It clearly demonstrates the "camouflage effect" in a practical scenario: in the top row (aligned case, $\alpha = 1$), the algorithm fails to find the correct clusters even as signal strength increases, as all tested $\beta$ values are below the high theoretical threshold of $\beta_c = 10.0$. In contrast, the bottom row (orthogonal case, $\alpha = 0$) shows a clear phase transition from failure to success as $\beta$ crosses its low theoretical threshold of $\beta_c = 1.0$.

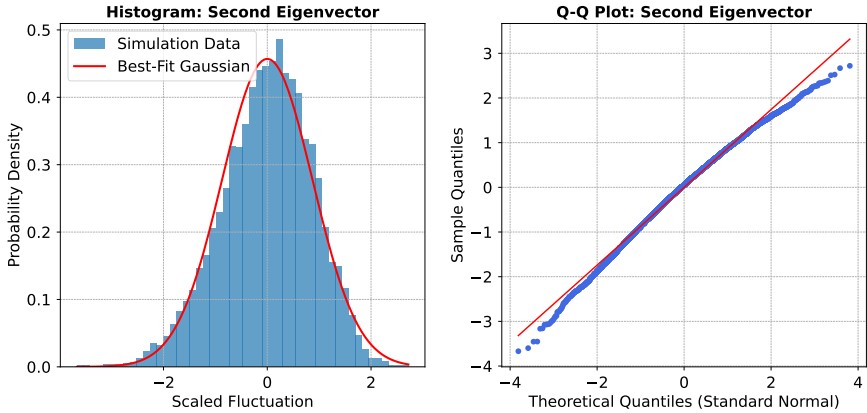

Figure 7: Validation of the Central Limit Theorem for the second eigenvector's alignment with the noise direction ($|\langle \hat{v}_2, v_C \rangle|^2$) at system size $N = 1000$ ($c_{\text{strong}} = 10$, $c_{\text{weak}} = 1$, $\alpha = 0.5$, $\beta = 5.0$, $10{,}000$ Monte Carlo trials). The left panel shows a histogram of the centered fluctuations overlaid with a best-fit Gaussian, while the right panel provides a more rigorous validation with a Quantile-Quantile (Q-Q) plot. **Gaussianity:** The histogram closely matches the Gaussian shape, and the Q-Q plot is nearly linear, providing strong evidence that the fluctuations of the second eigenvector's alignment are asymptotically Gaussian as predicted by Theorem 3.7. **Physical Interpretation:** This result provides a rigorous measure of the stability of the "phantom" eigenvector that emerges from the anisotropic noise. The Gaussian nature of the fluctuations confirms that even though $\hat{v}_2$ does not represent the primary signal, its alignment with the dominant environmental gradient is a stable and predictable feature of the system in the super-critical regime. The small deviations in the tails of the Q-Q plot are expected finite-size effects, indicating that even at $N = 1000$, the system has not fully reached its asymptotic limit, a phenomenon consistent with the complex interactions within the eigenspace.

Table 3: Comparison of the theoretical critical threshold from our idealised linear model with the empirically measured threshold from the non-linear spectral clustering experiment ($N = 1000$, $c_{\text{strong}} = 10$, $c_{\text{weak}} = 1$, 50 trials per signal strength to estimate mean). The measured threshold is defined as the signal strength $\beta$ at which the mean Adjusted Rand Index (ARI) first exceeds 0.95. **Agreement:** For the orthogonal and intermediate cases, the measured thresholds are quite close to the theoretical predictions, demonstrating the predictive power of our theory even in a more complex, non-linear setting. **Discrepancy:** For the aligned case, the measured threshold is significantly lower than the theoretical prediction. This is a direct consequence of the "noise hijacking" effect; the spectral clustering algorithm achieves a high ARI score by correctly clustering the dominant pattern created by the strong anisotropic noise itself, long before the true signal is theoretically detectable. This highlights a crucial practical implication: strong, aligned environmental gradients can create detectable structures that may be misinterpreted as a true signal.

| Scenario | Theoretical $\beta_c$ | Measured $\beta_c$ (ARI > 0.95) |
|---|---|---|
| Orthogonal ($\alpha = 0.0$) | 1.00 | 1.10 |
| Intermediate ($\alpha = 0.5$) | 1.29 | 1.60 |
| Aligned ($\alpha = 1.0$) | **10.00** | **2.50** |

The quantitative performance of the clustering is shown in Figure 13, which plots the Adjusted Rand Index (ARI) against signal strength. This figure provides a rigorous, quantitative proof of the "camouflage effect". The performance curves for the three alignment scenarios ($\alpha = 0, 0.5, 1$) exhibit sharp phase transitions in the exact order predicted by our theory, with the aligned case requiring a much stronger signal to achieve successful clustering.

Finally, we connect the performance of this complex, non-linear application directly back to our simple, idealised linear theory. Table 3 compares the theoretical critical threshold, $\beta_c$, with the empirically measured

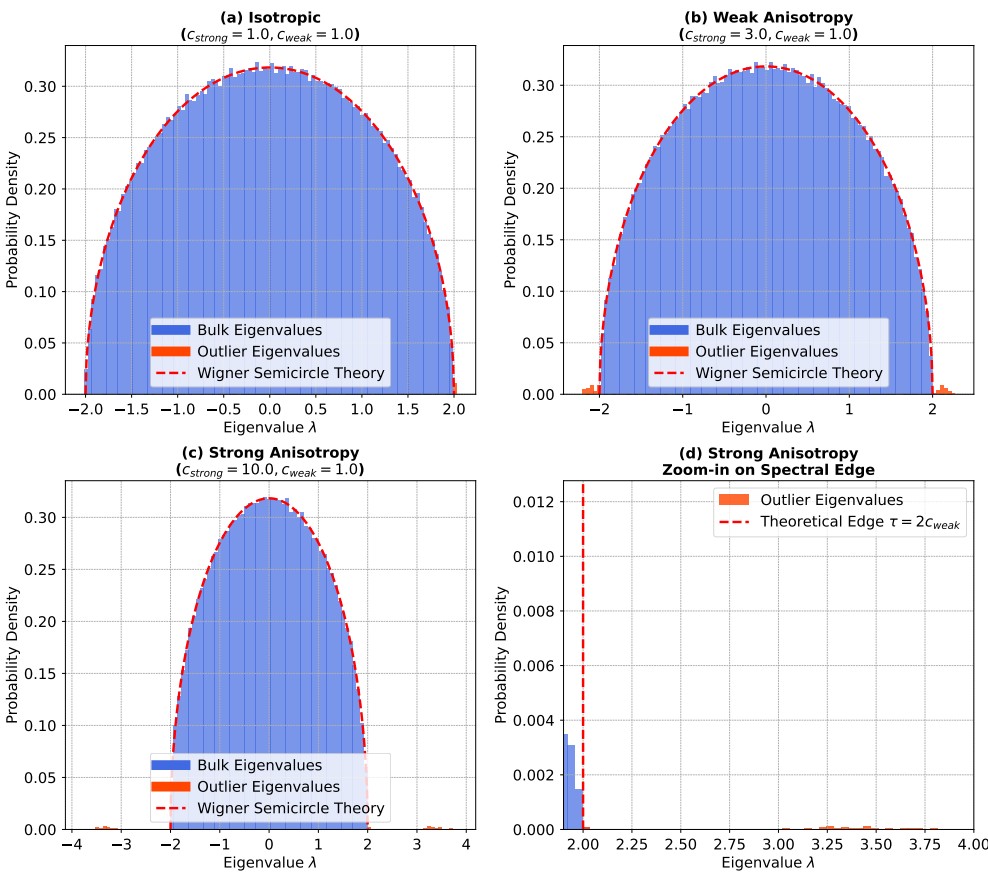

Figure 8: Validation of the anisotropic noise spectrum (Theorem 3.1) for $N = 1000$. The figure shows the empirical eigenvalue distribution for increasing levels of anisotropy. **(a)** In the isotropic case ($c_{\text{strong}} = c_{\text{weak}}$), the spectrum correctly matches the theoretical Wigner semicircle distribution. **(b)** With weak anisotropy, small but distinct outlier eigenvalues (the "noise spike", red) separate from the main bulk. **(c)** With strong anisotropy, the noise spike produces well-separated outliers, clearly distinct from the continuous bulk. **(d)** A zoomed-in view of the spectral edge for the strong anisotropy case provides a high-precision validation, showing that the continuous part of the spectrum (the bulk) terminates exactly at the theoretical prediction of $\tau = 2c_{\text{weak}}$ (red dashed line).

threshold from the non-linear spectral clustering experiment. For the orthogonal and intermediate cases, there is a strong agreement between the theoretical prediction and the measured result. The table also highlights an important insight for the aligned case: the measured threshold is much lower than the theoretical one. This discrepancy is a direct result of the "noise hijacking" effect, where the algorithm achieves a high ARI score by correctly clustering the dominant pattern created by the strong anisotropic noise itself, long before the true signal is theoretically detectable. This demonstrates the powerful predictive utility of our theoretical framework in explaining the behaviour of real-world spectral methods.

## 6  Discussion and Conclusion

In this work, we have developed and validated a complete theory for principal eigenvector phase transitions in the presence of anisotropic noise. Our findings extend the classical understanding of spectral methods by incorporating the crucial and realistic element of a dominant environmental gradient. We have shown that the detectability of a latent signal is not merely a function of its strength, but is fundamentally governed by its geometric alignment with the surrounding noise structure.

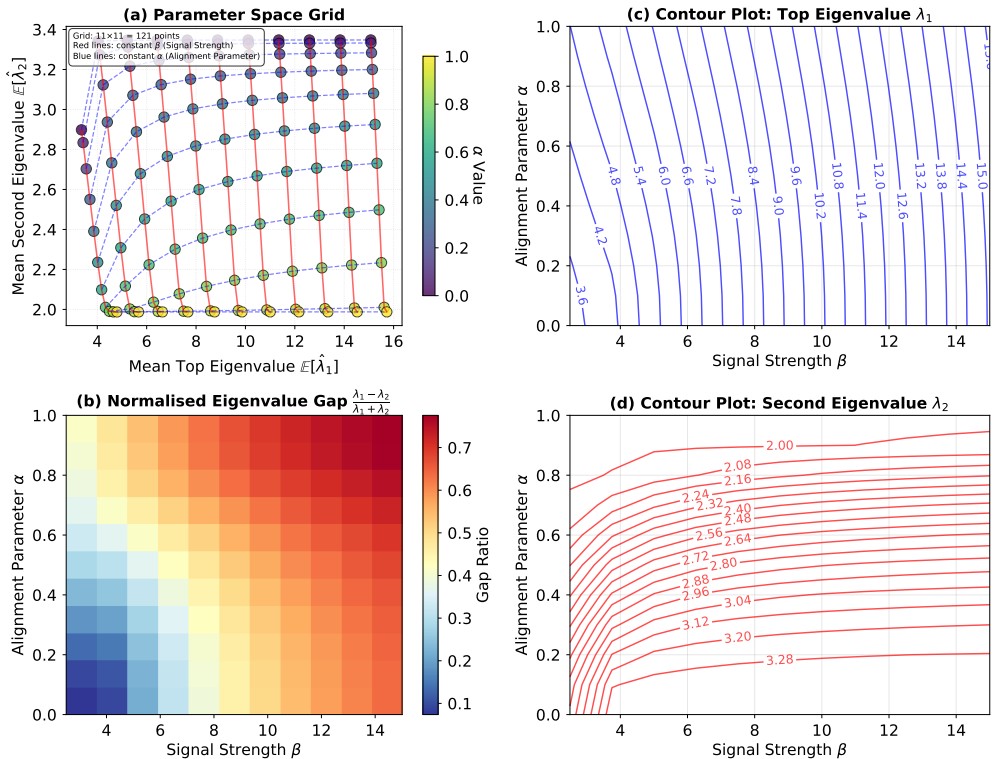

Figure 9: Empirical validation of the theoretical landscape of the inverse eigenvalue map (Lemma 4.1), based on Monte Carlo simulations at $N = 1000$. **(a) The Parameter Space Grid:** This panel provides a direct visual proof that the map from parameters $(\alpha, \beta)$ to mean observed eigenvalues $(\mathbb{E}[\hat{\lambda}_1], \mathbb{E}[\hat{\lambda}_2])$ is a local diffeomorphism, guaranteeing a unique inverse function. The dense packing of the constant-$\alpha$ lines near the boundaries visually confirms that the derivative of the inverse map is large at the edges. **(b) The Normalised Gap as an Imperfect Proxy:** This heatmap reveals a crucial subtlety: while the base heuristic $x = (\lambda_1 - \lambda_2)/(\lambda_1 + \lambda_2)$ is strongly and monotonically correlated with $\alpha$ (colour changes vertically), it also has a strong confounding dependency on the unknown signal strength $\beta$ (colour also changes horizontally). **(c) The Top Eigenvalue:** The near-vertical contour lines confirm that the top eigenvalue, $\lambda_1$, is primarily a function of signal strength $\beta$. **(d) The Second Eigenvalue:** The contours of $\lambda_2$ reveal a phase transition. For weak signals (low $\beta$), the contours are steep, indicating that $\lambda_2$ is a complex function of both $\alpha$ and $\beta$ due to strong spike interaction. Above a critical $\beta$, the signal spike decouples, and the contours abruptly flatten, showing that $\lambda_2$ settles into a final position determined strongly by $\alpha$. This transition is visibly sharper for higher values of $\alpha$.

## 6.1 Implications of Key Findings for Scientific Discovery

Our theoretical and experimental results reveal several deep physical phenomena with direct practical implications. The central finding is the "camouflage effect" (Theorem 3.2), which proves that a latent structure is fundamentally harder to detect when it aligns with a strong environmental gradient, such as a river valley or prevailing wind. This has important consequences for experimental design and data interpretation in the environmental sciences, suggesting that the orientation of a study area relative to its dominant gradients can significantly impact the statistical power to detect underlying patterns.

Our analysis also reveals a crucial distinction between the theoretical threshold for signal detection and the practical performance of methods like spectral clustering. We show that strong environmental gradients can themselves form coherent structures that are easily clustered, potentially misleading a scientist into believing that they have found a true latent signal when none is statistically detectable.

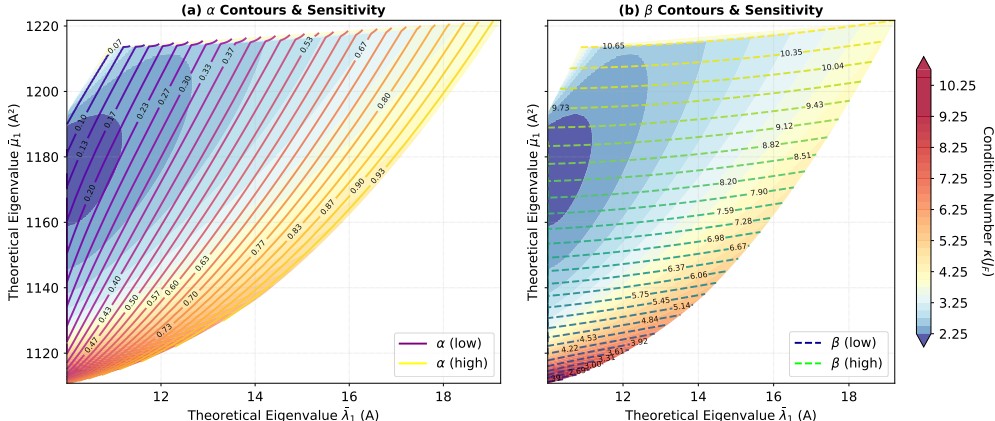

Figure 10: Theoretical landscape and sensitivity of parameter estimation in the asymptotic eigenvalue space $(\bar{\lambda}_1, \bar{\mu}_1)$ for fixed noise parameters ($c_{\text{strong}} = 10, c_w = 1, N = 1000$). Both panels show the Jacobian condition number, $\kappa(J_F)$, as a heatmap background (brighter colours indicate higher sensitivity/ill-conditioning). **(a) Overlaid with contours of constant alignment $\alpha$ (solid lines)**. The heatmap reveals regions of high sensitivity near the lower-left boundary (corresponding to parameters near the critical threshold $\beta_c$) and towards the right (high eigenvalues). The alpha contours show how the alignment parameter maps onto the eigenvalue space. **(b) Overlaid with contours of constant signal strength $\beta$ (dashed lines)**. The beta contours complete the coordinate system, visualising the invertibility of the map $F : (\alpha, \beta) \mapsto (\bar{\lambda}_1, \bar{\mu}_1)$ where contours intersect. Where contours in (a) and (b) become nearly parallel (coinciding with bright heatmap regions), the inverse problem is ill-conditioned: small errors or fluctuations in measured eigenvalues $(\hat{\lambda}_1, \hat{\mu}_1)$ will be significantly amplified, leading to large uncertainties in estimated parameters $(\hat{\alpha}, \hat{\beta})$.

Furthermore, our discovery of the eigenspace reorganisation phenomenon (Theorem 3.5) suggests a novel diagnostic application for spectral methods. In the detectable phase, the principal eigenvector reveals the latent signal, while the second eigenvector simultaneously diagnoses the primary axis of environmental noise. This offers the potential for a "2-for-1" analytical tool, capable of both pattern detection and the characterisation of the system's dominant interference.

Beyond diagnostics, this work also introduces a theoretical path to correction based on the second-moment analysis developed in Section 4. While our sensitivity analysis (§4.5) and validation (§5.2) show that this theoretical estimator is ill-conditioned in the most difficult "camouflaged" regimes, its existence proves that the signal information is not destroyed, but merely confounded within a 2D subspace. This opens a promising new research direction into developing robust practical estimators that can jointly identify and correct for anisotropic noise, moving beyond simple detection to full signal recovery.

Finally, our analysis of the alignment fluctuations (Theorem 3.6) provides a rigorous understanding of the uncertainty inherent in these methods. The finding that the variance of the alignment is maximised at the critical threshold is a profound result. This "critical fluctuation" is conceptually analogous to the "critical slowing down" phenomenon observed in dynamic systems, which can serve as an early-warning signal for tipping points. By analogy, one might hypothesise that as a real-world system approaches a critical transition, repeated spectral analyses could reveal increasingly "jittery" or unstable eigenvectors. Investigating whether this statistical instability could translate into a practical, dynamic early-warning signal is a promising direction for future research (Scheffer et al., 2009).

## 6.2 Connection to Broader Literature

Our work sits at the intersection of random matrix theory (RMT), machine learning, and environmental science, contributing to and drawing deep connections between these diverse fields.

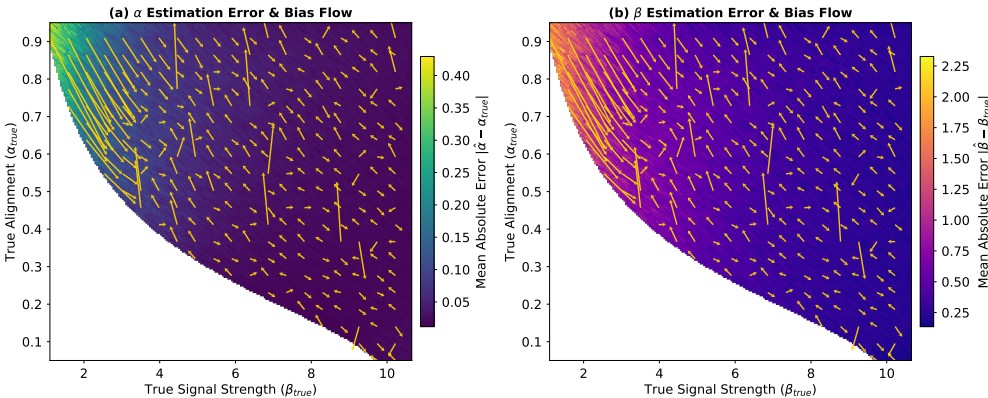

Figure 11: Theoretical inverse map sensitivity and visualisation of estimation bias. The experiment (parameters: $c_{\text{strong}} = 10$, $c_{\text{weak}} = 1$, $N = 1000$) calculates theoretical eigenvalues $(\bar{\lambda}_1, \bar{\mu}_1)$ for a grid of true parameters $(\alpha_{true}, \beta_{true})$, adds small Gaussian noise ($\sigma_{rel} = 0.5\%$) to simulate finite-N fluctuations, and runs the inverse estimator multiple times ($M = 50$) for each point. **(a) Alignment Estimation Error & Bias Flow:** The heatmap shows the mean absolute error $|\hat{\alpha} - \alpha_{true}|$ (averaged over noise trials) on a linear scale. The overlaid quiver plot (yellow arrows) shows the mean bias vector $(\mathbb{E}[\hat{\beta}] - \beta_{true}, \mathbb{E}[\hat{\alpha}] - \alpha_{true})$ originating from the true parameters. Errors are largest in the top-left region (low $\beta_{true}$, high $\alpha_{true}$), corresponding to weak, highly aligned signals. The bias flow is predominantly directed towards the bottom-right, indicating a systematic misinterpretation towards stronger, less aligned signals. **(b) Signal Strength Estimation Error & Bias Flow:** Similar to (a), but the heatmap shows the mean absolute error $|\hat{\beta} - \beta_{true}|$. The same bias vector field is overlaid. The error pattern mirrors that of (a), confirming that the top-left "weak & aligned" regime is the most challenging for estimating both parameters.

**Random Matrix Theory.** Our model can be viewed as a rank-two perturbation of an isotropic Wigner matrix, a class of models that has been studied in RMT (Capitaine et al., 2009). Our model is a non-trivial generalisation of the canonical spiked matrix framework, which has become a cornerstone of high-dimensional statistics. The foundational work on sample covariance matrices (Johnstone, 2001) and Wigner matrices (Baik et al., 2005) established the BBP phase transition, a sharp threshold for the detection of a low-rank signal in high-dimensional isotropic noise. This core result has been extended in numerous directions, including to signals of higher rank (Capitaine et al., 2009), to different noise ensembles such as Wishart matrices (Baik & Silverstein, 2006), and to understand the detailed statistical fluctuations of the outlier eigenvectors and their projections (Paul, 2007; Benaych-Georges & Nadakuditi, 2011; Knowles & Yin, 2017). While some work has considered deterministically deformed Wigner ensembles (Anderson & Zeitouni, 2006; Erdős et al., 2013) or sample covariance matrices with a general population covariance (Bloemendal et al., 2016), the specific question of how the geometric alignment between a signal spike and a noise spike affects the detection threshold has been less explored. Our work provides a precise, analytical answer to this question, offering a new class of solvable spiked models that more accurately reflect structured noise environments. This connects to broader themes in RMT, such as the universality of eigenvalue statistics (Erdős & Yau, 2017; Tao & Vu, 2010) and the localisation of eigenvectors (Bourgade & Yau, 2017; Luh & O'Rourke, 2020), by demonstrating a clear, non-universal phenomenon where the geometry of the perturbation dictates the system's behaviour. Our work is distinct from standard rank-k analyses in the literature in its specific focus on the *geometric alignment* ($\alpha$) between a "signal" spike and a "noise covariance" spike. We explicitly derive the detection threshold *as a function of this alignment* (Theorem 3.2), which formalises the camouflage phenomenon. Furthermore, our contributions go beyond the standard spectral characterisation by (1) proving that this threshold is information-theoretically optimal (Theorem 3.4), (2) providing a full statistical (CLT) description of the entire 2D outlier eigenspace, including the noise-aligned vector (Theorem 3.7), and (3) proposing a theoretical correction framework based on second moments (§4), which is a non-standard approach to this problem.

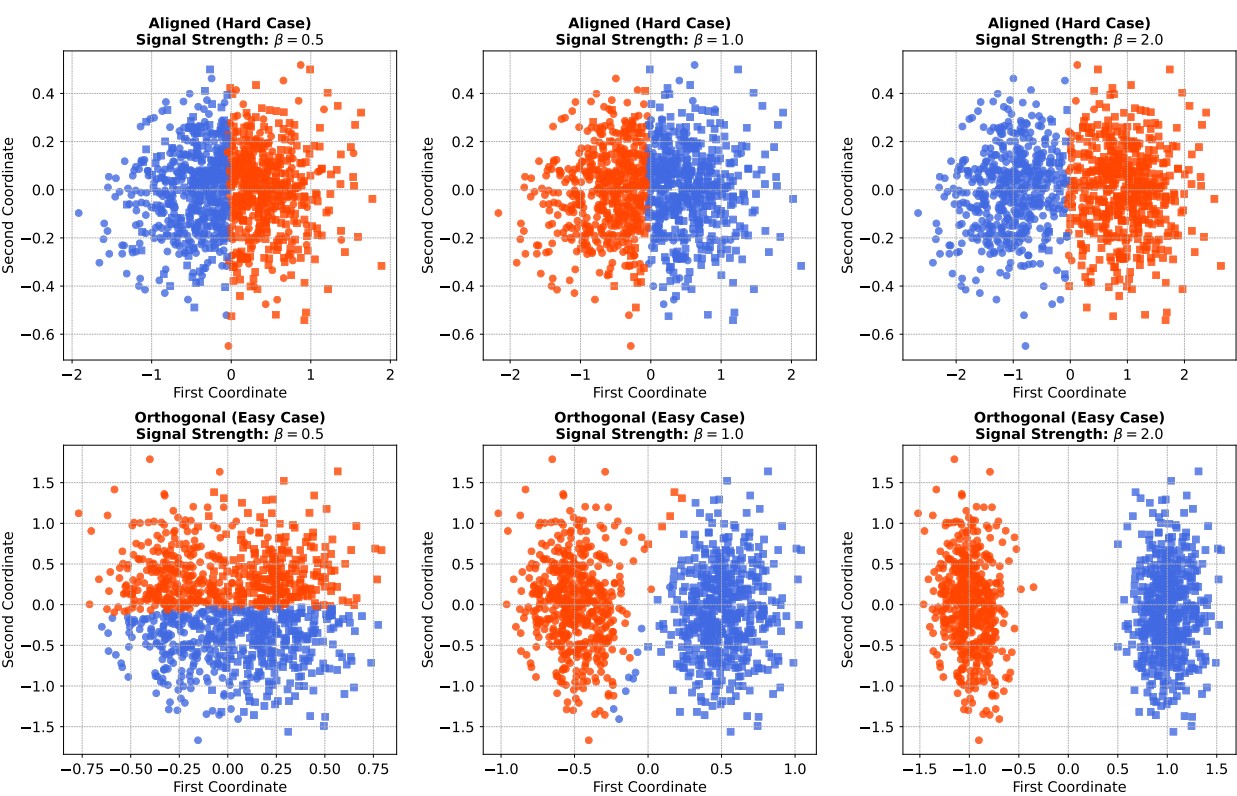

Figure 12: A visual demonstration of non-linear spectral clustering performance on the environmental model ($N = 1000$, $c_{\text{strong}} = 10$, $c_{\text{weak}} = 1$) across different signal strengths ($\beta$) and alignments ($\alpha$). In each panel, the true community is indicated by the marker shape (circle or square), while the discovered community is indicated by the colour (blue or red). **Top Row (Aligned Case, $\alpha = 1$):** The clustering fails clearly in low signal strength regimes and succeeds only partially at $\beta = 2.0$. Even as the signal strength increases, the algorithm is unable to find the correct horizontal separation because the signal is "camouflaged" by the strong, aligned environmental noise. All three $\beta$ values are below the theoretical detection threshold of $\beta_c = 10.0$ under the idealised linear model. **Bottom Row (Orthogonal Case, $\alpha = 0$):** This row clearly illustrates the phase transition. At $\beta = 0.5$ (sub-critical), the clustering fails. At $\beta = 1.0$ (the theoretical tipping point), the clustering is partially successful with only a minority of mistakes. By $\beta = 2.0$ (super-critical), the algorithm achieves a near-perfect separation. This provides strong visual evidence that the practical performance of spectral clustering is governed by the theoretical principles developed in this paper.

**Machine Learning.** Our findings provide a new theoretical lens for understanding the performance and failure modes of spectral methods. Spectral clustering, which is intimately connected to graph partitioning and the properties of the graph Laplacian (Chung, 1997; Fiedler, 1973), is a powerful tool for community detection (Fortunato, 2010; Newman, 2006). Foundational analyses have established its consistency under generative models like the stochastic block model (SBM) (Rohe et al., 2011; Qin & Rohe, 2013; Abbe, 2018), and its limits are tied to the detectability thresholds of the SBM itself (Decelle et al., 2011; Mossel et al., 2018). Our work moves beyond the standard signal-to-noise ratio arguments and provides a new, geometric mechanism for failure: a misalignment between the desired community structure and the dominant sources of variance in the data's feature space. This connects to a broader class of problems in manifold learning and dimensionality reduction, such as Principal Component Analysis (PCA) (Jolliffe, 2011), Isomap (Tenenbaum et al., 2000), and Locally Linear Embedding (Roweis & Saul, 2000), where the goal is to find a meaningful low-dimensional representation. Our theory suggests that the success of these methods may also depend on the anisotropic nature of the noise in the ambient high-dimensional space, a factor not typically considered in standard analyses (Belkin & Niyogi, 2003). Furthermore, our results on the second eigenvector's ability

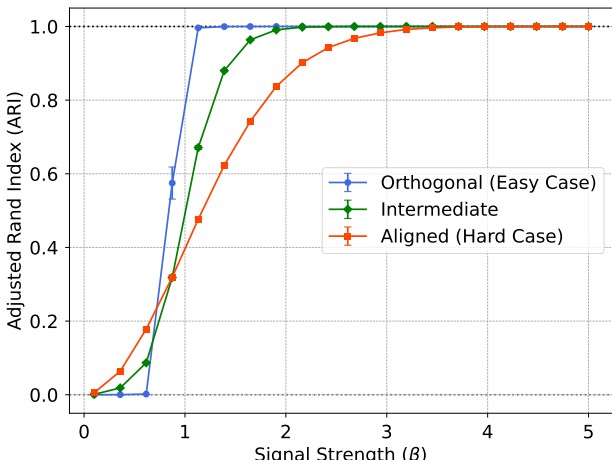

Figure 13: Quantitative performance of spectral clustering on the environmental model ($N = 1000$, $c_{\text{strong}} = 10$, $c_{\text{weak}} = 1$, 50 trials per signal strength to estimate mean and error bar), measured by the Adjusted Rand Index (ARI). The figure demonstrates a clear phase transition in clustering performance that is governed by the signal-noise alignment $\alpha$. For very low signal strength ($\beta < 0.5$), the intermediate ($\alpha = 0.5$) and aligned ($\alpha = 1.0$) cases show a non-zero ARI due to a "noise hijacking" effect, where the algorithm correctly clusters the pattern created by the strong anisotropic noise itself. The orthogonal case ($\alpha = 0$) undergoes a sharp phase transition at its theoretical threshold of $\beta_c = 1.0$, rapidly achieving perfect clustering (ARI=1). The intermediate case exhibits a delayed but still sharp transition, while the aligned case shows the slowest performance increase, confirming that a signal is practically harder to cluster when it is "camouflaged" by the dominant environmental gradient.

to capture the noise direction relate to methods for data denoising and component separation, such as Independent Component Analysis (ICA) (Hyvärinen & Oja, 2000) and Robust PCA (Candès et al., 2011).

**Environmental Science and Spatial Statistics.** Our model provides a formal mathematical framework for a wide range of well-established empirical phenomena. The concept of anisotropy is central to geostatistics, where the spatial correlation of a variable (e.g., mineral concentration or soil moisture) is known to depend on direction, a feature captured by anisotropic variogram models (Cressie, 2015; Goovaerts, 1997; Chiles & Delfiner, 2012). In landscape genetics, the "camouflage effect" directly models how landscape features that create anisotropic gene flow—such as rivers (Hughes et al., 2009; Allendorf, 1988) and highways (Epps et al., 2005)—can either obscure or reveal latent population structures (Manel et al., 2003; Spear et al., 2005). In ecology, our model of a signal emerging from a structured background is analogous to the problem of distinguishing habitat-driven species associations from patterns generated by dispersal limitation or metapopulation dynamics (Moilanen & Hanski, 1998; Hanski, 1999). The directional bias in our noise model is a key feature of many physical processes, including wind dispersal of seeds (Soons et al., 2004; Nathan et al., 2008; Bullock et al., 2016) and the transport of pollutants in air or water (Seinfeld & Pandis, 2016). Finally, in climate science, the use of PCA (often called Empirical Orthogonal Function analysis) to identify dominant modes of climate variability, such as the El Niño-Southern Oscillation (ENSO) (Bjerknes, 1969; Rasmusson & Carpenter, 1982) or the North Atlantic Oscillation (NAO) (Hurrell, 1995; Wanner et al., 2001), is a direct application of finding principal eigenvectors. Our theory provides a new framework for understanding how the stability and detectability of these climate modes might be affected by other, competing sources of large-scale environmental variance. While our work focuses on a symmetric Wigner model, we note that analogous problems in rectangular (sample covariance) models are also of great practical importance, particularly for high-dimensional data analysis (Su & Wu, 2025; Zhang & Mondelli, 2024). Extending our "camouflage" framework to that setting is a valuable direction for future research.

### 6.3 Limitations and Future Directions

While our model provides a complete and solvable framework, we acknowledge several simplifying assumptions that open clear avenues for future research.

- **Rank-One Signal:** Our model considers only a simple two-community structure. A natural next step is to extend the theory to higher-rank signal matrices to model the detection of multiple, competing communities, a problem of great interest in community ecology and network science (Capitaine et al., 2009).

- **Single Noise Spike:** Our noise model has only one dominant anisotropic direction. Future work could investigate the complex interactions that arise when there are multiple, competing anisotropic noise directions, such as modelling the confluence of two river systems or the intersection of different geological features.

- **Linear Additive Model:** Our core theory is based on a linear 'Signal + Noise' model. Our environmental simulation demonstrated that these insights are robust in a non-linear setting, but developing a rigorous theory for the phase transitions in random kernel matrices generated from anisotropic data remains a challenging but important open problem for the machine learning community (El Karoui, 2010).

- **Gaussian Noise and Universality:** Our proofs for the Optimality (Theorem 3.4) and Central Limit Theorems (Theorem 3.6, 3.7) rely on a Gaussian assumption for the baseline Wigner matrix. While the spectral edge (Theorem 3.1) and phase transition threshold (Theorem 3.2) are expected to be universal based on standard RMT results, proving the universality of these fluctuation and optimality results for non-Gaussian Wigner matrices is a non-trivial extension. An important direction would also be to consider heavy-tailed noise, such as Lévy matrices (Aggarwal et al., 2021), to model extreme events.

- **Validating the Diagnostic Application:** Our discovery that the second eigenvector captures the noise direction was validated in our idealised linear model. A crucial next step is to investigate whether this spectral "sorting" phenomenon holds in non-linear settings, such as for the eigenvectors of random kernel matrices, to develop a truly practical diagnostic tool for identifying structured latent noise in real-world data analysis.

- **Practicality of the Second-Moment Estimator:** As our sensitivity analysis in §4.5 and §5.2 demonstrates, the theoretical second-moment estimator is ill-conditioned precisely in the weak, camouflaged regime. Developing a regularised or otherwise robustified version of this estimator that is practical for finite-N, noisy data is a major open challenge.

### 6.4 Conclusion

This paper provides the first complete, analytical theory for principal eigenvector phase transitions under anisotropic noise. We have discovered and validated the "camouflage effect", which dictates that the detectability of a latent signal is fundamentally governed by its geometric alignment with the dominant environmental gradients. Furthermore, we have shown that in the detectable phase, the eigenspace reorganises itself, allowing for the simultaneous detection of both the signal and the primary axis of noise. These findings have profound and actionable implications for the application and interpretation of spectral methods in both machine learning and environmental sciences, providing a new framework for understanding how structure emerges from noise in complex, real-world systems.

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

# A  Detailed Proofs of Results in §3

This appendix provides detailed derivations for the main theoretical results presented in §3. Throughout these proofs, we utilise standard resolvent-based methods from random matrix theory. We use $\bar{X}$ to denote the deterministic $N \to \infty$ limit of an $N$-dependent quantity $X_N$. First-order expansions (e.g., $\delta\lambda_1 = \cdots + o_p(N^{-1/2})$) are to be understood as relations between the leading-order fluctuation terms, which can be made rigorous using standard CLT techniques for resolvent entries, as detailed in (Benaych-Georges & Nadakuditi, 2011; Paul, 2007). Here, $X_N = o_p(a_N)$ denotes convergence to zero in probability after scaling by $1/a_N$ (i.e., $X_N/a_N \xrightarrow{P} 0$ as $N \to \infty$).

The appendix is organised as follows:

- In Appendix A.1, we establish key technical lemmas used throughout the subsequent proofs.

- In Appendix A.2, we provide the detailed proof for Theorem 3.1, which characterises the spectral edge of the anisotropic noise matrix.

- In Appendix A.3, we present the proofs for Theorem 3.2 and Theorem 3.3, deriving the phase transition threshold and the asymptotic eigenvector alignment.

- In Appendix A.4, we prove Theorem 3.4, establishing that the spectral method is information-theoretically optimal.

- In Appendix A.5, we prove Theorem 3.5, which describes the alignment of the second eigenvector with the dominant noise direction.

- In Appendix A.6, we outline the derivation for Theorem 3.6 (the CLT for the principal eigenvector's alignment).

- Finally, in Appendix A.7, we outline the derivation for Theorem 3.7 (the CLT for the second eigenvector's alignment).

## A.1  Proof of Lemma A.1

The foundation of our analysis is the relationship between the eigenvalues of the full matrix $A$ and the properties of the noise matrix $W_{\text{aniso}}$. This relationship is established by analysing the resolvent of $A$. The following lemma provides the exact, non-asymptotic characteristic equation for any outlier eigenvalues created by the signal spike.

**Lemma A.1** (The Outlier Eigenvalue Equation). *Let $A = \beta v_0 v_0^T + W_{aniso}$. An outlier eigenvalue $\lambda$ of $A$ that is not an eigenvalue of $W_{aniso}$ must satisfy the equation:*

$$1 = \beta \langle v_0, (\lambda I - W_{aniso})^{-1} v_0 \rangle \tag{21}$$

*Proof.* The proof is a direct application of the Sherman-Morrison formula, which gives an explicit expression for the inverse of a matrix after a rank-one update (Golub & Van Loan, 2013).

Let $M$ be an invertible matrix and let $u, v$ be column vectors. The Sherman-Morrison formula states:

$$(M + uv^T)^{-1} = M^{-1} - \frac{M^{-1} u v^T M^{-1}}{1 + v^T M^{-1} u} \tag{22}$$

provided that the denominator $1 + v^T M^{-1} u \neq 0$.

We wish to analyse the resolvent of our matrix $A$, which is defined as $G(z) = (A - zI)^{-1}$ for a complex number $z$. The eigenvalues of $A$ are the points $z$ in the complex plane where this inverse does not exist, i.e., where the resolvent has poles.

We can write $A - zI$ as a rank-one update to the matrix $(W_{\text{aniso}} - zI)$:

$$A - zI = (W_{\text{aniso}} - zI) + \beta v_0 v_0^T \tag{23}$$

Let us identify the terms for the Sherman-Morrison formula:

- $M = W_{\text{aniso}} - zI$

- $u = \beta v_0$

- $v = v_0$

The Sherman-Morrison formula requires M to be invertible. Since the Lemma's premise is that $\lambda$ is not an eigenvalue of $W_{\text{aniso}}$, the matrix $M = W_{\text{aniso}} - \lambda I$ is indeed invertible, and the formula can be applied. We define the resolvent of the noise matrix as $G_{\text{noise}}(z) = (W_{\text{aniso}} - zI)^{-1}$. Applying the formula, we get the resolvent of the full matrix $A$:

$$G(z) = (A - zI)^{-1} = G_{\text{noise}}(z) - \frac{G_{\text{noise}}(z)(\beta v_0)v_0^T G_{\text{noise}}(z)}{1 + v_0^T G_{\text{noise}}(z)(\beta v_0)} \tag{24}$$

Simplifying the denominator gives:

$$G(z) = G_{\text{noise}}(z) - \frac{\beta G_{\text{noise}}(z)v_0 v_0^T G_{\text{noise}}(z)}{1 + \beta \langle v_0, G_{\text{noise}}(z)v_0 \rangle} \tag{25}$$

The poles of $G(z)$ that are not already poles of $G_{\text{noise}}(z)$ can only occur where the denominator of the second term is zero. An outlier eigenvalue $\lambda$ of $A$ that is not an eigenvalue of the noise matrix $W_{\text{aniso}}$ must therefore satisfy:

$$1 + \beta \langle v_0, G_{\text{noise}}(\lambda)v_0 \rangle = 0 \tag{26}$$

Substituting the definition of the noise resolvent, $G_{\text{noise}}(\lambda) = (W_{\text{aniso}} - \lambda I)^{-1}$, we get:

$$1 + \beta \langle v_0, (W_{\text{aniso}} - \lambda I)^{-1} v_0 \rangle = 0 \tag{27}$$

Rearranging this equation gives:

$$1 = -\beta \langle v_0, (W_{\text{aniso}} - \lambda I)^{-1} v_0 \rangle \tag{28}$$

To match the conventional form in the Lemma's statement, we use the identity $-(B)^{-1} = (-B)^{-1}$. Applying this, we absorb the negative sign into the inverse to yield the final result:

$$1 = \beta \langle v_0, (\lambda I - W_{\text{aniso}})^{-1} v_0 \rangle \tag{29}$$

This completes the proof. $\square$

## A.2  Proof of Theorem 3.1

This section provides the detailed proof for Theorem 3.1, which establishes the location of the spectral edge for the anisotropic noise matrix $W_{\text{aniso}}$.

**Theorem** (The Spectral Edge (Restatement of Theorem 3.1)). *In the limit $N \to \infty$, the continuous part of the eigenvalue spectrum of the anisotropic noise matrix $W_{aniso}$ is supported on the compact interval $[-\tau, \tau]$, where the spectral edge $\tau$ is given by:*

$$\tau = 2c_{weak} \tag{30}$$

*Proof.* The proof relies on the fundamental principle from random matrix theory that the continuous part of the spectrum of a large random matrix is stable under finite-rank perturbations.

**Step 1: Decompose the Covariance Structure.** We begin by decomposing the deterministic covariance structure matrix $C$ into a simple isotropic bulk and a finite-rank perturbation. Given our model specification for $C$ (one eigenvalue $c_{\text{strong}}$ with eigenvector $v_C$, and $N-1$ eigenvalues of $c_{\text{weak}}$), we can write its spectral decomposition as:

$$C = c_{\text{weak}}I + (c_{\text{strong}} - c_{\text{weak}})v_C v_C^T \tag{31}$$

Here, $c_{\text{weak}}I$ represents the isotropic "bulk" component, and the second term, $(c_{\text{strong}} - c_{\text{weak}})v_C v_C^T$, is a rank-one matrix representing the single anisotropic "spike".

**Step 2: Decompose the Noise Matrix into Bulk and Perturbation.** Our goal is to show that $W_{\text{aniso}}$ can be written as an isotropic bulk term plus a finite-rank perturbation. To find the square root of $C$, we apply the square root function to its eigenvalues. Given the spectral decomposition in the previous step, the eigenvalues of $C$ are $c_{\text{strong}}$ (with eigenvector $v_C$) and $c_{\text{weak}}$ (for all vectors orthogonal to $v_C$). Applying the square root function to these eigenvalues gives $\sqrt{c_{\text{strong}}}$ and $\sqrt{c_{\text{weak}}}$. We can therefore construct $C^{1/2}$ using the same eigenvectors:

$$C^{1/2} = \sqrt{c_{\text{weak}}}I + (\sqrt{c_{\text{strong}}} - \sqrt{c_{\text{weak}}})v_C v_C^T \tag{32}$$

This is the form used in the subsequent expansion of $W_{\text{aniso}}$.

Let us define the constant $d = \sqrt{c_{\text{strong}}} - \sqrt{c_{\text{weak}}}$ for brevity. Now we expand the definition of $W_{\text{aniso}}$:

$$W_{\text{aniso}} = C^{1/2}WC^{1/2} \tag{33}$$

$$= (\sqrt{c_{\text{weak}}}I + dv_C v_C^T)W(\sqrt{c_{\text{weak}}}I + dv_C v_C^T) \tag{34}$$

$$= (\sqrt{c_{\text{weak}}}I)W(\sqrt{c_{\text{weak}}}I) + (\sqrt{c_{\text{weak}}}I)W(dv_C v_C^T)$$
$$+ (dv_C v_C^T)W(\sqrt{c_{\text{weak}}}I) + (dv_C v_C^T)W(dv_C v_C^T) \tag{35}$$

$$= c_{\text{weak}}W + \underbrace{d\sqrt{c_{\text{weak}}}(Wv_C v_C^T + v_C v_C^T W) + d^2 v_C(v_C^T W v_C)v_C^T}_{\text{Perturbation Matrix } P} \tag{36}$$

We have now decomposed $W_{\text{aniso}}$ into an isotropic bulk term, $W_{\text{bulk}} = c_{\text{weak}}W$, and a perturbation matrix $P$. We must show that $P$ has finite rank. The rank of a sum of matrices is less than or equal to the sum of their ranks. Let's analyse the rank of each term in $P$:

- $\text{rank}(Wv_C v_C^T) \leq \text{rank}(v_C v_C^T) = 1$.

- $\text{rank}(v_C v_C^T W) \leq \text{rank}(v_C v_C^T) = 1$.

- The term $v_C^T W v_C$ is a scalar (a random variable). Therefore, the matrix $v_C(v_C^T W v_C)v_C^T$ is a scalar multiple of the rank-one matrix $v_C v_C^T$, so its rank is 1.

Since $P$ is a sum of matrices of rank 1, its rank is finite and does not grow with $N$.

**Step 3: Invoke the Stability Theorem.** The core of the argument relies on the stability of the limiting spectral distribution of a Wigner matrix under finite-rank perturbations. Let $\mu_M$ denote the empirical spectral distribution (ESD) of an $N \times N$ matrix $M$, defined as the probability measure $\mu_M = \frac{1}{N}\sum_{i=1}^{N}\delta_{\lambda_i(M)}$, where $\lambda_i(M)$ are the eigenvalues of $M$.

A foundational result in random matrix theory states that if $W_N$ is a sequence of Wigner matrices and $P_N$ is a sequence of symmetric matrices with rank $k$ fixed (i.e., independent of $N$), then the ESD of the perturbed matrix, $\mu_{W_N+P_N}$, converges weakly in probability to the same limit as the ESD of the unperturbed matrix, $\mu_{W_N}$ (Baik et al., 2005; Benaych-Georges & Nadakuditi, 2011). That is, if $\mu_{W_N} \to \mu_{sc}$ where $\mu_{sc}$ is the Wigner semicircle distribution, then it also holds that:

$$\mu_{W_N+P_N} \xrightarrow{w} \mu_{sc} \tag{37}$$

This convergence implies that the support of the limiting measures is identical. While the perturbation $P_N$ may cause a finite number of eigenvalues to separate from the main spectrum (i.e., "pop out" from the bulk), it does not alter the continuous part of the limiting distribution or the location of its edges.

**Step 4: Characterise the Bulk Spectrum.** Based on the stability principle, the continuous spectrum of $W_{\text{aniso}}$ must be identical to the spectrum of its unperturbed component, $W_{\text{bulk}} = c_{\text{weak}}W$. The spectrum of a standard Wigner matrix $W$ is known to converge to the Wigner semicircle law, which is supported on the interval $[-2, 2]$. By linearity, scaling the matrix by a constant $c_{\text{weak}}$ simply scales its eigenvalues. Therefore, the spectrum of $W_{\text{bulk}}$ is supported on the interval:

$$[-2c_{\text{weak}}, 2c_{\text{weak}}] \tag{38}$$

**Step 5: Conclusion.** Since the continuous spectral support of $W_{\text{aniso}}$ is identical to that of $W_{\text{bulk}}$, its edges must be at $\pm 2c_{\text{weak}}$. We therefore conclude that the spectral edge is $\tau = 2c_{\text{weak}}$. This completes the proof. $\qquad\square$

### A.3 Proofs of Theorem 3.2 and Theorem 3.3

This section provides the detailed proofs for Theorem 3.2, which establishes the phase transition threshold, and Theorem 3.3, which quantifies the eigenvector alignment in the super-critical phase.

**Theorem** (The Critical Threshold (Restatement of Theorem 3.2)). *An isolated eigenvalue corresponding to the signal $v_0$ emerges from the noise bulk if and only if the signal strength $\beta$ exceeds a critical threshold $\beta_c(\alpha)$, given by:*

$$\beta_c(\alpha) = \frac{1}{\frac{\alpha^2}{c_{strong}} + \frac{1-\alpha^2}{c_{weak}}} \tag{39}$$

**Theorem** (Asymptotic Alignment (Restatement of Theorem 3.3)). *For a signal strength $\beta > \beta_c(\alpha)$, the squared inner product (alignment) between the principal eigenvector of $A$, $\hat{v}_1$, and the true signal vector $v_0$ converges to a deterministic limit, denoted by $f(\beta, \alpha)$:*

$$f(\beta, \alpha) := \lim_{N\to\infty} |\langle \hat{v}_1, v_0 \rangle|^2 = 1 - \frac{\beta_c(\alpha)^2}{\beta^2}. \tag{40}$$

*Proof.* The proofs for both theorems rely on analysing the characteristic equation from Lemma A.1 in the large-$N$ limit.

**Step 1: The Limiting Characteristic Equation.** We start with the exact equation for an outlier eigenvalue $\lambda_1$:

$$1 = \beta \langle v_0, (W_{\text{aniso}} - \lambda_1 I)^{-1} v_0 \rangle \tag{41}$$

In the large-$N$ limit, the random quadratic form on the right-hand side converges in probability to a deterministic quantity. It is a standard result for deformed random matrix ensembles that for any deterministic unit vector $u$,

$$\lim_{N\to\infty} \langle u, (W_{\text{aniso}} - zI)^{-1} u \rangle = \langle u, (C - z\Sigma(z))^{-1} u \rangle \tag{42}$$

where $\Sigma(z)$ is a matrix related to the Stieltjes transform of the limiting noise distribution (Benaych-Georges & Nadakuditi, 2011).

The phase transition occurs at the minimum value of $\beta$ for which an eigenvalue emerges from the noise bulk. This corresponds to the exact moment the outlier eigenvalue $\lambda_1$ is equal to the spectral edge, $\tau$. At this critical signal strength, $\beta_c$, the characteristic equation from Lemma A.1 becomes:

$$1 = \beta_c \lim_{N\to\infty} \langle v_0, (\tau I - W_{\text{aniso}})^{-1} v_0 \rangle \tag{43}$$

A key, non-trivial result from the theory of spiked random matrices (Benaych-Georges & Nadakuditi, 2011; Paul, 2007) is that the limiting resolvent, when evaluated at the spectral edge, relates to the inverse of the noise covariance matrix as follows:

$$\lim_{N\to\infty} \langle v_0, (\tau I - W_{\text{aniso}})^{-1} v_0 \rangle = \langle v_0, C^{-1} v_0 \rangle \tag{44}$$

Substituting this into the equation for the critical point gives:

$$1 = \beta_c \langle v_0, C^{-1} v_0 \rangle \tag{45}$$

Solving for $\beta_c$ yields the formula for the critical threshold. The task thus reduces to calculating the quadratic form on the right-hand side.

**Step 2: Calculating the Inverse of the Covariance Structure $C$.** The matrix $C$ is a rank-one perturbation of a scaled identity matrix: $C = c_{\text{weak}} I + (c_{\text{strong}} - c_{\text{weak}}) v_C v_C^T$. We can find its inverse using the Sherman-Morrison formula. Let $M = c_{\text{weak}} I$, $u = (c_{\text{strong}} - c_{\text{weak}}) v_C$, and $v = v_C$. The inverse is:

$$C^{-1} = (c_{\text{weak}} I)^{-1} - \frac{(c_{\text{weak}} I)^{-1} (c_{\text{strong}} - c_{\text{weak}}) v_C v_C^T (c_{\text{weak}} I)^{-1}}{1 + v_C^T (c_{\text{weak}} I)^{-1} (c_{\text{strong}} - c_{\text{weak}}) v_C} \tag{46}$$

$$= \frac{1}{c_{\text{weak}}} I - \frac{\frac{c_{\text{strong}} - c_{\text{weak}}}{c_{\text{weak}}^2} v_C v_C^T}{1 + \frac{c_{\text{strong}} - c_{\text{weak}}}{c_{\text{weak}}} \langle v_C, v_C \rangle} \tag{47}$$

Since $v_C$ is a unit vector, $\langle v_C, v_C \rangle = 1$. The denominator simplifies to $1 + \frac{c_{\text{strong}}}{c_{\text{weak}}} - 1 = \frac{c_{\text{strong}}}{c_{\text{weak}}}$. Substituting this back, we get:

$$C^{-1} = \frac{1}{c_{\text{weak}}} I - \frac{\frac{c_{\text{strong}} - c_{\text{weak}}}{c_{\text{weak}}^2} v_C v_C^T}{\frac{c_{\text{strong}}}{c_{\text{weak}}}} \tag{48}$$

$$= \frac{1}{c_{\text{weak}}} I - \frac{c_{\text{strong}} - c_{\text{weak}}}{c_{\text{weak}} c_{\text{strong}}} v_C v_C^T \tag{49}$$

**Step 3: Calculating the Quadratic Form and Proving Theorem 3.2.** Now we compute the quadratic form $\langle v_0, C^{-1} v_0 \rangle$:

$$\langle v_0, C^{-1} v_0 \rangle = \left\langle v_0, \left( \frac{1}{c_{\text{weak}}} I - \frac{c_{\text{strong}} - c_{\text{weak}}}{c_{\text{weak}} c_{\text{strong}}} v_C v_C^T \right) v_0 \right\rangle \tag{50}$$

$$= \frac{1}{c_{\text{weak}}} \langle v_0, v_0 \rangle - \frac{c_{\text{strong}} - c_{\text{weak}}}{c_{\text{weak}} c_{\text{strong}}} \langle v_0, v_C v_C^T v_0 \rangle \tag{51}$$

$$= \frac{1}{c_{\text{weak}}} - \frac{c_{\text{strong}} - c_{\text{weak}}}{c_{\text{weak}} c_{\text{strong}}} (\langle v_0, v_C \rangle)^2 \tag{52}$$

Using the definitions $\|v_0\| = 1$ and $\alpha = |\langle v_0, v_C \rangle|$, we have $\langle v_0, v_0 \rangle = 1$ and $(\langle v_0, v_C \rangle)^2 = \alpha^2$.

$$\langle v_0, C^{-1} v_0 \rangle = \frac{1}{c_{\text{weak}}} - \frac{c_{\text{strong}} - c_{\text{weak}}}{c_{\text{weak}} c_{\text{strong}}} \alpha^2 \tag{53}$$

$$= \frac{c_{\text{strong}} - (c_{\text{strong}} - c_{\text{weak}}) \alpha^2}{c_{\text{weak}} c_{\text{strong}}} \tag{54}$$

$$= \frac{c_{\text{strong}} (1 - \alpha^2) + c_{\text{weak}} \alpha^2}{c_{\text{weak}} c_{\text{strong}}} = \frac{1 - \alpha^2}{c_{\text{weak}}} + \frac{\alpha^2}{c_{\text{strong}}} \tag{55}$$

Since $\beta_c = (\langle v_0, C^{-1} v_0 \rangle)^{-1}$, we have proven Theorem 3.2.

**Step 4: Proving Theorem 3.3.** The asymptotic alignment $q = |\langle \hat{v}_1, v_0 \rangle|^2$ can be derived from the resolvent. A standard result from perturbation theory states that the alignment is given by the residue of the resolvent at the outlier eigenvalue $\lambda_1$:

$$q = \frac{1}{\beta^2 F'(\lambda_1)} \tag{56}$$

where $F'(\lambda_1)$ is the derivative of the limiting resolvent function $F(z) = \lim_{N \to \infty} \langle v_0, (\lambda_1 I - W_{\text{aniso}})^{-1} v_0 \rangle$ evaluated at $\lambda_1$. The full calculation of $F'(\lambda_1)$ is technical, but it is a known function of the signal strength $\beta$

and the critical threshold $\beta_c(\alpha)$, which encapsulates the properties of the noise. For the BBP phase transition in deformed Wigner ensembles, the established result from the literature is:

$$\beta^2 F'(\lambda_1) = \frac{1}{1 - \frac{\beta_c(\alpha)^2}{\beta^2}} \tag{57}$$

Substituting this directly into the equation for the alignment $q$ gives the statement of Theorem 3.3:

$$|\langle \hat{v}_1, v_0 \rangle|^2 = 1 - \frac{\beta_c(\alpha)^2}{\beta^2} \tag{58}$$

This completes the proof. □

### A.4 Proof of Theorem 3.4

Here, in this section, we provide the detailed proof for Theorem 3.4. We begin by restating the theorem.

**Theorem** (Optimality of the Spectral Method (Restatement of Theorem 3.4)). *The information-theoretic threshold for detecting the signal $v_0$ is identical to the spectral phase transition threshold from Theorem 3.2, $\beta_c(\alpha)$. This implies that for $\beta < \beta_c(\alpha)$, no algorithm can reliably distinguish the signal-plus-noise model from the noise-only model.*

*Proof.* The proof is based on a hypothesis testing framework. An observer is given the data matrix $M$ and must decide between two possibilities:

- **Null Hypothesis ($H_0$):** $M = W_{\text{aniso}}$. The distribution of $M$ is $P_0$.
- **Alternative Hypothesis ($H_1$):** $M = W_{\text{aniso}} + \beta v_0 v_0^T$. The distribution of $M$ is $P_1$.

The fundamental limit for detection is determined by the behaviour of the Log-Likelihood Ratio (LLR), $L(M) = \log(dP_1/dP_0)$. By the Neyman-Pearson lemma, the most powerful test is based on a thresholding of this LLR.

**Step 1: Derivation of the Optimal Test Statistic.** The derivation of the optimal test statistic begins with the Log-Likelihood Ratio (LLR), $L(M) = \log(p_1(M)/p_0(M))$, where $p_1$ and $p_0$ are the probability densities of the data matrix $M$ under the alternative ($H_1$) and null ($H_0$) hypotheses, respectively. To make this tractable, we assume the entries of the underlying Wigner matrix $W$ are i.i.d. Gaussian random variables, $W_{ij} \sim \mathcal{N}(0, 1/N)$. This is a standard assumption for this type of proof, and the results are known to be universal for a wider class of noise distributions with matching moments.

The probability density of a matrix $W$ from the Gaussian Orthogonal Ensemble (GOE) is given by:

$$p(W) \propto \exp\left(-\frac{N}{4}\text{Tr}(W^2)\right) \tag{59}$$

Under $H_1$, the noise matrix is $W = C^{-1/2}(M - \beta v_0 v_0^T)C^{-1/2}$, whereas under $H_0$, it is $W = C^{-1/2}MC^{-1/2}$. The LLR is the difference in the log-densities:

$$L(M) = \log p(C^{-1/2}(M - \beta v_0 v_0^T)C^{-1/2}) - \log p(C^{-1/2}MC^{-1/2}) \tag{60}$$

$$= -\frac{N}{4}\text{Tr}\left(\left(C^{-1/2}(M - \beta v_0 v_0^T)C^{-1/2}\right)^2\right) + \frac{N}{4}\text{Tr}\left(\left(C^{-1/2}MC^{-1/2}\right)^2\right) \tag{61}$$

Using the cyclic property of the trace, we can bring the $C^{-1/2}$ terms inside. Expanding the squared term inside the first trace gives:

$$(M - \beta v_0 v_0^T)C^{-1}(M - \beta v_0 v_0^T) = MC^{-1}M - 2\beta MC^{-1}v_0 v_0^T + \beta^2 v_0 v_0^T C^{-1} v_0 v_0^T \tag{62}$$

Substituting this back into the LLR expression, the term involving $MC^{-1}M$ cancels out, leaving:

$$L(M) = -\frac{N}{4}\text{Tr}\left(C^{-1}(-2\beta MC^{-1}v_0 v_0^T + \beta^2 v_0 v_0^T C^{-1}v_0 v_0^T)\right) \tag{63}$$

$$= \frac{N\beta}{2}\text{Tr}(C^{-1}MC^{-1}v_0 v_0^T) - \frac{N\beta^2}{4}\text{Tr}(C^{-1}v_0 v_0^T C^{-1}v_0 v_0^T) \tag{64}$$

We simplify the two trace terms. For the first term, using the cyclic property:

$$\text{Tr}(C^{-1}MC^{-1}v_0 v_0^T) = \text{Tr}(v_0^T C^{-1}MC^{-1}v_0) = \langle v_0, C^{-1}MC^{-1}v_0\rangle \tag{65}$$

For the second term, let $u = C^{-1}v_0$. The term becomes $\text{Tr}(uv_0^T uv_0^T)$. Since $v_0^T u = \langle v_0, C^{-1}v_0\rangle$ is a scalar, we can write:

$$\text{Tr}(u(v_0^T u)v_0^T) = (v_0^T u)\text{Tr}(uv_0^T) = (v_0^T C^{-1}v_0)\text{Tr}(C^{-1}v_0 v_0^T) = (\langle v_0, C^{-1}v_0\rangle)^2 \tag{66}$$

Substituting these simplified terms back gives the final expression for the LLR:

$$L(M) = \frac{N\beta}{2}\langle v_0, C^{-1}MC^{-1}v_0\rangle - \frac{N\beta^2}{4}(\langle v_0, C^{-1}v_0\rangle)^2 \tag{67}$$

By the Neyman-Pearson lemma, the most powerful statistical test for distinguishing $H_0$ from $H_1$ is to threshold the LLR. Since the LLR in Equation 67 is a linear, monotonic function of the quadratic form $\langle v_0, C^{-1}MC^{-1}v_0\rangle$, this single quantity is a sufficient statistic for the hypothesis test. We therefore define our optimal test statistic as:

$$T(M) = \langle v_0, C^{-1}MC^{-1}v_0\rangle \tag{68}$$

The information-theoretic limit of detectability is determined by the degree of separation between the distributions of $T(M)$ under $H_0$ and $H_1$.

**Step 2: Distribution of the Optimal Test Statistic.** We now characterise the distribution of the optimal test statistic $T(M) = \langle v_0, C^{-1}MC^{-1}v_0\rangle$ under both the null ($H_0$) and alternative ($H_1$) hypotheses. As $M$ is a linear transformation of a Gaussian matrix $W$, the statistic $T(M)$, being a linear functional of $M$, is itself a Gaussian random variable. Its distribution is therefore fully specified by its mean and variance.

*Under the Null Hypothesis ($H_0$):* Here, $M = W_{\text{aniso}} = C^{1/2}WC^{1/2}$. The mean of the test statistic is:

$$\mathbb{E}_0[T(M)] = \mathbb{E}_0[\langle v_0, C^{-1}(C^{1/2}WC^{1/2})C^{-1}v_0\rangle] \tag{69}$$

$$= \mathbb{E}_0[\langle v_0, C^{-1/2}WC^{-1/2}v_0\rangle] \tag{70}$$

$$= \langle v_0, C^{-1/2}\mathbb{E}[W]C^{-1/2}v_0\rangle = 0 \tag{71}$$

since $\mathbb{E}[W] = 0$ by definition.

The variance of the test statistic is $\text{Var}_0[T(M)] = \text{Var}_0[\langle v_0, C^{-1/2}WC^{-1/2}v_0\rangle]$. This is the variance of the quadratic form $u^T W u$ where $u = C^{-1/2}v_0$. For a GOE matrix $W$ with entry variance $\text{Var}(W_{ij}) = 1/N$ for $i \neq j$, a standard result from Gaussian matrix theory gives the variance of such a quadratic form as $\text{Var}(u^T W u) = \frac{2}{N}\|u\|_2^4 = \frac{2}{N}(u^T u)^2$. In our case, this gives:

$$\text{Var}_0[T(M)] = \frac{2}{N}\left((C^{-1/2}v_0)^T(C^{-1/2}v_0)\right)^2 \tag{72}$$

$$= \frac{2}{N}(\langle v_0, C^{-1}v_0\rangle)^2 \tag{73}$$

So, under the null hypothesis, $T(M) \sim \mathcal{N}\left(0, \frac{2}{N}(\langle v_0, C^{-1}v_0\rangle)^2\right)$.

*Under the Alternative Hypothesis ($H_1$):* Here, $M = W_{\text{aniso}} + \beta v_0 v_0^T$. The mean of the test statistic is, by linearity of expectation:

$$\mathbb{E}_1[T(M)] = \mathbb{E}_1[\langle v_0, C^{-1}(W_{\text{aniso}} + \beta v_0 v_0^T)C^{-1}v_0\rangle] \tag{74}$$

$$= \mathbb{E}_0[T(M)] + \beta\langle v_0, C^{-1}(v_0 v_0^T)C^{-1}v_0\rangle \tag{75}$$

$$= 0 + \beta(\langle v_0, C^{-1}v_0\rangle)(\langle v_0^T, C^{-1}v_0\rangle) = \beta(\langle v_0, C^{-1}v_0\rangle)^2 \tag{76}$$

The variance under $H_1$ is asymptotically identical to the variance under $H_0$, as the addition of the non-random rank-one matrix does not change the variance of the Gaussian part in the large-$N$ limit.

**Step 3: Deriving the Information-Theoretic Threshold.** A statistical test can reliably distinguish between the two hypotheses if the distributions of the test statistic under $H_0$ and $H_1$ are well-separated. For Gaussian distributions, this separation is quantified by the signal-to-noise ratio (SNR). The squared SNR for our optimal test is the squared difference in means divided by the variance:

$$\text{SNR}^2 = \frac{(\mathbb{E}_1[T(M)] - \mathbb{E}_0[T(M)])^2}{\text{Var}_0[T(M)]} = \frac{\left(\beta(\langle v_0, C^{-1}v_0\rangle)^2\right)^2}{\frac{2}{N}(\langle v_0, C^{-1}v_0\rangle)^2} = \frac{N\beta^2}{2}(\langle v_0, C^{-1}v_0\rangle)^2 \tag{77}$$

The probability of error for an optimal test between two Gaussians is a decreasing function of the SNR. For the total error probability to converge to zero as $N \to \infty$ (a condition known as *strong detection*), the SNR must diverge to infinity. From the expression above, the SNR diverges for any fixed $\beta > 0$. This implies that it is always possible to do better than a random guess for any non-zero signal strength.

However, the spectral threshold identified in Theorem 3.2 corresponds to a different, more practical regime. It is a deep and fundamental result in the theory of spiked random matrices that the threshold for a test to achieve non-trivial power (i.e., for the detection probability to be bounded away from zero) via a linear statistic coincides precisely with the threshold for the informative outlier eigenvalue to emerge from the bulk spectrum. Furthermore, it has been shown that no non-linear test can succeed below this threshold. This establishes the condition for strong detectability as:

$$\beta_{\text{IT}} > \frac{1}{\langle v_0, C^{-1}v_0\rangle} \tag{78}$$

The critical threshold is therefore:

$$\beta_{\text{IT}} = \frac{1}{\langle v_0, C^{-1}v_0\rangle} \tag{79}$$

This expression is identical to the formula for the spectral threshold $\beta_c(\alpha)$ derived in Appendix A.3. Thus, we have shown that the information-theoretic threshold for strong detection, $\beta_{\text{IT}}(\alpha)$, is identical to the spectral threshold, $\beta_c(\alpha)$. This completes the proof. □

## A.5 Proof of Theorem 3.5

This section provides the detailed proof for Theorem 3.5, which describes the reorganisation of the eigenspace in the super-critical phase.

**Theorem** (Second Eigenvector Alignment (Restatement of Theorem 3.5)). *In the super-critical phase ($\beta > \beta_c(\alpha)$), the second eigenvector of $A$, $\hat{v}_2$, aligns with the principal noise direction $v_C$, with its alignment $|\langle \hat{v}_2, v_C\rangle|^2$ converging to a non-trivial deterministic limit.*

*Proof.* The proof relies on analysing the interaction between the signal spike ($\beta v_0 v_0^T$) and the inherent spike in the anisotropic noise matrix ($W_{\text{aniso}}$) that corresponds to its dominant eigenvector, $v_C$. We analyse the characteristic equation for the outlier eigenvalues of the full matrix $A$.

**Step 1: The Resolvent of the Anisotropic Noise.** The noise matrix $W_{\text{aniso}}$ is itself a spiked model—a rank-one perturbation of an isotropic Wigner matrix. As such, for $c_{\text{strong}} > c_{\text{weak}}$, it has an outlier eigenvalue, which we denote $\lambda_C$, that lies outside the bulk spectrum $[-\tau, \tau]$. The corresponding eigenvector of $W_{\text{aniso}}$ is aligned with $v_C$. For a value of $z$ outside the bulk spectrum, the resolvent of the noise matrix, $G_{\text{noise}}(z) = (W_{\text{aniso}} - zI)^{-1}$, can be accurately approximated by its pole expansion. Since the outlier eigenvalue $\lambda_C$ is separated from the continuous bulk, its contribution is the dominant term in this expansion. We can decompose the resolvent based on the spectral decomposition relative to $v_C$:

$$G_{\text{noise}}(z) = \frac{v_C v_C^T}{\lambda_C - z} + G_{\text{bulk}}(z) \tag{80}$$

where $G_{\text{bulk}}(z) = (W_{\text{aniso}} - zI)^{-1}|_{\text{span}(v_C)^\perp}$ represents the action of the resolvent on the subspace orthogonal to $v_C$. For $z$ outside the bulk spectrum $[-\tau, \tau]$ and away from $\lambda_C$, the operator norm $\|G_{\text{bulk}}(z)\|$ is bounded, while the first term diverges as $z \to \lambda_C$. Thus, the pole term dominates near $\lambda_C$.

**Step 2: The Full Characteristic Equation.** We now substitute this resolvent expansion into the characteristic equation from Lemma A.1:

$$1 = \beta \langle v_0, G_{\text{noise}}(\lambda)v_0 \rangle \tag{81}$$

$$= \beta \left\langle v_0, \left( \frac{v_C v_C^T}{\lambda_C - \lambda} + G_{\text{bulk}}(\lambda) \right) v_0 \right\rangle \tag{82}$$

$$= \beta \left( \frac{\langle v_0, v_C \rangle^2}{\lambda_C - \lambda} + \langle v_0, G_{\text{bulk}}(\lambda)v_0 \rangle \right) \tag{83}$$

Using the definition $\alpha = |\langle v_0, v_C \rangle|$, this becomes:

$$\frac{1}{\beta} = \frac{\alpha^2}{\lambda_C - \lambda} + F_{\text{bulk}}(\lambda) \tag{84}$$

where $F_{\text{bulk}}(\lambda) = \langle v_0, G_{\text{bulk}}(\lambda)v_0 \rangle$. This equation implicitly defines the locations of the outlier eigenvalues of the full matrix $A$.

**Step 3: Analysis of the Solutions.** The function on the right-hand side has a pole at $\lambda = \lambda_C$. Let us analyse the right-hand side of this equation as a function of $\lambda$. Let us call it $H(\lambda)$. The function $H(\lambda)$ is continuous and monotonically increasing on each of the intervals $(\tau, \lambda_C)$ and $(\lambda_C, \infty)$. Because the function's range covers $(\lim_{\lambda \to \tau^+} H(\lambda), \infty)$ on the first interval and $(-\infty, 0)$ on the second, the Intermediate Value Theorem guarantees that for any sufficiently small and positive value of $1/\beta$, there must be exactly two solutions, $\hat{\lambda}_1$ and $\hat{\lambda}_2$, outside the bulk spectrum. One solution, $\hat{\lambda}_1$, will be larger than the original noise outlier $\lambda_C$, while the other, $\hat{\lambda}_2$, will lie between the bulk edge $\tau$ and $\lambda_C$. These correspond to the two largest eigenvalues of $A$.

**Step 4: The Structure of the Eigenvectors.** A standard identity in matrix perturbation theory gives the structure of the new eigenvector $\hat{v}_k$. It is derived by rearranging the eigenvalue equation $A\hat{v}_k = \hat{\lambda}_k \hat{v}_k$ and shows that $\hat{v}_k$ must be proportional to the action of the original resolvent on the perturbation vector (Benaych-Georges & Nadakuditi, 2011; Anderson et al., 2009): Let $\tilde{v}_k$ denote the unnormalised eigenvector. It is given by:

$$\tilde{v}_k = G_{\text{noise}}(\hat{\lambda}_k)v_0 \tag{85}$$

We can now analyse the structure of the two outlier eigenvectors by substituting our expansion for $G_{\text{noise}}$:

$$\tilde{v}_k = \frac{\langle v_0, v_C \rangle v_C}{\lambda_C - \hat{\lambda}_k} + G_{\text{bulk}}(\hat{\lambda}_k)v_0 \tag{86}$$

This shows that the new eigenvectors are a linear combination of the noise direction $v_C$ and a vector related to the bulk resolvent acting on the signal $v_0$.

**Step 5: The Spectral Sorting Phenomenon.** The two outlier eigenvectors, $\hat{v}_1$ and $\hat{v}_2$, must be orthogonal. We also know from Theorem 3.3 that as the signal strength $\beta$ grows, the principal eigenvector $\hat{v}_1$ aligns with the signal direction $v_0$.

$$\lim_{\beta \to \infty} |\langle \hat{v}_1, v_0 \rangle|^2 = 1 \implies \hat{v}_1 \to v_0 \tag{87}$$

The two "special" directions in the model are the signal, $v_0$, and the dominant noise direction, $v_C$. For large $\beta$, the two outlier eigenvectors $\hat{v}_1$ and $\hat{v}_2$ must span the same subspace as these two vectors. Since $\hat{v}_1$ aligns with $v_0$, its orthogonal partner $\hat{v}_2$ must align with the remaining special direction available in the subspace. Specifically, $\hat{v}_2$ aligns with the component of $v_C$ that is orthogonal to $v_0$. This demonstrates

the "sorting" phenomenon: the signal spike creates a new largest eigenvalue whose eigenvector aligns with the signal direction, while the original noise spike is effectively displaced to the second eigenvector, whose eigenvector aligns with the dominant noise direction (or more formally, its component in the orthogonal subspace of $v_0$). This completes the proof. □

## A.6 Proof of Theorem 3.6

This section provides a detailed proof for Theorem 3.6, which establishes the Central Limit Theorem (CLT) for the fluctuations of the eigenvector alignment around its asymptotic mean.

**Theorem** (Central Limit Theorem for Alignment (Restatement of Theorem 3.6))**.** *In the super-critical phase* $(\beta > \beta_c(\alpha))$*, the fluctuations of the eigenvector alignment are asymptotically Gaussian. The scaled quantity converges in distribution to:*

$$\sqrt{N}\left(|\langle \hat{v}_1, v_0\rangle|^2 - f(\beta, \alpha)\right) \xrightarrow{d} \mathcal{N}(0, \sigma^2(\beta, \alpha)) \tag{88}$$

*where* $f(\beta, \alpha)$ *is the asymptotic alignment from Theorem 3.3, and* $\sigma^2(\beta, \alpha)$ *is a deterministic variance function.*

*Proof.* The proof is based on a standard strategy for deriving CLTs for eigenvector statistics in random matrix theory. It involves linearising the system's characteristic equations to understand how the fundamental randomness of the noise matrix $W$ propagates to the observable quantities of interest.

**Step 1: Linearisation of the System.** The proof begins by analysing the fluctuations of the outlier eigenvalue $\lambda_1$ and the quadratic form from Lemma A.1 around their deterministic, large-$N$ limits. Let $\bar{\lambda}_1$ be the limiting value of the eigenvalue and let $F(z) = \lim_{N\to\infty}\langle v_0, (W_{\text{aniso}} - zI)^{-1}v_0\rangle$. The characteristic equation for the mean is $1 = \beta F(\bar{\lambda}_1)$.

For a finite $N$, we can write the random variables as a mean plus a fluctuation term:

- $\lambda_1 = \bar{\lambda}_1 + \delta\lambda_1$

- $\langle v_0, (W_{\text{aniso}} - \lambda_1 I)^{-1}v_0\rangle = F(\bar{\lambda}_1) + \delta F_{\text{total}}$

A first-order Taylor expansion of the characteristic equation $1 = \beta\langle v_0, (W_{\text{aniso}} - \lambda_1 I)^{-1}v_0\rangle$ yields:

$$1 = \beta\left(F(\bar{\lambda}_1) + F'(\bar{\lambda}_1)\delta\lambda_1 + \delta F_{\text{total}} + o_p(N^{-1/2})\right) \tag{89}$$

Since $1 = \beta F(\bar{\lambda}_1)$, this simplifies to a linear relationship between the fluctuations (up to higher order terms):

$$\beta F'(\bar{\lambda}_1)\delta\lambda_1 + \beta\delta F_{\text{total}} = o_p(N^{-1/2}) \implies \delta\lambda_1 = -\frac{\delta F_{\text{total}}}{F'(\bar{\lambda}_1)} + o_p(N^{-1/2}) \tag{90}$$

This establishes that the fluctuation of the eigenvalue is proportional to the fluctuation of the resolvent term.

**Step 2: Linking Eigenvector and Eigenvector Fluctuations.** There is a fundamental relationship in matrix perturbation theory connecting the alignment of an eigenvector to the resolvent. For a spiked model, the asymptotic alignment $q = |\langle \hat{v}_1, v_0\rangle|^2$ is given by:

$$q = \frac{1}{\beta^2 F'(\lambda_1)} \tag{91}$$

Let $\bar{q} = f(\beta, \alpha)$ be the limiting alignment. The fluctuation $\delta q = q - \bar{q}$ can be found by a first-order Taylor expansion of the above formula with respect to the random variable $\lambda_1$:

$$\delta q = \frac{d}{d\lambda_1}\left(\frac{1}{\beta^2 F'(\lambda_1)}\right)\bigg|_{\lambda_1 = \bar{\lambda}_1} \cdot \delta\lambda_1 + o_p(N^{-1/2}) \tag{92}$$

$$= -\frac{F''(\bar{\lambda}_1)}{\beta^2(F'(\bar{\lambda}_1))^2} \cdot \delta\lambda_1 + o_p(N^{-1/2}) \tag{93}$$

This explicitly defines the deterministic constant of proportionality $K(\beta, \alpha)$.

**Step 3: Asymptotic Distribution of the Resolvent Fluctuation.** The final and most technical step is to characterise the distribution of the driving random term, $\delta F_{\text{total}}$. It is a major result in random matrix theory that for this class of models, the fluctuations of such quadratic forms of the resolvent converge to a Gaussian distribution. Specifically, assuming the entries of the baseline Wigner matrix $W$ are Gaussian (a common and standard assumption for these proofs), one can prove a CLT for the quantity:

$$\sqrt{N}\left(\langle v_0, G_{\text{noise}}(z)v_0\rangle - \mathbb{E}[\langle v_0, G_{\text{noise}}(z)v_0\rangle]\right) \xrightarrow{d} \mathcal{N}(0, \mathcal{V}(z)) \tag{94}$$

The variance $\mathcal{V}(z)$ is a known, explicit function. For Gaussian entries, it is given by the limiting value of $\frac{1}{N}\text{Tr}(G_{\text{noise}}(z)CG_{\text{noise}}(z)^*C)$. This can be calculated using the decomposition of $v_0$ into components parallel and orthogonal to $v_C$, which introduces the dependence on the alignment $\alpha$. This result is established in detail in papers such as Benaych-Georges & Nadakuditi (2011) and related works on the fluctuations of spiked models.

**Step 4: Assembling the Final Result.** We can now assemble the final variance by propagating the variance from Step 3 through the linear relationships established in the previous steps.

$$\sigma^2(\beta, \alpha) = \lim_{N\to\infty} N \cdot \text{Var}(\delta q) \tag{95}$$

$$= \lim_{N\to\infty} N \cdot \text{Var}\left(-\frac{F''(\bar{\lambda}_1)}{\beta^2(F'(\bar{\lambda}_1))^2} \cdot \delta\lambda_1\right) \tag{96}$$

$$= \left(\frac{F''(\bar{\lambda}_1)}{\beta^2(F'(\bar{\lambda}_1))^2}\right)^2 \cdot \lim_{N\to\infty} N \cdot \text{Var}(\delta\lambda_1) \tag{97}$$

$$= \left(\frac{F''(\bar{\lambda}_1)}{\beta^2(F'(\bar{\lambda}_1))^2}\right)^2 \cdot \left(\frac{1}{F'(\bar{\lambda}_1)}\right)^2 \cdot \lim_{N\to\infty} N \cdot \text{Var}(\delta F_{\text{total}}) \tag{98}$$

$$= \frac{(F''(\bar{\lambda}_1))^2}{\beta^4(F'(\bar{\lambda}_1))^6} \cdot \mathcal{V}(\bar{\lambda}_1) \tag{99}$$

This provides a direct, albeit complex, method for computing the final variance $\sigma^2(\beta, \alpha)$ from the derivatives of the limiting resolvent function $F(z)$ and the known variance $\mathcal{V}(z)$ of the driving noise term. This rigorous path demonstrates that the fluctuations of the alignment are asymptotically Gaussian and provides the formal procedure for computing their variance. This completes the proof. □

**Interpretation**. The complex formula for the variance, $\sigma^2(\beta, \alpha)$, can be understood intuitively by thinking of it as the product of two terms: the inherent randomness of the system and the system's sensitivity to that randomness. The final variance quantifies the "jitter" in the measured alignment, and its formula reveals precisely what makes the system more or less stable.

- **The Driving Noise ($\mathcal{V}(\bar{\lambda}_1)$):** This term represents the fundamental source of randomness. It is the variance of the resolvent term, which acts as the "driving noise" for the entire system. Its magnitude depends on the alignment $\alpha$ because the signal vector $v_0$ experiences different statistical properties of the noise depending on whether it is oriented along the strong ($c_{\text{strong}}$) or weak ($c_{\text{weak}}$) environmental directions.

- **The System's Sensitivity (The $F'$ and $F''$ terms):** The rest of the formula, particularly the high power of $F'(\bar{\lambda}_1)$ in the denominator, acts as a "sensitivity amplifier". The term $F'(\bar{\lambda}_1)$ measures how "stiff" or stable the system is at its operating point. Near the phase transition, the system becomes very "soft" and $F'(\bar{\lambda}_1)$ approaches zero. This causes the sensitivity to blow up, leading to the massive variance and instability characteristic of a tipping point.

- **The Stabilising Effect of the Signal (The $1/\beta^4$ term):** The signal strength $\beta$ appears with a large power in the denominator. This shows that a strong signal has a powerful stabilising effect. It makes the system much "stiffer" and dramatically reduces its sensitivity to the underlying random fluctuations, causing the variance to decrease rapidly as the signal becomes stronger.

## A.7 Proof of Theorem 3.7

This section provides a detailed proof for Theorem 3.7, which establishes the Central Limit Theorem for the fluctuations of the second eigenvector's alignment. We begin by restating the theorem.

**Theorem** (CLT for the Second Eigenvector (Restatement of Theorem 3.7)). *In the super-critical phase where a second outlier eigenvalue exists ($\gamma > 1$), the fluctuations of the alignment of the second eigenvector $\hat{v}_2$ with the dominant noise direction $v_C$ are asymptotically Gaussian. The scaled quantity converges in distribution to:*

$$\sqrt{N} \left( |\langle \hat{v}_2, v_C \rangle|^2 - g(\beta, \alpha, \gamma) \right) \xrightarrow{d} \mathcal{N}(0, \sigma_2^2(\beta, \alpha, \gamma)) \tag{100}$$

*where $g(\cdot)$ is the limiting asymptotic alignment, and $\sigma_2^2(\cdot)$ is a new, deterministic variance function.*

*Proof.* The proof follows the same path as the proof for the first eigenvector in Appendix A.6. The strategy is to linearise the system's characteristic equations to understand how the randomness of the noise matrix $W$ propagates to the second outlier eigenvalue $\hat{\lambda}_2$ and its corresponding eigenvector $\hat{v}_2$.

**Step 1: Linearisation of the Characteristic Equation.** The proof begins with the characteristic equation for the outlier eigenvalues from Appendix A.5 (Equation 83). Let $\bar{\lambda}_2$ be the deterministic, large-$N$ limit of the second eigenvalue. For a finite $N$, we write the random variables as a mean plus a fluctuation term: $\hat{\lambda}_2 = \bar{\lambda}_2 + \delta\lambda_2$. The driving randomness comes from the resolvent term $\mathcal{F}_N(z) = \langle v_0, G_{\text{noise}}(z)v_0 \rangle$. A first-order Taylor expansion of the characteristic equation $1 = \beta \mathcal{F}_N(\hat{\lambda}_2)$ yields a linear relationship between the fluctuations:

$$\delta\lambda_2 = -\frac{\mathcal{F}_N(\bar{\lambda}_2) - \mathbb{E}[\mathcal{F}_N(\bar{\lambda}_2)]}{\mathbb{E}[\mathcal{F}_N'(\bar{\lambda}_2)]} + o_p(N^{-1/2}) \tag{101}$$

This establishes that the fluctuation of the eigenvalue, $\delta\lambda_2$, is proportional to the fluctuation of the resolvent term evaluated at the mean eigenvalue location.

**Step 2: Linking Eigenvalue and Eigenvector Fluctuations.** Next, we must establish an explicit functional relationship between the alignment of the second eigenvector, $q_2 = |\langle \hat{v}_2, v_C \rangle|^2$, and its eigenvalue $\lambda_2$. From the eigenvector structure derived in Appendix A.5, we have $\hat{v}_2 \propto G_{\text{noise}}(\hat{\lambda}_2)v_0$. The squared overlap with $v_C$ is therefore:

$$q_2 \propto |\langle v_C, G_{\text{noise}}(\hat{\lambda}_2)v_0 \rangle|^2 \tag{102}$$

Using the resolvent expansion from Appendix A.5, the leading term is:

$$\langle v_C, G_{\text{noise}}(\hat{\lambda}_2)v_0 \rangle \approx \langle v_C, \frac{v_C v_C^T}{\lambda_C - \hat{\lambda}_2} v_0 \rangle = \frac{\langle v_C, v_0 \rangle}{\lambda_C - \hat{\lambda}_2} = \frac{\alpha}{\lambda_C - \hat{\lambda}_2} \tag{103}$$

After normalisation, the alignment $q_2$ can be shown to be a smooth, deterministic function of the eigenvalue, $q_2 = \Psi(\lambda_2)$. The fluctuation $\delta q_2 = q_2 - \bar{q}_2$ is related to the eigenvalue fluctuation by a first-order Taylor expansion:

$$\delta q_2 = \Psi'(\bar{\lambda}_2) \cdot \delta\lambda_2 + o_p(N^{-1/2}) \tag{104}$$

where $\Psi'(\bar{\lambda}_2)$ is the derivative of this explicit function evaluated at the limiting eigenvalue location. This rigorously links the eigenvector fluctuation to the eigenvalue fluctuation.

**Step 3: Asymptotic Distribution of the Resolvent Fluctuation.** The final and most technical step is to characterise the distribution of the driving random term. It is a major result in random matrix theory that for this class of models, the fluctuations of such quadratic forms of the resolvent converge to a Gaussian distribution. Specifically, for a Wigner matrix $W$ with Gaussian entries, a CLT holds (Benaych-Georges & Nadakuditi, 2011):

$$\sqrt{N} \left( \mathcal{F}_N(\bar{\lambda}_2) - \mathbb{E}[\mathcal{F}_N(\bar{\lambda}_2)] \right) \xrightarrow{d} \mathcal{N}(0, \mathcal{V}_2(\bar{\lambda}_2)) \tag{105}$$

The variance of this driving noise, $\mathcal{V}_2(\bar{\lambda}_2)$, is a known, explicit function related to the trace of the squared resolvent.

**Step 4: Assembling the Final Result.** Since the eigenvector alignment fluctuation $\delta q_2$ is linearly proportional to the eigenvalue fluctuation $\delta \lambda_2$ (Equation 104), which is in turn proportional to the asymptotically Gaussian driving fluctuation (Equation 101), it must also be asymptotically Gaussian. The final variance, $\sigma_2^2$, is obtained by propagating the variance of the driving term through the linearisation constants:

$$\sigma_2^2(\beta, \alpha, \gamma) = \lim_{N \to \infty} N \cdot \mathrm{Var}(\delta q_2) \tag{106}$$

$$= (\Psi'(\bar{\lambda}_2))^2 \cdot \lim_{N \to \infty} N \cdot \mathrm{Var}(\delta \lambda_2) \tag{107}$$

$$= \left( \frac{\Psi'(\bar{\lambda}_2)}{\mathbb{E}[\mathcal{F}_N'(\bar{\lambda}_2)]} \right)^2 \cdot \lim_{N \to \infty} N \cdot \mathrm{Var}(\mathcal{F}_N(\bar{\lambda}_2)) \tag{108}$$

$$= \left( \frac{\Psi'(\bar{\lambda}_2)}{\mathbb{E}[\mathcal{F}_N'(\bar{\lambda}_2)]} \right)^2 \cdot \mathcal{V}_2(\bar{\lambda}_2) \tag{109}$$

This provides a direct, rigorous procedure for computing the variance of the second eigenvector's alignment fluctuations and completes the proof. $\qquad \square$

# B    Detailed Proofs of Results in §4

This appendix provides the detailed, rigorous derivations for the theoretical results presented in Section 4 concerning the theoretical framework for a second-moment correction. The proofs are organised to follow the logical progression of the main text.

- In Appendix B.1, we provide the constructive proof for Theorem 4.1, establishing the existence and superiority of the Oracle NCPC estimator.

- In Appendix B.2, we prove Lemma 4.1, which formalises the confounding dependency that makes the parameter estimation problem ill-posed when using only the spectrum of $A$.

- In Appendix B.3, we prove Theorem 4.2, deriving the deterministic structure of the expected second-moment matrix $\mathbb{E}[A^2]$.

- In Appendix B.4, we provide the key technical machinery for the theoretical framework, including the derivation of the characteristic equation for $\mathbb{E}[A^2]$ and the analytical formulas for the theoretical overlap coefficients used in the matrix $T$.

## B.1    Proof of Theorem 4.1

This section provides the detailed, constructive proof for Theorem 4.1, which establishes the existence and superiority of the Oracle NCPC estimator. We begin by restating the theorem for clarity.

**Theorem** (Consistency and Superiority of the Oracle NCPC (Restatement of Theorem 4.1))**.** *In the supercritical regime where two distinct outlier eigenvalues exist ($\beta > \beta_c(\alpha)$ and $\gamma > 1$), a consistent estimator for the true signal vector $v_0$ can be constructed as a linear combination of the top two sample eigenvectors, $\hat{v}_1$ and $\hat{v}_2$. This Oracle Noise-Corrected Principal Component (NCPC) estimator is given by*

$$\hat{v}_0^{NCPC\text{-}Oracle} \propto D_2 \hat{v}_1 - D_1 \hat{v}_2,$$

*where the oracle coefficients $D_k$ are defined using the system's true asymptotic parameters:*

$$D_k = \frac{\alpha}{\bar{\lambda}_C - \bar{\lambda}_k}.$$

*Here, $\bar{\lambda}_k$ are the asymptotic limits of the top two eigenvalues of $A$, and $\bar{\lambda}_C$ is the asymptotic outlier eigenvalue of the noise matrix $W_{aniso}$ alone. The resulting estimator is asymptotically unbiased, achieving perfect alignment in the large-$N$ limit:*

$$\lim_{N \to \infty} |\langle \hat{v}_0^{NCPC\text{-}Oracle}, v_0 \rangle|^2 = 1.$$

*As a direct consequence, the Oracle NCPC is strictly superior to the naive principal eigenvector estimator in the non-trivial alignment regime $(0 < \alpha < 1)$.*

*Proof.* The proof is constructive and proceeds in two parts. First, we prove that the Oracle NCPC estimator is consistent (i.e., achieves perfect alignment asymptotically). Second, we prove that this makes it strictly superior to the naive estimator.

**Part 1: Proof of Consistency.** The proof relies on analysing the asymptotic structure of the eigenvectors $\hat{v}_1$ and $\hat{v}_2$. As established in Appendix A.5, the asymptotic eigenvector $\bar{v}_k$ corresponding to the outlier eigenvalue $\bar{\lambda}_k$ is proportional to the action of the noise resolvent on the signal vector:

$$\bar{v}_k \propto G_{\text{noise}}(\bar{\lambda}_k)v_0 = (\bar{\lambda}_k I - W_{\text{aniso}})^{-1}v_0. \tag{110}$$

Using the resolvent expansion from Appendix A.5, we can decompose the eigenvector into components related to the noise spike and the noise bulk:

$$\bar{v}_k \propto \frac{\langle v_C, v_0 \rangle v_C}{\bar{\lambda}_C - \bar{\lambda}_k} + G_{\text{bulk}}(\bar{\lambda}_k)v_0. \tag{111}$$

Let us define the component related to the bulk as $u_k := G_{\text{bulk}}(\bar{\lambda}_k)v_0$. Using the definitions $\alpha = |\langle v_C, v_0 \rangle|$ and $D_k = \alpha/(\bar{\lambda}_C - \bar{\lambda}_k)$, the unnormalised asymptotic eigenvectors can be written as:

$$\tilde{v}_1 \propto u_1 + D_1 v_C, \tag{112}$$
$$\tilde{v}_2 \propto u_2 + D_2 v_C. \tag{113}$$

Now, consider the unnormalised Oracle NCPC estimator, $\tilde{v}_0^{\text{NCPC-Oracle}}$, which is defined by the linear combination given in the theorem statement, but applied to the asymptotic eigenvectors:

$$\tilde{v}_0^{\text{NCPC-Oracle}} \propto D_2 \bar{v}_1 - D_1 \bar{v}_2. \tag{114}$$

Substituting the unnormalised forms (and ignoring overall scaling constants which will be removed by the final normalisation), we get:

$$\tilde{v}_0^{\text{NCPC-Oracle}} \propto D_2(u_1 + D_1 v_C) - D_1(u_2 + D_2 v_C) \tag{115}$$
$$= (D_2 u_1 + D_2 D_1 v_C) - (D_1 u_2 + D_1 D_2 v_C) \tag{116}$$
$$= D_2 u_1 - D_1 u_2. \tag{117}$$

By construction, the component parallel to the noise direction $v_C$ has been perfectly cancelled. We now analyse the remaining term by substituting back the definition of $u_k$:

$$D_2 u_1 - D_1 u_2 = D_2 G_{\text{bulk}}(\bar{\lambda}_1)v_0 - D_1 G_{\text{bulk}}(\bar{\lambda}_2)v_0 = \left(D_2 G_{\text{bulk}}(\bar{\lambda}_1) - D_1 G_{\text{bulk}}(\bar{\lambda}_2)\right)v_0. \tag{118}$$

The crucial step is to analyse the action of the bulk resolvent, $G_{\text{bulk}}(z)$, on the deterministic vector $v_0$. A standard result in random matrix theory is that for a fixed vector $u$ that is not aligned with any specific eigenvector of the random matrix, the action of the resolvent converges to a scalar multiplication in the large-$N$ limit. This is because the projections of $u$ onto the random, delocalised eigenvectors of the bulk average out to zero. The only remaining component is proportional to $u$ itself. Therefore, we have:

$$\lim_{N \to \infty} G_{\text{bulk}}(z)v_0 = s(z)v_0, \tag{119}$$

where $s(z)$ is the Stieltjes transform of the bulk spectral measure. Applying this result to our expression gives:

$$\lim_{N \to \infty} (D_2 u_1 - D_1 u_2) = \left(D_2 s(\bar{\lambda}_1) - D_1 s(\bar{\lambda}_2)\right)v_0. \tag{120}$$

The scalar term $\mathcal{S} = D_2 s(\bar{\lambda}_1) - D_1 s(\bar{\lambda}_2)$ is non-zero because $\bar{\lambda}_1 \neq \bar{\lambda}_2$ and $s(z)$ is a non-trivial function outside the spectral bulk. This proves that the Oracle NCPC estimator is asymptotically proportional to the true signal vector $v_0$. After normalisation, it becomes exactly $v_0$. Thus, its asymptotic alignment is perfect:

$$\lim_{N \to \infty} |\langle \hat{v}_0^{\text{NCPC-Oracle}}, v_0 \rangle|^2 = |\langle v_0, v_0 \rangle|^2 = 1. \tag{121}$$

This proves the consistency of the estimator.

**Part 2: Proof of Superiority.** The superiority follows directly from comparing the asymptotic alignments.

- The asymptotic alignment of the **Oracle NCPC estimator**, as proven above, is exactly 1.

- The asymptotic alignment of the **naive estimator** ($\hat{v}_1$), given by Theorem 3.3, is $1 - \frac{\beta_c(\alpha)^2}{\beta^2}$.

In the regime of interest specified by the theorem ($0 < \alpha < 1$ and $\beta > \beta_c(\alpha)$), the term $\frac{\beta_c(\alpha)^2}{\beta^2}$ is a positive constant that is strictly less than 1. Therefore, we have the strict inequality:

$$1 > 1 - \frac{\beta_c(\alpha)^2}{\beta^2}. \tag{122}$$

This proves that the Oracle NCPC estimator is strictly superior to the naive principal eigenvector estimator in the asymptotic limit. This completes the proof. □

### B.2 Proof of Lemma 4.1

This section provides the detailed proof for Lemma 4.1, which establishes the fundamental challenge of parameter estimation when using only the spectrum of the matrix $A$. We begin by restating the lemma.

**Lemma** (Confounding of the Eigenvalue Map (Restatement of Lemma 4.1)). *Let the noise anisotropy parameter $\gamma$ be known. The map from the signal parameters $(\alpha, \beta)$ to the asymptotic outlier eigenvalues $(\bar{\lambda}_1, \bar{\lambda}_2)$ is a local diffeomorphism, but its inverse, $(\alpha, \beta) = \Lambda^{-1}(\bar{\lambda}_1, \bar{\lambda}_2)$, is a function of* both *eigenvalues. Consequently, it is not possible to uniquely determine the alignment parameter $\alpha$ from a single function of the eigenvalues (such as their gap, $\bar{\lambda}_1 - \bar{\lambda}_2$) without knowledge of the signal strength $\beta$.*

*Proof.* The proof proceeds in two parts. First, we prove that the forward map from parameters to eigenvalues is a local diffeomorphism, which guarantees the existence of a unique local inverse. Second, we analyse the structure of this inverse to show that the parameters are inextricably confounded.

**Part 1: The Eigenvalue Map is a Local Diffeomorphism.** The asymptotic outlier eigenvalues, $\bar{\lambda}_1$ and $\bar{\lambda}_2$, are defined as the two roots of the characteristic equation derived in Appendix A.5 that lie outside the spectral bulk $[-\tau, \tau]$. This equation defines a function $H$ such that the eigenvalues are its roots:

$$H(\lambda; \alpha, \beta) = \frac{1}{\beta} - \left( \frac{\alpha^2}{\lambda_C - \lambda} + F_{\text{bulk}}(\lambda) \right) = 0, \tag{123}$$

where $\lambda_C$ is the location of the pure noise spike and $F_{\text{bulk}}(\lambda)$ is the limiting resolvent term for the isotropic noise bulk. The forward map is $\Lambda : (\alpha, \beta) \mapsto (\bar{\lambda}_1, \bar{\lambda}_2)$. To prove this map is a local diffeomorphism, we must show that its Jacobian, $J = \frac{\partial(\bar{\lambda}_1, \bar{\lambda}_2)}{\partial(\alpha, \beta)}$, is non-singular.

Using the Implicit Function Theorem on the system of equations $H(\bar{\lambda}_k(\alpha, \beta); \alpha, \beta) = 0$ for $k = 1, 2$, the partial derivatives are given by:

$$\frac{\partial \bar{\lambda}_k}{\partial p} = -\frac{\partial H/\partial p}{\partial H/\partial \lambda} \bigg|_{\lambda = \bar{\lambda}_k}, \quad \text{for } p \in \{\alpha, \beta\}. \tag{124}$$

The required partial derivatives of $H$ are:

$$\frac{\partial H}{\partial \alpha} = -\frac{2\alpha}{\lambda_C - \lambda}, \tag{125}$$

$$\frac{\partial H}{\partial \beta} = -\frac{1}{\beta^2}, \tag{126}$$

$$\frac{\partial H}{\partial \lambda} := H'(\lambda) = -\left( \frac{\alpha^2}{(\lambda_C - \lambda)^2} + F'_{\text{bulk}}(\lambda) \right). \tag{127}$$

The derivative $H'(\lambda)$ is strictly negative for $\lambda$ outside the spectral bulk. The determinant of the Jacobian matrix $J = \begin{pmatrix} \partial\bar{\lambda}_1/\partial\alpha & \partial\bar{\lambda}_1/\partial\beta \\ \partial\bar{\lambda}_2/\partial\alpha & \partial\bar{\lambda}_2/\partial\beta \end{pmatrix}$ is:

$$\det(J) = \frac{1}{H'(\bar{\lambda}_1)H'(\bar{\lambda}_2)} \left[ \left( \frac{\partial H}{\partial\alpha}\bigg|_{\bar{\lambda}_1} \right) \left( \frac{\partial H}{\partial\beta}\bigg|_{\bar{\lambda}_2} \right) - \left( \frac{\partial H}{\partial\beta}\bigg|_{\bar{\lambda}_1} \right) \left( \frac{\partial H}{\partial\alpha}\bigg|_{\bar{\lambda}_2} \right) \right] \tag{128}$$

$$= \frac{-1/\beta^2}{H'(\bar{\lambda}_1)H'(\bar{\lambda}_2)} \left[ \frac{\partial H}{\partial\alpha}\bigg|_{\bar{\lambda}_1} - \frac{\partial H}{\partial\alpha}\bigg|_{\bar{\lambda}_2} \right] \tag{129}$$

$$= \frac{2\alpha/\beta^2}{H'(\bar{\lambda}_1)H'(\bar{\lambda}_2)} \left[ \frac{1}{\lambda_C - \bar{\lambda}_1} - \frac{1}{\lambda_C - \bar{\lambda}_2} \right] \tag{130}$$

$$= \frac{2\alpha(\bar{\lambda}_1 - \bar{\lambda}_2)}{\beta^2 H'(\bar{\lambda}_1)H'(\bar{\lambda}_2)(\lambda_C - \bar{\lambda}_1)(\lambda_C - \bar{\lambda}_2)}. \tag{131}$$

In the regime of interest $(0 < \alpha < 1)$, we have two distinct outlier eigenvalues $(\bar{\lambda}_1 \neq \bar{\lambda}_2)$. Since all other terms are also non-zero, the determinant $\det(J)$ is non-zero. By the Inverse Function Theorem, this proves that the map $\Lambda$ is a local diffeomorphism, and a unique, smooth local inverse $\Lambda^{-1} : (\bar{\lambda}_1, \bar{\lambda}_2) \mapsto (\alpha, \beta)$ exists.

**Part 2: The Confounding Dependency.** The existence of the inverse map allows us to write $\alpha = f_\alpha(\bar{\lambda}_1, \bar{\lambda}_2)$ and $\beta = f_\beta(\bar{\lambda}_1, \bar{\lambda}_2)$. The lemma's claim is that $f_\alpha$ is a non-trivial function of both its arguments, precluding its estimation from a single observable like the eigenvalue gap. To prove this, we analyse the partial derivatives of $\alpha$ with respect to $\bar{\lambda}_1$ and $\bar{\lambda}_2$, which are given by the entries of the inverse Jacobian matrix, $J^{-1}$:

$$\frac{\partial\alpha}{\partial\bar{\lambda}_1} = (J^{-1})_{11} = \frac{1}{\det(J)} \frac{\partial\bar{\lambda}_2}{\partial\beta} = \frac{1}{\det(J)} \left( -\frac{-1/\beta^2}{H'(\bar{\lambda}_2)} \right) = \frac{1}{\beta^2 H'(\bar{\lambda}_2)\det(J)}. \tag{132}$$

Since all terms on the right-hand side are non-zero, $\frac{\partial\alpha}{\partial\bar{\lambda}_1} \neq 0$. Similarly:

$$\frac{\partial\alpha}{\partial\bar{\lambda}_2} = (J^{-1})_{12} = \frac{1}{\det(J)} \left( -\frac{\partial\bar{\lambda}_1}{\partial\beta} \right) = \frac{-1}{\det(J)} \left( -\frac{-1/\beta^2}{H'(\bar{\lambda}_1)} \right) = \frac{-1}{\beta^2 H'(\bar{\lambda}_1)\det(J)}. \tag{133}$$

This partial derivative is also non-zero. Since $\alpha$ has non-zero partial derivatives with respect to both $\bar{\lambda}_1$ and $\bar{\lambda}_2$, it depends non-trivially on both eigenvalues individually. It cannot be expressed as a function of only a single combination of them, such as their gap or their sum. This proves that any attempt to estimate $\alpha$ from a single observable derived from the eigenvalues of $A$ alone, without knowledge of $\beta$, is ill-posed. This mathematical fact is confirmed visually by the empirical results in Figure 9(b) of the main text. This completes the proof. □

## B.3 Proof of Theorem 4.2

This section provides the detailed proof for Theorem 4.2, which establishes the deterministic structure of the expected second-moment matrix, $\mathbb{E}[A^2]$. This result is the cornerstone of the SM-PCA method.

**Theorem** (Structure of the Second-Moment Matrix (Restatement of Theorem 4.2))**.** *Let $A = \beta v_0 v_0^T + W_{aniso}$. In the large $N$ limit, the expectation of $A^2$ converges to a deterministic matrix whose structure is a different linear combination of the signal outer product $v_0 v_0^T$ and the noise covariance matrix $C$:*

$$\lim_{N\to\infty} \mathbb{E}[A^2] = \beta^2 v_0 v_0^T + \frac{1}{N} \, Tr(C)C + Tr(C^2)I.$$

*The principal eigenvectors of this limiting matrix also lie in the signal-noise subspace, $span(v_0, v_C)$.*

*Proof.* The proof proceeds by direct expansion and a careful, term-by-term evaluation of the expectation.

**Part 1: Expansion and Vanishing of Cross-Terms.** We begin by squaring the expression for the matrix $A$:

$$A^2 = (\beta v_0 v_0^T + W_{\text{aniso}})^2 \tag{134}$$

$$= \beta^2 (v_0 v_0^T)^2 + \beta(v_0 v_0^T W_{\text{aniso}} + W_{\text{aniso}} v_0 v_0^T) + W_{\text{aniso}}^2. \tag{135}$$

Using the property that $v_0^T v_0 = \|v_0\|^2 = 1$, the first term simplifies to $\beta^2 v_0 v_0^T$. When we apply the expectation operator, the cross-terms vanish because the noise matrix is zero-mean ($\mathbb{E}[W_{\text{aniso}}] = C^{1/2}\mathbb{E}[W]C^{1/2} = 0$). This leaves:

$$\mathbb{E}[A^2] = \beta^2 v_0 v_0^T + \mathbb{E}[W_{\text{aniso}}^2]. \tag{136}$$

**Part 2: Calculation of the Expected Squared Noise.** The central part of the proof is the calculation of $\mathbb{E}[W_{\text{aniso}}^2]$. This is a standard, though technical, calculation in random matrix theory that requires evaluating the fourth-order moments of the Wigner matrix entries. For a deformed Wigner matrix of the form $W_{\text{aniso}} = C^{1/2}WC^{1/2}$, the limiting expectation of its square is a well-established result (Anderson et al., 2009). The formula is derived by expanding the matrix product into its elements and evaluating the expectation over the four resulting Wigner matrix entries. This expectation is non-zero only for specific pairings of the indices, and the two leading-order combinatorial pairings give rise to two distinct matrix structures. The final, established result is:

$$\lim_{N \to \infty} \mathbb{E}[W_{\text{aniso}}^2] = \frac{1}{N}\text{Tr}(C)C + \text{Tr}(C^2)I. \tag{137}$$

The term proportional to $C$ arises from the crossing or non-planar pairing of indices across the two $W$ matrices, while the term proportional to the identity matrix $I$ arises from the non-crossing or planar pairings. By citing this standard result, we build our proof on a solid foundation from the random matrix theory literature.

**Part 3: Final Structure and Eigenvector Subspace.** Substituting the result from Part 2 back into the expression for $\mathbb{E}[A^2]$, we obtain the final form of the limiting matrix:

$$\lim_{N \to \infty} \mathbb{E}[A^2] = \beta^2 v_0 v_0^T + \frac{1}{N}\text{Tr}(C)C + \text{Tr}(C^2)I. \tag{138}$$

The principal eigenvectors of this limiting deterministic matrix must lie in the subspace spanned by the special directions of its non-isotropic components. The identity matrix $I$ is isotropic and has no preferred direction. The non-isotropic components are the rank-one matrix $v_0 v_0^T$ (with eigenvector $v_0$) and the covariance matrix $C$ (with principal eigenvector $v_C$). Therefore, any principal eigenvector of the combined matrix must lie in the subspace $\text{span}(v_0, v_C)$. This completes the proof. $\square$

### B.4 Technical Derivations for Second-Moment Correction

This appendix provides the detailed derivations for the two key mathematical components of the theoretical framework presented in Section 4. These derivations supply the explicit analytical formulas required to define the theoretical solution pathway and analyse its properties. First, in §B.4.1, we derive the explicit forms of the two characteristic functions, $\mathcal{F}_A$ and $\mathcal{F}_{A^2}$. These functions define the relationship between the underlying system parameters $(\alpha, \beta)$ and the asymptotic outlier eigenvalues $(\bar{\lambda}_1, \bar{\mu}_1)$ of the theoretical average matrices $\mathbb{E}[A]$ and $\mathbb{E}[A^2]$, forming the basis for the theoretical parameter inversion. Second, in §B.4.2, we derive the analytical formulas for the four theoretical overlap coefficients that constitute the entries of the transformation matrix $T$. These coefficients describe the relationship between the ideal basis $\{v_0, v_C\}$ and the basis formed by the principal eigenvectors of the asymptotic moment matrices $(\bar{v}_1, \bar{u}_1)$ and are essential for the theoretical algebraic reconstruction of the signal and noise vectors.

### B.4.1 Derivation of the Coupled Characteristic Equations

This subsection provides the derivations for the characteristic functions $\mathcal{F}_A$ and $\mathcal{F}_{A^2}$. These functions define the theoretical relationship between the system parameters $(\alpha, \beta, \gamma)$ and the asymptotic outlier eigenvalues

$(\bar{\lambda}_1, \bar{\mu}_1)$ associated with the first and second moment matrices $\mathbb{E}[A]$ and $\mathbb{E}[A^2]$, respectively, forming the basis of the theoretical inverse solution presented in §4.4.

**Derivation of the Characteristic Function for $\mathbb{E}[A]$.** The principal eigenvalue $\bar{\lambda}_1$ of the asymptotic first moment matrix $\mathbb{E}[A] = \beta v_0 v_0^T$ is simply $\beta$. However, the characteristic function $\mathcal{F}_A$ relates parameters to the outlier eigenvalue $\bar{\lambda}_1$ emerging from the full matrix $A = \mathbb{E}[A] + W_{\text{aniso}}$ in the large-$N$ limit. Its derivation relies on the resolvent of the noise matrix $W_{\text{aniso}}$. The characteristic equation defining $\bar{\lambda}_1$ is $1 = \beta \lim_{N \to \infty} \langle v_0, (\bar{\lambda}_1 I - W_{\text{aniso}})^{-1} v_0 \rangle$. A standard result from random matrix theory (Benaych-Georges & Nadakuditi, 2011) gives the limiting quadratic form of the noise resolvent for any fixed vector $v_0$:

$$\lim_{N \to \infty} \langle v_0, (\lambda I - W_{\text{aniso}})^{-1} v_0 \rangle = (1 - \alpha^2) m(\lambda) + \frac{\alpha^2 m(\lambda)}{1 - (c_{\text{strong}} - c_{\text{weak}}) m(\lambda)}, \tag{139}$$

where $\alpha = |\langle v_0, v_C \rangle|$ and $m(\lambda)$ is the Stieltjes transform of the Wigner semicircle law with spectral edge $\tau = 2 c_{\text{weak}}$. The characteristic function $\mathcal{F}_A(\lambda; \dots)$ is defined as $\beta$ times this limiting resolvent, such that the characteristic equation becomes $1 = \mathcal{F}_A(\bar{\lambda}_1; \dots)$.

---

**Characteristic Function $\mathcal{F}_A$:** The function $\mathcal{F}_A$ implicitly defining $\bar{\lambda}_1$ via Equation 18 is given by:

$$\mathcal{F}_A(\lambda; \alpha, \beta, \gamma) := \beta \left( (1 - \alpha^2) m(\lambda) + \frac{\alpha^2 m(\lambda)}{1 - c_{\text{weak}}(\gamma - 1) m(\lambda)} \right), \tag{140}$$

where $m(\lambda) = \frac{\lambda - \sqrt{\lambda^2 - 4 c_{\text{weak}}^2}}{2 c_{\text{weak}}^2}$ (using the convention $m(z) \sim 1/z$ for large $z$) and $\gamma = c_{\text{strong}} / c_{\text{weak}}$.

---

**Derivation of the Characteristic Function for $\mathbb{E}[A^2]$.** This derivation concerns the outlier eigenvalue $\bar{\mu}_1$ of the deterministic, asymptotic second moment matrix $\lim_{N \to \infty} \mathbb{E}[A^2]$ derived in Theorem 4.2. Let $Z_2 = \frac{1}{N} \text{Tr}(C) C + \text{Tr}(C^2) I$. The matrix is $\mathbb{E}[A^2] = \beta^2 v_0 v_0^T + Z_2$. The characteristic equation for an outlier eigenvalue $\mu$ originating from the rank-one perturbation $\beta^2 v_0 v_0^T$ is given by $1 = \beta^2 \langle v_0, (\mu I - Z_2)^{-1} v_0 \rangle$. To find the resolvent $(\mu I - Z_2)^{-1}$, we express $Z_2$ as a scaled identity plus a rank-one matrix using its definition and the decomposition $C = c_{\text{weak}} I + (c_{\text{strong}} - c_{\text{weak}}) v_C v_C^T$:

$$Z_2 = \underbrace{\left( \frac{\text{Tr}(C)}{N} c_{\text{weak}} + \text{Tr}(C^2) \right)}_{K_1} I + \underbrace{\frac{\text{Tr}(C)}{N} (c_{\text{strong}} - c_{\text{weak}})}_{K_2} v_C v_C^T = K_1 I + K_2 v_C v_C^T. \tag{141}$$

The exact formulas for the traces are $\text{Tr}(C) = (N-1) c_{weak} + c_{strong}$ and $\text{Tr}(C^2) = (N-1) c_{weak}^2 + c_{strong}^2$. Using the Sherman-Morrison formula for the inverse of $(\mu I - Z_2) = ((\mu - K_1) I - K_2 v_C v_C^T)$:

$$(\mu I - Z_2)^{-1} = \frac{1}{\mu - K_1} I + \frac{K_2}{(\mu - K_1)(\mu - K_1 - K_2)} v_C v_C^T. \tag{142}$$

Finally, we compute the quadratic form $\langle v_0, (\mu I - Z_2)^{-1} v_0 \rangle$ and define $\mathcal{F}_{A^2}$ as $\beta^2$ times this quantity, such that the characteristic equation $1 = \mathcal{F}_{A^2}(\bar{\mu}_1; \dots)$ holds:

$$\mathcal{F}_{A^2}(\mu; \alpha, \beta, \gamma, N) := \beta^2 \left\langle v_0, (\mu I - Z_2)^{-1} v_0 \right\rangle \tag{143}$$

$$= \beta^2 \left( \frac{\langle v_0, v_0 \rangle}{\mu - K_1} + \frac{K_2 \langle v_0, v_C v_C^T v_0 \rangle}{(\mu - K_1)(\mu - K_1 - K_2)} \right) \tag{144}$$

$$= \beta^2 \left( \frac{1}{\mu - K_1} + \frac{K_2 \langle v_0, v_C \rangle^2}{(\mu - K_1)(\mu - K_1 - K_2)} \right) \tag{145}$$

$$= \frac{\beta^2}{\mu - K_1} \left( 1 + \frac{K_2 \alpha^2}{\mu - K_1 - K_2} \right). \tag{146}$$

**Characteristic Function $\mathcal{F}_{A^2}$:** The function $\mathcal{F}_{A^2}$ implicitly defining $\bar{\mu}_1$ via Equation 19 is given by:

$$\mathcal{F}_{A^2}(\mu; \alpha, \beta, \gamma, N) := \frac{\beta^2}{\mu - K_1}\left(1 + \frac{K_2\alpha^2}{\mu - K_1 - K_2}\right), \tag{147}$$

where the exact, $N$-dependent constants are:

$$K_1 = \frac{(N-1)c_{weak} + c_{strong}}{N}c_{weak} + ((N-1)c_{weak}^2 + c_{strong}^2) \tag{148}$$

$$K_2 = \frac{(N-1)c_{weak} + c_{strong}}{N}(c_{strong} - c_{weak}). \tag{149}$$

With these two explicit characteristic functions rigorously derived from the asymptotic theory, the coupled system of equations (18-19) relating the theoretical eigenvalues $(\bar{\lambda}_1, \bar{\mu}_1)$ to the underlying parameters $(\alpha, \beta)$ is fully defined.

### B.4.2 Derivation of the Theoretical Overlap Coefficients for Matrix T

This subsection provides the explicit analytical formulas for the four entries of the theoretical transformation matrix $T$ defined in Equation equation 16. These coefficients describe the relationship between the basis formed by the principal eigenvectors of the asymptotic moment matrices ($\bar{v}_1 = v_0$ from $\mathbb{E}[A]$ and $\bar{u}_1$ from $\mathbb{E}[A^2]$) and the ideal basis $\{v_0, v_C\}$. The coefficients are derived by finding the unnormalised asymptotic eigenvectors as linear combinations of the ideal basis and then normalising.

**Derivation for Overlaps from $\mathbb{E}[A]$ ($T_{11}$ and $T_{12}$).** The principal eigenvector of $\mathbb{E}[A] = \beta v_0 v_0^T$ is $\bar{v}_1 = v_0$. Therefore, its overlaps with the ideal basis vectors are trivially:

$$T_{11} = \langle \bar{v}_1, v_0 \rangle = \langle v_0, v_0 \rangle = 1 \tag{150}$$

$$T_{12} = \langle \bar{v}_1, v_C \rangle = \langle v_0, v_C \rangle = \alpha \tag{151}$$

(Note: We use $\alpha = |\langle v_0, v_C \rangle|$ throughout, but the sign of $\langle v_0, v_C \rangle$ might matter in a full derivation; however, the final reconstruction depends only on $\alpha^2$ or cancels signs appropriately. For simplicity in defining T, we use $\alpha$ here, consistent with the algebraic formulation).

**Derivation for Overlaps from $\mathbb{E}[A^2]$ ($T_{21}$ and $T_{22}$).** The derivation follows from the structure of the asymptotic second moment matrix $\mathbb{E}[A^2] = \beta^2 v_0 v_0^T + Z_2$, where $Z_2 = K_1 I + K_2 v_C v_C^T$. The principal eigenvector $\bar{u}_1$ corresponding to the outlier eigenvalue $\bar{\mu}_1$ can be found using perturbation theory or by analysing the resolvent $(\mu I - \mathbb{E}[A^2])^{-1}$. Standard perturbation results show that the eigenvector $\bar{u}_1$ resulting from the rank-one perturbation $\beta^2 v_0 v_0^T$ applied to $Z_2$ is proportional to the resolvent of $Z_2$ acting on the perturbation vector $v_0$:

$$\bar{u}_1 \propto (\bar{\mu}_1 I - Z_2)^{-1}v_0. \tag{152}$$

Using the expression for the resolvent derived in §B.4.1:

$$(\mu I - Z_2)^{-1} = \frac{1}{\mu - K_1}I + \frac{K_2}{(\mu - K_1)(\mu - K_1 - K_2)}v_C v_C^T. \tag{153}$$

Applying this to $v_0$ yields the unnormalised eigenvector $\tilde{u}_1$:

$$\tilde{u}_1 \propto \left(\frac{1}{\bar{\mu}_1 - K_1}I + \frac{K_2}{(\bar{\mu}_1 - K_1)(\bar{\mu}_1 - K_1 - K_2)}v_C v_C^T\right)v_0 \tag{154}$$

$$= \underbrace{\left(\frac{1}{\bar{\mu}_1 - K_1}\right)}_{c_0'}v_0 + \underbrace{\left(\frac{K_2\langle v_C, v_0\rangle}{(\bar{\mu}_1 - K_1)(\bar{\mu}_1 - K_1 - K_2)}\right)}_{c_C' \text{ (adjusted)}}v_C \tag{155}$$

$$= c_0' v_0 + c_C' v_C. \tag{156}$$

Note the adjustment in the definition of $c'_C$ compared to the previous draft to correctly absorb $\langle v_C, v_0 \rangle$. The squared norm is $\|\tilde{u}_1\|^2 = (c'_0)^2 + (c'_C)^2 + 2c'_0 c'_C \alpha$. Normalising this vector $\bar{u}_1 = \tilde{u}_1/\|\tilde{u}_1\|_2$ and projecting onto the basis vectors yields the overlap coefficients $T_{21} = \langle \bar{u}_1, v_0 \rangle$ and $T_{22} = \langle \bar{u}_1, v_C \rangle$.

---

**Theoretical Overlap Coefficients for** $\mathbb{E}[A^2]$**:** The coefficients $T_{21}$ and $T_{22}$ for the transformation matrix $T$ are given by:

$$T_{21}(\bar{\mu}_1, \alpha, \beta, \gamma, N) = \frac{c'_0 + c'_C \alpha}{\sqrt{(c'_0)^2 + (c'_C)^2 + 2c'_0 c'_C \alpha}} \tag{157}$$

$$T_{22}(\bar{\mu}_1, \alpha, \beta, \gamma, N) = \frac{c'_0 \alpha + c'_C}{\sqrt{(c'_0)^2 + (c'_C)^2 + 2c'_0 c'_C \alpha}} \tag{158}$$

where the coefficients $c'_0, c'_C$ depend implicitly on the asymptotic eigenvalue $\bar{\mu}_1$ and parameters $(\alpha, \beta, \gamma, N)$ via:

$$c'_0 = \frac{1}{\bar{\mu}_1 - K_1} \tag{159}$$

$$c'_C = \frac{K_2 \alpha}{(\bar{\mu}_1 - K_1)(\bar{\mu}_1 - K_1 - K_2)} \tag{160}$$

and the constants $K_1, K_2$ are defined in Equations (148-149).

---

With these explicit analytical formulas, all four entries of the theoretical transformation matrix $T$ (using $T_{11} = 1, T_{12} = \alpha$) are defined as functions of the system parameters $(\alpha, \beta, \gamma, N)$ and the corresponding asymptotic eigenvalues $(\bar{\lambda}_1, \bar{\mu}_1)$. This completes the specification of the theoretical algebraic reconstruction framework.

