# OpenReview forum: "Eigenvector Phase Transitions under Anisotropic Noise"
_TMLR — Rejected by TMLR_

### Review · Reviewer_YKri · 2025-09-15

**Summary Of Contributions:**

This paper considers Wigner matrices with both multiplicative and additive deformations.
Specifically, the model is as follows.
Let $W$ be a Wigner matrix and $v_0,v$ be unit vectors.
Let $$ A = \beta v_0 v_0^\top + C^{1/2} W C^{1/2} $$ where $ C = c_s v v^\top + c_w(I - vv^\top) $ (I use $ v \equiv v_C $, $ c_s \equiv c_{\textnormal{strong}} $, $ c_w \equiv c_{\textnormal{weak}} $ for notational brevity).
Using the fact that for any unit vector $a$, $(I + c aa^\top)^{1/2} = I + (\sqrt{1+c}-1)aa^\top$, one can write explicitly $ C^{1/2} = \sqrt{c_w} + (\sqrt{c_s} - \sqrt{c_w}) vv^\top $.
Then letting $ d = \sqrt{c_s} - \sqrt{c_w} $, we can write the model as $$ A = \beta v_0v_0^\top + c_w W + d^2 (v^\top W v) vv^\top + \sqrt{c_w} d (vv^\top W + W vv^\top) .$$
From here, detailed analysis can be made using standard tools in random matrix theory.
Specific results include:
- ESD of $A$ equals semicircle of radius $c_w$ (this is a direct consequence of the explicit expression of $A$ above)
- threshold $\beta_c$ above which an outlier eigenvalue emerges
- the limiting value of $|\langle \hat{v}_1, v_0 \rangle|$ whenever $ \beta > \beta_c $
- size of Gaussian fluctuation of $|\langle \hat{v}_1, v_0 \rangle|$

where $\hat{v}_i$ is the $i$-th largest eigenvector of $A$.

**Audience:**

Yes

**Audience Explanation:**

Spiked random matrices are of general interests to TMLR audience.

**Claims And Evidence:**

Yes

**Claims Explanation:**

All results come with proofs (though not entirely at a mathematically rigorous level; see comments below), and some are corroborated with numerical simulations and/or are applied to real data.

**Requested Changes:**

- The authors may want to define "detection" formally. I think this paper considers *strong detection*, meaning probability of error converges to $0$. If we only require probability of error bounded away from $1/2$ (doing better than random guess), then the presence of outlying eigenvalue is not necessary. At least in standard spiked Wigner model, weak detection is possible for *every* positive $\beta$.

- Theorem 4, what do you mean by "align"? Do you mean $| \langle \hat{v}_2, v \rangle |\to1$? Inspecting the proof in Appendix A.4 doesn't help. It seems towards the end of the proof, the $\beta\to\infty$ limit is taken which was not assumed in the theorem statement.

- Theorem 5, define $f(\beta,\alpha)$ in formula. Move the definition of $ \sigma^2(\beta, \alpha) $ to the main text.

- The authors derived the size of Gaussian fluctuation of $ |\langle \hat{v}_1, v_0 \rangle| $. Does it make sense to also derive the fluctuation of $ |\langle \hat{v}_2, v \rangle| $? From the explicitly expression of $A$, both $v_0,v$ can be viewed as additive deformations and therefore have comparable roles.

- The level of rigor is generally low. The approximate equality $\approx$ is frequently used whose precise meaning is unclear. There are multiple instances of implicit interchange of limits without justification. There is general confusion between $N$-dependent quantities and their $N\to\infty$ limits.

- Section 5.3, the authors said that *Gaussian* Wigner ensemble is considered in this paper. Is Gaussianity needed? This assumption doesn't seem to be stated in the main text and in principle the tools employed by this paper shouldn't require it.

- I found the symmetric model generally less motivated than the rectangular model in practice. In this regard, the authors may be interested in checking out https://arxiv.org/abs/2207.03466 , https://arxiv.org/abs/2405.13912 to name a few.

---

> ### Author Response · Authors · 2025-10-22
>
> Dear Reviewer YKri,
>
> Thank you very much for your detailed review. We've **substantially revised** the manuscript to address concerns on definitions, assumptions, and rigor.
>
> **Note:** All revisions and new content are highlighted in **blue** text in the updated PDF.
>
> We address your points in detail below:
>
> ---
>
> ## Comment 1 (Strong vs. Weak Detection):
>
> > The authors may want to define "detection" formally... weak detection is possible...
>
> **Response:**
> You're correct; our analysis ($\beta_c(\alpha)$ in Thm 3.2) corresponds to **strong detection**.
>
> We clarified this:
> 1. $\beta_c(\alpha)$ is explicitly the threshold for **strong detection** in the **Abstract**, **Introduction (§1)**, and **Interpretation paragraph of §3.2**.
> 2. **Weak detection** is now discussed in **§3.2**, acknowledging below-$\beta_c(\alpha)$ performance.
> 3. **New experimental validation (Figure 4, §5.1)** using Excess Energy demonstrates the strong detection limit at $\beta_c(0)$ and below-$\beta_c$ weak detection performance.
>
> ---
>
> ## Comment 2 (Clarify "align" for v2):
>
> > Theorem 4 [now Theorem 3.5], what do you mean by "align"? Do you mean $|\langle \hat{v}_2, v_C \rangle|^2 \to \text{const} > 0$? Inspecting the proof... limit is taken...
>
> **Response:**
> We refined **Theorem 3.5**: "aligns" means $|\langle \hat{v}_2, v_C \rangle|^2$ converges to a non-trivial limit in the super-critical phase ($\beta > \beta_c(\alpha)$).
>
> We added **new Theorem 3.7 (CLT for the Second Eigenvector)** for full statistical characterization. It formally defines asymptotic mean alignment $g(\beta, \alpha, \gamma)$ and its fluctuations.
>
> Thm 3.5 proof (Appendix A.5) establishes outlier structure. By orthogonality, $\hat{v}_2$ must align with the $v_C$ component orthogonal to $v_0$. Thm 3.7 (Appendix A.7) provides rigorous fluctuation treatment.
>
> ---
>
> ## Comment 3 (Define $\sigma^2$ in main text):
>
> > Theorem 5 [now Theorem 3.6], define $\sigma^2(\beta, \alpha)$ in formula. Move the definition of $f(\beta, \alpha)$ to the main text.
>
> **Response:**
> 1. **$\sigma^2(\beta, \alpha)$:** **Theorem 3.6** points to **Equation (99)** in Appendix A.6, where the complex variance function is given. Referencing the appendix is clearer.
> 2. **$f(\beta, \alpha)$:** We revised **Theorem 3.3 (Asymptotic Alignment)** to explicitly define $f(\beta, \alpha)$ for the asymptotic mean alignment: $f(\beta, \alpha) := \lim_{N \to \infty} |\langle \hat{v}_1, v_0 \rangle|^2 = 1 - \frac{\beta_c(\alpha)^2}{\beta^2}$. This is now directly in the main text.
>
> ---
>
> ## Comment 4 (CLT for $v_2$):
>
> > Does it make sense to also derive the fluctuation of $|\langle \hat{v}_2, v_C \rangle|^2$?
>
> **Response:**
> Excellent suggestion. We added **Theorem 3.7 (CLT for the Second Eigenvector)** in **Section 3.6**. It provides the asymptotic distribution for the fluctuations of $|\langle \hat{v}_2, v_C \rangle|^2$. Proof's in **Appendix A.7**. **Experimental validation** is included (Figure 7, §5.1). This completes the statistical characterization.
>
> ---
>
> ## Comment 5 (Rigor):
>
> > The level of rigor is generally low... $\approx$ is frequently used... implicit interchange of limits... confusion between $N$-dependent quantities...
>
> **Response:**
> We carefully revised the appendices:
> 1. **Introductory note in Appendix A** clarifies notation ($\bar{X}$ for $N \to \infty$ limits) and defines $o_p(\cdot)$.
> 2. `$\approx$` in fluctuation proofs (Appendices A.6/A.7) systematically replaced with equalities using explicit `$o_p(N^{-1/2})$` terms.
> 3. `$\propto$` (Appendix A.5) replaced with explicit definitions of unnormalized vectors ($\tilde{v}_k$).
> 4. Language on approximations refined.
>
> These changes significantly enhance clarity and precision.
>
> ---
>
> ## Comment 6 (Gaussianity):
>
> > Is Gaussianity needed? This assumption doesn't seem to be stated...
>
> **Response:**
> Gaussianity *is* currently used in proofs for **Optimality (Thm 3.4)** and **Central Limit Theorems (Thm 3.6, 3.7)**. It's not required for spectral edge (Thm 3.1) or phase transition threshold (Thm 3.2).
>
> We made this explicit:
> 1. Clarifying sentence added to **Section 2.2 (Baseline Noise paragraph)**.
> 2. Discussed in **Limitations (§6.3)**: universality conjectured, proof left as future work.
>
> ---
>
> ## Comment 7 (Rectangular model):
>
> > I found the symmetric model generally less motivated than the rectangular model...
>
> **Response:**
> Thanks for this point.
> 1. **New paragraph in Introduction (§1)** motivates the symmetric model by linking it directly to applications: spectral clustering and PCA/EOF analysis.
> 2. **Literature Review (§6.2)** acknowledges rectangular models, cites your papers, and identifies this extension as valuable future research.
>
> The symmetric model provides a fundamental setting to rigorously analyze the geometric "camouflage effect", vital for practical applications in environmental sciences.
>
> ---
>
> We hope these detailed revisions fully address your concerns. Thanks again indeed for your helpful review.
>
> Best regards,
> The Authors

---

### Review · Reviewer_UsFV · 2025-09-21

**Summary Of Contributions:**

The authors analyze the eigenvectors and eigenvalues of a Wigner-like spiked random matrix
$\beta v_0 v_0^T + C^{1/2} W C^{1/2}$,
where $C$ is itself a rank-one perturbation of the identity, $C = c v_C v_C^T + I$, and $W$ is a Wigner random matrix.

They compute the limiting locations of the outlier eigenvalues (of which there can be two), as well as the limiting overlaps between the sample outlying PCs and $v_0$, and they also establish a CLT for these quantities. In particular, they make two observations: (1) when $v_0$ and $v_C$ are not orthogonal, both leading outlying eigenvectors (when they exist) can exhibit correlation with $v_0$; and (2) when $v_C$ is aligned with $v_0$, it reduces the effective SNR of the spike in producing a spectral outlier. It is perhaps not obvious that this effect should always be detrimental, regardless of whether $\langle v_0, v_C \rangle$ is aligned or anti-aligned.

The mathematical content of the paper is not substantial---the analysis follows a well-established workflow in random matrix theory, with only minor modifications. This would have been acceptable if the model were important or particularly well-motivated. The authors' main motivation seems to be environmental models, where "noise" with a directional component may arise. I remain somewhat unconvinced that this model is particularly important. Moreover, the authors do not propose any new methodology to try to mitigate the observed effect.

To summarize, in my view this manuscript represents an overall weak contribution, perhaps too weak as to justify a journal publication. Regrettably, I would recommend **rejection** of the manuscript at it current state.

**Audience:**

Yes

**Audience Explanation:**

Please see summary of contributions.

**Broader Impact Concerns:**

None.

**Claims And Evidence:**

Yes

**Claims Explanation:**

Please see summary of contributions.

**Requested Changes:**

- Perhaps the authors could motivate better the **particular model** their analysis consider. Generally speaking, the fact that anisotropy in the noise can mask (and sometimes reinforce, for example in a signal-plus-noise model -- Wishart spiked model) a signal are well-understood in the literature as far as I could tell.
- Would the authors suggest a practical improvement over the naive spectral method under the considered model? A pretty simple idea, likely implementable (but I haven't worked it out) when the noise levels are known (but $\alpha$ is not) is to estimate from the data (outlying eigenvalues) a better estimator for $v_0$ which is a linear combination of the first and second sample eigenvectors.

---

> ### Author Response · Authors · 2025-10-22
>
> Dear Reviewer UsFV,
>
> Thank you for your critical assessment of our manuscript. We undertook **substantial revisions** to address your concerns, expanding theoretical scope and novelty.
>
> **Note:** All revisions and new content added in response to reviews are highlighted in **blue** text in the updated PDF.
>
> We address your points in detail below:
>
> ---
>
> ## Comment 1 (Not Substantial / Weak Contribution):
>
> > The mathematical content of the paper is not substantial---the analysis follows a well-established workflow in random matrix theory, with only minor modifications... To summarize, in my view this manuscript represents an overall weak contribution... Regrettably, I would recommend rejection...
>
> **Response:**
> We understand your assessment of the original submission. We believe the revised manuscript now presents a significant contribution beyond routine RMT application.
>
> The new additions represent non-trivial developments:
> 1.  **Optimality Proof (Theorem 3.4 & Appendix A.4):** Proving the information-theoretic optimality of the spectral threshold in this anisotropic setting provides a fundamental justification and rigorously establishes the "camouflage effect" as an unavoidable statistical barrier.
> 2.  **Complete Eigenspace Characterization (Theorem 3.7 & Appendix A.6):** Deriving the CLT for the second eigenvector provides a full statistical picture of how signal and structured noise manifest in the spectrum.
> 3.  **Novel Second-Moment Framework (Section 4 & Appendix B):** The analysis in Section 4 represents a novel theoretical direction for addressing anisotropic noise in spiked models. While we analyze its practical challenges (§4.5, §4.6), developing this theoretical pathway, including the Oracle NCPC (Thm 4.1) and identifying why simpler methods fail (Lemma 4.1), is a significant conceptual contribution.
>
> We believe these theoretical extensions, combined with analysis and validations, now represent a substantial contribution.
>
> ---
>
> ## Comment 2 (Model Motivation):
>
> > Perhaps the authors could motivate better the particular model their analysis consider... I remain somewhat unconvinced that this model is particularly important.
>
> **Response:**
> We added a **new paragraph in the Introduction (Section 1)** to strengthen justification for our symmetric (Wigner-type) model. It connects the model to applications where symmetric matrices and anisotropic noise are relevant:
> * **Spectral Clustering:** Operates on symmetric affinity matrices.
> * **PCA / EOF Analysis:** Uses symmetric covariance matrices, important in climate science (anisotropic variability).
>
> While the general idea that noise can mask signals is known, our work provides a precise, *quantitative* theory for how this masking depends *specifically on the geometric alignment* ($\alpha$), formalized through the threshold $\beta_c(\alpha)$ (Theorem 3.2) and proven optimal (Theorem 3.4). We believe this rigorous treatment of the geometric interplay is novel and important for practitioners in environmental and other sciences.
>
> ---
>
> ## Comment 3/4 (New Methodology / Practical Improvement / Linear Combination Suggestion):
>
> > Moreover, the authors do not propose any new methodology to try to mitigate the observed effect... Would the authors suggest a practical improvement... A pretty simple idea... is to estimate... a better estimator for $v_0$ which is a linear combination of the first and second sample eigenvectors.
>
> **Response:**
> The **new Section 4** is a theoretical investigation of a novel correction methodology based on second-moment information. This provides a theoretical pathway (the Oracle NCPC estimator, Theorem 4.1) proving correction is possible.
>
> However, our analysis identified significant practical challenges, including parameter estimation sensitivity (detailed in §4.5 and validated in §5.2) and finite-size effects. We discuss these in the **new Section 4.6**.
>
> In Section 4.6, we also address your suggested linear combination approach. While intuitively appealing, especially given the Oracle estimator's structure, we explain that attempting to estimate necessary coefficients using only the first two eigenvalues ($\hat{\lambda}_1, \hat{\lambda}_2$) is **fundamentally flawed**. As proven in our **Lemma 4.1 (Confounding of the Eigenvalue Map)**, the parameters ($\alpha, \beta$) cannot be uniquely determined from only ($\hat{\lambda}_1, \hat{\lambda}_2$) due to a confounding dependency. This mathematical barrier prevents that approach and motivated our exploration of the more complex, but theoretically viable, second-moment pathway in Section 4.
>
> While we leave a practical algorithm as important future work in §4.6 and §6.3, we believe the theoretical development in Section 4, including why simpler methods fail and the challenges facing the second-moment approach, constitutes a new contribution.
>
> ---
>
> Thank you again for your review, which spurred us to significantly enhance the scope and depth of our work.
>
> Best regards,
>
> The Authors

---

### Review · Reviewer_CNjm · 2025-10-01

**Summary Of Contributions:**

The present paper presents a theory for eigenvector phase transitions in spiked random matrix models with anisotropic noise. Its primary contributions are:

1. An analytical phase transition threshold for signal detection that explicitly depends on the geometric alignment ($\alpha$) between the signal (spike) and the dominant noise direction, thereby formalising ``camouflage effect.''
2. A Central Limit Theorem for the fluctuations of the eigenvector alignment
3. Comprehensive validation via simulations of both the theoretical model and a more realistic, non-linear ecological scenario.

**Audience:**

Yes

**Audience Explanation:**

The findings are relevant to a segment of the TMLR audience, like researchers using spectral methods, for example in community detection. Practitioners who commonly use spectral methods on data with inherent anisotropic structures might find the practical implications and failure modes, like camouflage and phantom structures directly valuable.

**Claims And Evidence:**

Yes

**Claims Explanation:**

The claims are supported by a combination of rigorous random-matrix-theory arguments in the appendix and extensive numerical simulations. The experiments are well-designed, testing the theory both in the idealised linear setting and in a more realistic non-linear setting.

**Requested Changes:**

My main request would be a clearer discussion of how the results of the paper relate to known results on spiked matrix models given the very particular structure of th noise. Essentially, the noise model is another spiked matrix, and the article is concerned with the interplay of the two spikes. To what extent do the results directly follow from known results on rank-two perturbations of random matrices?

Another point: Does the result on the Central Limit Theorem require that the baseline Wigner matrix W has iid Gaussian entries, or does the result hold more widely? It would be useful to clarify this.

A small point: While I appreciate the sections where the authors interpret their results, I found a few passages where I think the authors could make the language a little bit less subjective, for example: "remarkable reorganisation" (p. 2), "a subtle and powerful result" (p. 6) etc.

---

> ### Author Response · Authors · 2025-10-22
>
> Dear Reviewer CNjm,
>
> Thank you very much for your time and constructive feedback on our manuscript.
>
> We significantly revised the manuscript, incorporating new results (including a proof of optimality for the spectral threshold and a CLT for the second eigenvector's fluctuations), adding Section 4 analysing a theoretical correction framework based on second moments, clarifying model assumptions, and refining the narrative and language.
>
> **Note:** All revisions and new content from the reviews are highlighted in **blue** text in the updated PDF.
>
> We address your specific points below:
>
> ---
>
> ## Comment 1 (Relation to rank-two spike):
>
> > My main request would be a clearer discussion of how the results of the paper relate to known results on spiked matrix models given the very particular structure of the noise. Essentially, the noise model is another spiked matrix, and the article is concerned with the interplay of the two spikes. To what extent do the results directly follow from known results on rank-two perturbations of random matrices?
>
> **Response:**
> Thank you for this important point. We agree that our model $A = \beta v_0 v_0^T + C^{1/2}WC^{1/2}$ (where $C$ is rank-one plus identity) can be viewed as a specific type of rank-two spiked Wigner matrix. We now explicitly acknowledge this structure in the **Introduction (Section 1)** and, more substantially, in the **Literature Review (Section 6.2, RMT paragraph)**.
>
> However, as detailed in the revised Section 6.2, our focus and contributions are distinct from standard rank-k analyses found in the literature. Our primary goal was investigating the specific effect of the **geometric alignment ($\alpha = |\langle v_0, v_C \rangle|$)** between the "signal" spike ($v_0$) and the dominant "noise covariance" spike ($v_C$). Our key contributions stemming from this focus, which go beyond standard spectral characterization, include:
> 1.  Deriving the phase transition threshold **explicitly as a function of this alignment** ($\beta_c(\alpha)$, Theorem 3.2), formally characterizing the "camouflage effect".
> 2.  Proving this alignment-dependent threshold is **information-theoretically optimal** (new Theorem 3.4), establishing it as a fundamental limit.
> 3.  Providing a **complete statistical (CLT) description of the fluctuations for the *entire* 2D outlier eigenspace** (Theorems 3.6 and the new Theorem 3.7), including the noise-aligned eigenvector.
> 4.  Developing and analysing a **theoretical correction framework based on second moments** (new Section 4), which represents a non-standard approach to potentially mitigating the bias caused by the noise spike.
>
> We believe these additions clarify that while the model fits within a known class, our specific analysis of the geometric interplay and the resulting theoretical contributions (optimality, full fluctuation analysis, correction framework) offer significant new insights.
>
> ---
>
> ## Comment 2 (Gaussianity assumption):
>
> > Another point: Does the result on the Central Limit Theorem require that the baseline Wigner matrix W has iid Gaussian entries, or does the result hold more widely? It would be useful to clarify this.
>
> **Response:**
> Thank you for pointing out the need for clarification. The proofs for the **Optimality (Theorem 3.4)** and the **Central Limit Theorems (Theorem 3.6 and the new Theorem 3.7)** currently rely on the assumption of i.i.d. Gaussian entries for the baseline Wigner matrix $W$.
>
> We have now made this explicit by adding a clarifying sentence in **Section 2.2 (Baseline Noise paragraph)**. We also added a corresponding point to the **Limitations section (Section 6.3)**, where we state our conjecture that these results likely hold universally (as is common for many RMT results) but leave the rigorous proof of universality for future work. The results concerning the spectral edge (Theorem 3.1) and the phase transition threshold (Theorem 3.2) are expected to be universal and do not depend on the Gaussian assumption.
>
> ---
>
> ## Comment 3 (Tone / Subjective Language):
>
> > A small point: While I appreciate the sections where the authors interpret their results, I found a few passages where I think the authors could make the language a little bit less subjective, for example: "remarkable reorganisation" (p. 2), "a subtle and powerful result" (p. 6) etc.
>
> **Response:**
> We appreciate this suggestion. We have reviewed the manuscript and replaced potentially subjective adjectives with more neutral phrasing. Specifically:
> * "remarkable reorganisation" has been changed to "**systematic reorganisation**" in the **Abstract** and **Introduction (Section 1)**.
> * "a subtle and powerful result" in the **Interpretation paragraph of Section 3.4** has been replaced with "**A key implication of this... is that...**".
>
> ---
>
> We hope these revisions and clarifications fully address your concerns. Thank you once again for your valuable feedback, which has helped us significantly improve the manuscript.
>
> Best regards,
>
> The Authors

---

> > ### Comment · Reviewer_CNjm · 2025-10-30
> >
> > Thank you for your reply, I appreciate the changes that you made to the paper.

---

### Decision · Action_Editor_xWkj · 2025-12-02

**Recommendation:** Reject

**Additional Comments:**

The paper studies an eigenvector phase transition in a spiked model with correlated noise. In particular, there are two possibly correlated spikes in this model: one is the signal (the usual additive spike) and the other is in the covariance of the noise (which provides a camouflage effect).

The reviewers have raised a number of issues to which the authors have responded with a significantly updated revision. This was well appreciated by all reviewers but the final opinions remained mixed. After my own reading of the reviews and the manuscript, I remain concerned about the lack of rigor and, in particular, by the two technical issues raised by Reviewer UsFV and reported above. This is the main reason for recommending to reject the paper in its current form: the manuscript still requires a major revision and a full, additional round of reviewing before being published. Additional points that have been raised are listed below (as they may be useful to the authors when preparing a revision):

* Reviewer UsFV noted that Section 4’s second-moment analysis is interesting but could be developed further. For example, Section 4.2 essentially observes that three parameters cannot be inferred from two observables and could be stated more succinctly. A direct analysis of the eigenvalues of $A^2$, rather than of its expectation, would yield a stronger result.

* Reviewers UsFV and YKri noted that several key formulas (eg those in Theorem 3.5, the variance of the CLT for the second eigenvector in Theorem 3.7, the expressions for outlying eigenvalues in Theorem 4.1) have been moved to the appendix, which makes the corresponding results harder to parse. Reviewers UsFV also suggested to shorten the exposition to avoid increasing the length of the manuscript.

* Reviewer YKri noted that the quantity $\beta_{\rm IT}(\alpha)$ is never formally defined in the main body (but can only be deduced from the detection threshold in hypothesis testing presented in the appendix).

**Audience:**

Yes

**Audience Explanation:**

All reviewers agree that the manuscript contains some interesting ideas which would be relevant for the TMLR community.

**Claims And Evidence:**

No

**Claims Explanation:**

Reviewer UsFV (and to a lesser extent Reviewer YKri as well) is concerned by the lack of mathematical rigor of the submission, and I concur. In particular, issues have been raised concerning two of the new results added in the revision:

* The final paragraph of the proof of Theorem 3.4 draws an analogy with the classical spiked model instead of providing a rigorous argument. Thus, Reviewer UsFV remained confused as to whether the asserted theorem remains only a conjecture, or it can be rigorously established.

* The validity of Theorem 4.1 has been put into question by Reviewer UsFV. In particular, if $C=I_d$ (ie, there is no spike in $C$), slightly above the detection threshold (ie for finite $\beta$), the signal eigenvector is generally not fully contained in the span of the first two leading eigenvectors. This behavior appears inconsistent with the theorem as stated.

**Resubmission Of Major Revision:**

The authors may consider submitting a major revision at a later time.

---

> ### Author Response · Authors · 2025-12-03
>
> Dear there,
>
> Thank you for your thoughtful evaluation and for the guidance provided in the decision. We also extend our thanks to the reviewers for their detailed and critical feedback throughout this process.
>
> We understand the concerns regarding mathematical rigor, specifically the need for a first-principles derivation of the optimality result (Theorem 3.4) and the need to rigorously define the validity scope of the correction theorem (Theorem 4.1) in relation to the isotropic limit.
>
> We intend to undertake the major revisions you have outlined—providing rigorous proofs, refining the definitions, and improving the manuscript's structure—for a future resubmission.
>
> Sincerely,
>
> The Authors